# Learning from Uncertain Data: From Possible Worlds to Possible Models

Jiongli Zhu[1]    Su Feng[2]    Boris Glavic[3]    Babak Salimi[1]

[1]University of California, San Diego    [2]Nanjing Tech University    [3]University of Illinois, Chicago

## Abstract

We introduce an efficient method for learning linear models from uncertain data, where uncertainty is represented as a set of possible variations in the data, leading to predictive multiplicity. Our approach leverages abstract interpretation and zonotopes, a type of convex polytope, to compactly represent these dataset variations, enabling the symbolic execution of gradient descent on all possible worlds simultaneously. We develop techniques to ensure that this process converges to a fixed point and derive closed-form solutions for this fixed point. Our method provides sound over-approximations of all possible optimal models and viable prediction ranges. We demonstrate the effectiveness of our approach through theoretical and empirical analysis, highlighting its potential to reason about model and prediction uncertainty due to data quality issues in training data.

## 1   Introduction

This paper addresses the challenges of learning from uncertain datasets by employing the framework of *possible world semantics*, a well-established concept in AI and database theory [59, 23, 41, 28]. In this approach, uncertainty in a dataset $D$ is conceptualized through a collection of possible datasets $\{D_1, D_2, \ldots\}$, each representing a potential state of the real world, reflecting variations due to missing entries, errors, inconsistencies, and biases. Given this framework and a learning algorithm, our objective is to construct a set of models $\{f_1, f_2, \ldots\}$, where each model $f_i$ is trained on a corresponding potential dataset $D_i$. This method, that we implement in a system called ZORRO (*ZO*notope-based *R*obustness Analysis for *R*egression with *O*ver-approximations), allows for a thorough evaluation of how data uncertainties affect the robustness, reliability, and fairness of models in predictive modeling and statistical inference, particularly in scenarios where the ground truth is unidentifiable, necessitating consideration of all possible dataset variations.

While the framework of possible world semantics is essential for modeling dataset uncertainties, it poses significant challenges due to the potentially infinite number of scenarios each dataset might represent. Exploring every possibility and training a model for each is impractical. The concept of *model multiplicity*, which highlights situations where models with similar accuracy differ in individual predictions, has gained traction [7, 42], yet it primarily focuses on competing models without addressing the full range of dataset variations. Similarly, *dataset multiplicity* introduced in [47] recognizes data variations due to uncertainty but [47] only proposed a solution for linear models with label errors. Our approach expands these ideas by using possible world semantics to systematically manage uncertainty across all features and labels, thus creating a comprehensive framework for evaluating model robustness amid data uncertainties.

To address the challenges of learning from uncertain datasets, we employ the method of *abstract interpretation* [13]. Utilizing zonotopes—a type of convex polytope well-suited for compactly representing high-dimensional data spaces [74, 4]—we over-approximate the set of all possible dataset variations. This framework allows for the simultaneous symbolic execution of gradient descent

38th Conference on Neural Information Processing Systems (NeurIPS 2024).

across all possible datasets, compactly over-approximating all possible optimal model parameters as a zonotope. We demonstrate that for linear regression with $\ell_2$ regularization (Ridge), our method admits a non-trivial closed-form solution. The zonotope representation of model parameters enables efficient inference, facilitating reasoning about the range of possible predictions or of specific parameters.

**Contributions.** The key contributions of this research are:

1. We introduce an abstract gradient descent algorithm for learning linear regression models from uncertain data. This method over-approximates data variations using zonotopes and symbolically executes gradient descent on all possible datasets concurrently. We define and prove the existence of a fixed point that soundly over-approximates all potential models.
2. Symbolic execution generates intractable polynomial zonotopes for gradients due to non-linear terms and monomial growth that is exponential in the number of iterations. We use linearization and order reduction to compactly over-approximate these polynomial zonotopes using linear zonotopes at each step, introducing an efficient version of abstract gradient descent.
3. The efficient version, however, does not guarantee a fixed point for arbitrary order reduction techniques. To address this, we develop advanced order reduction techniques that ensure fixed points and provide a non-trivial closed-form solution for these fixed points in ridge regression.
4. We implement our approach in a system called ZORRO and use it to evaluate the impact of data uncertainty on linear regression models. Our empirical results and analytical solutions validate the effectiveness of our approach, demonstrating its efficacy in computing prediction ranges and verifying robustness.

**Related Work.**   Predictive multiplicity has shown that a single dataset can produce multiple optimally fitted models due to variations in training processes [8, 11, 44, 66, 69, 9, 15, 71, 42] or modifications to training parameters such as random seeds, data ordering, and hyperparameters. For predictive multiplicity due to missing data, Khosravi et al.[34] address the issue using a probabilistic method that computes expected predictions for all possible imputations. Our approach can be seen as an extreme case of multiple imputation [62, 61], where we consider all possible data variations rather than just a few plausible scenarios. Meyer et al. [47] recently introduced the concept of dataset multiplicity, using possible world semantics to model how uncertain, biased, or noisy training data can lead to predictive multiplicity. However, their focus is on uncertainty in training labels, and they use interval arithmetic for over-approximation of prediction intervals for linear regression. In contrast, our approach handles arbitrary uncertainty in features and labels during both training and testing using zonotope-based learning for over-approximation of prediction ranges and model parameters. We show that interval arithmetic fails to provide tight prediction ranges even for uncertainty in labels.

Our work is broadly related to robust model learning, which ensures robustness against data quality issues such as attacks [31, 76, 70, 60, 55, 52, 29]. Distributional robustness [6, 53, 65] studies model reliability against varying data distributions, while robust statistics [19, 18] examines model performance under outliers or data errors. Our approach provides exact provable robustness guarantees by exploring the entire range of models under extreme dataset variations, which is crucial for individual-level predictions and reasoning about the robustness of specific parameters.

Our work is also related to robustness certification, which certifies ML models' robustness against data perturbations and uncertainties [27, 68, 50]. These efforts mainly focus on test-time robustness, validating predictions for inputs in the vicinity of a test sample. In contrast, we address training-time robustness, considering the effects of possible datasets on training models. Closest to our approach is the work by Meyer et al. [46] for decision trees and Karlas et al. [33] for nearest neighbor classifiers. We use zonotopes to over-approximate prediction ranges for linear regression, generating robustness certificates. While zonotopes have been used for test-time robustness [49, 20, 22, 51], our work is the first to apply zonotopes for training-time robustness for an iterative learning algorithm.

Approaches for uncertainty quantification (UQ) aim to understand the range of outcomes a model may produce using Bayesian methods, ensembling, conformal prediction, and bootstrapping [54, 43, 35, 17, 16]. UQ focuses on epistemic and aleatoric uncertainty, stemming from insufficient data, noisy data, or uncertainty about the model parameters, and does not account for uncertainty due to systematic data quality issues, such as non-random data errors or missing values [25], which induce a multiplicity of possible datasets. In this case, UQ methods might underestimate the uncertainty as they rely on critical assumptions. Bayesian methods, for instance, require correctly specified priors to accurately model uncertainty, often failing under conditions with unknown or erroneous

priors [73, 67, 72], while conformal prediction (CP) assumes data exchangeability—a condition that breaks down when data errors are systematic [21, 75]. In contrast, our approach addresses this distinct challenge by computing sound over-approximations that guarantee complete coverage of potential predictions across all variations of the dataset. This sound coverage is essential in high-stakes settings such as evaluating the robustness of predictive models for medical use.

## 2 Notation, Problem Formulation and Background

Denote a training dataset $\boldsymbol{D} = (\boldsymbol{X}, \boldsymbol{y})$ with $\boldsymbol{X} = [\boldsymbol{x}_1 \quad \cdots \quad \boldsymbol{x}_n]^T \in \mathbb{R}^{n \times d}$ as the matrix of features, and $\boldsymbol{y} = [y_1 \quad \cdots \quad y_n]^T \in \mathbb{R}^n$ as the corresponding ground truth labels. Let $f(\boldsymbol{x}; \boldsymbol{w})$ be a model parameterized by $\boldsymbol{w} \in \mathbb{R}^p$ that maps an input data point to a label. A learning algorithm $\mathcal{A}$ maps a training dataset $\boldsymbol{D}$ to the parameters of the trained model, $\boldsymbol{w}^* = \mathcal{A}(\boldsymbol{D})$. Given a test dataset $\mathbf{X}_{\text{test}} \in \mathbb{R}^{n \times d}$, for any test sample $\boldsymbol{x}$ from $\mathbf{X}_{\text{test}}$, the function $f$ computes a prediction $\hat{y} = f(\boldsymbol{x}; \boldsymbol{w}^*)$.

### 2.1 Learning Possible Models from Possible Worlds

We use possible world semantics to represent the uncertainty in a dataset $\boldsymbol{D}$.

**Definition 2.1** (Possible Datasets). *Given an uncertain dataset $\boldsymbol{D}$, the uncertainty in $\boldsymbol{D}$ can be represented by a set of possible datasets $\boldsymbol{D}^{\odot}$: $\boldsymbol{D}^{\odot} = \{\boldsymbol{D}_1, \boldsymbol{D}_2, \ldots\}$.*

Each dataset $\boldsymbol{D}_i \in \boldsymbol{D}^{\odot}$ is a "possible world", i.e., a hypothetical variation of the dataset $\boldsymbol{D}$ that could potentially exist in the real world based on our knowledge about the uncertainty in $\boldsymbol{D}$.

**Example 2.2.** *Consider an e-commerce dataset $\boldsymbol{D}$ where some product price is missing, meaning the exact price is unknown. Using possible world semantics, we represent this uncertainty with a set of possible datasets $\boldsymbol{D}^{\odot}$, each containing a possible clean price, which could be obtained from prices of the same items on the market.*

In App. D, we discuss construction methods for common data quality issues. Our goal is to efficiently construct the set of all possible models from uncertain data and understand their behavior in making predictions.

**Definition 2.3** (Possible Models and Prediction Range). *Given a set of possible datasets $\boldsymbol{D}^{\odot}$ associated with an uncertain dataset $\boldsymbol{D}$, the possible models, denoted $f^{\odot}$, are obtained by applying the learning algorithm $\mathcal{A}$ to each training dataset $\boldsymbol{D}_i$ within $\boldsymbol{D}^{\odot}$ to obtain the set of all possible optimal model parameters $\boldsymbol{w}^{\odot *}$, i.e.,*

$$\boldsymbol{w}^{\odot *} = \{\boldsymbol{w}_i^* \mid \boldsymbol{w}_i^* = \mathcal{A}(\boldsymbol{D}_i), \boldsymbol{D}_i \in \boldsymbol{D}^{\odot}\}$$

*For a test data point $\boldsymbol{x}$, the viable prediction range $V(\boldsymbol{x})$ is defined as the interval between the least upper bound and the greatest lower bound of the outputs produced by all models in $f^{\odot}$, i.e.,*

$$V(\boldsymbol{x}) = \left[ \inf_{\boldsymbol{w}^* \in \boldsymbol{w}^{\odot *}} f(\boldsymbol{x}, \boldsymbol{w}^*), \sup_{\boldsymbol{w}^* \in \boldsymbol{w}^{\odot *}} f(\boldsymbol{x}, \boldsymbol{w}^*) \right]$$

This prediction range quantifies the minimum and maximum predictions that can be expected for $\boldsymbol{x}$, highlighting the variability in model outputs due to differences in the training data. Our framework supports uncertainty in training and test data. We discuss test data uncertainty in App. F.3.

### 2.2 Sound Approximation of Possible Models with Abstract Interpretation

The set of all possible datasets associated with uncertain data can be intractable. We use *abstract interpretation* [12] to over-approximate sets of elements of a concrete domain $\mathbb{D}$ (the training data and model weights) with elements from an abstract domain $\mathbb{D}^{\sharp}$. Specifically, we use the abstract domain of zonotopes, a type of convex polytope, to over-approximate the possible datasets $\boldsymbol{D}^{\odot}$ using a zonotope $\boldsymbol{D}^{\sharp}$ that has a compact symbolic representation. Instead of applying the learning algorithm $\mathcal{A}$ to each possible dataset to compute all possible optimal model parameters $\boldsymbol{w}^{\odot *}$, we develop an abstract learning algorithm $\mathcal{A}^{\sharp}$ that operates directly on the abstract domain of zonotopes. Given $\mathbf{D}^{\sharp}$, $\mathcal{A}^{\sharp}$ generates a zonotope $\boldsymbol{w}^{\sharp} = \mathcal{A}^{\sharp}(\mathbf{D}^{\sharp})$ that over-approximates $\boldsymbol{w}^{\odot *}$ (demonstrated in the graph on the right). Intuitively, this represents the symbolic execution of the learning algorithm across all possible datasets simultaneously.

**Definition 2.4** (Abstract Domain). *Let $\mathbb{D}$ be a concrete domain. An* abstract domain *for $\mathbb{D}$ is a set $\mathbb{D}^{\sharp}$ paired with two functions:*

$$\textbf{Abstraction } \alpha : \mathcal{P}(\mathbb{D}) \to \mathbb{D}^{\sharp} \qquad \textbf{Concretization } \gamma : \mathbb{D}^{\sharp} \to \mathcal{P}(\mathbb{D})$$

*which satisfy the following condition for any subset $S \subseteq \mathbb{D}$: $\gamma(\alpha(S)) \supseteq S$. Two abstract elements $d_1$ and $d_2$ are equivalent, written as $d_1 \simeq_{\sharp} d_2$, if $\gamma(d_1) = \gamma(d_2)$.*

Def. 2.4 ensures that the abstract element $\alpha(S)$ associated with a set $S$ through application of the abstraction function $\alpha$ encodes an over-approximation of $S$. We will use an abstract element $\mathbf{D}^{\sharp} = \alpha(\boldsymbol{D}^{\odot})$ to over-approximate the possible worlds of an uncertain training dataset $\boldsymbol{D}^{\odot}$. We discuss abstraction functions $\alpha$ for specific types of training data uncertainty in App. D.

**Definition 2.5** (Abstract Transformer). *Consider a function $F : \mathbb{D}_1 \to \mathbb{D}_2$ on concrete domains $\mathbb{D}_1$ and $\mathbb{D}_2$. An* abstract transformer $F^{\sharp} : \mathbb{D}_1^{\sharp} \to \mathbb{D}_2^{\sharp}$ *over-approximates $F$ in the abstract domain:*

$$\forall S \in \mathcal{P}(\mathbb{D}) : \gamma\left(F^{\sharp}(\alpha(S))\right) \supseteq F(S)$$

*An abstract transformer is* exact *(does not loose precision) if $\forall d \in \mathbb{D}^{\sharp} : \gamma\left(F^{\sharp}(d)\right) = F(\gamma(d))$.*

Importantly, (exact) abstract transformers compose (see App. B.2, Prop. B.1) and, thus, we can construct an abstract transformer for complex functions from simpler parts.

To over-approximate the set of possible model parameters $\boldsymbol{w}^{\odot *}$, we will develop an abstract transformer $\mathcal{A}^{\sharp}$ for the learning algorithm $\mathcal{A}$ to get $\gamma\left(\mathcal{A}^{\sharp}(\mathbf{D}^{\sharp})\right) \supseteq \boldsymbol{w}^{\odot *}$.

**Symbolic Abstract Domains and Zonotopes.** We consider a symbolic abstract domain $\Psi$ of vectors and matrices (marked with $\cdot^{\sharp}$) with elements that are polynomials $\psi$ over variables $\mathcal{E} = \{\epsilon_i\}$. The concretization of a polynomial $\psi$ is the result of evaluating $\psi$ on all assignments $e : \mathcal{E} \to [-1, 1]$, encoded as vectors $[-1, 1]^{|\mathcal{E}|}$: $\gamma(\psi) = \{\psi(e) \mid e \in [-1, 1]^{|\mathcal{E}|}\}$. We lift concretization to vectors and matrices through point-wise application. Such an object $\boldsymbol{z}^{\sharp}$ is typically referred to as a *polynomial zonotope* or *zonotope* if all symbolic expressions are linear (see App. C.2). The concretization of $\boldsymbol{z}^{\sharp}$ is: $\gamma(\boldsymbol{z}^{\sharp}) = \{\boldsymbol{z}^{\sharp}(e) \mid e \in [-1, 1]^{|\mathcal{E}|}\}$.

# 3 Exact Abstract Transformers for Learning Linear Models

Given an uncertain training dataset $\boldsymbol{D}^{\odot}$, we aim to over-approximate the set of possible optimal linear models $\boldsymbol{w}^{\odot *} = \{\boldsymbol{w}_1^*, \boldsymbol{w}_2^*, \ldots\}$, where $\boldsymbol{w}_i^* \in \mathbb{R}^p$ represents the optimal parameters of a linear model trained on $\boldsymbol{D}_i \in \boldsymbol{D}^{\odot}$. These optimal parameters are the fixed point of the sequence $\{\boldsymbol{w}_i^k\}_{k=0}^{\infty}$ generated by: $\boldsymbol{w}_i^{k+1} = \Phi(\boldsymbol{w}_i^k)$ where the operator $\Phi : \mathbb{R}^p \to \mathbb{R}^p$ captures one step of gradient descent, i.e., $\Phi(\boldsymbol{w}) = \boldsymbol{w} - \eta \nabla L(\boldsymbol{w})$, for a learning rate $\eta$ and a loss function $L(\boldsymbol{w})$ [57].

In the abstract domain, we use the zonotope representations $\mathbf{D}^{\sharp}$ and $\boldsymbol{w}^{\sharp}$ to abstract the possible datasets $\boldsymbol{D}^{\odot}$ and the set of possible model weights $\boldsymbol{w}^{\odot *}$. While it is theoretically possible to compute symbolic expressions for the standard closed form solution for linear regression, this can result in large expressions that contain fractions with polynomial numerators and denominators and, thus, computing prediction intervals based on such expressions is computationally infeasible (see App. L). Instead, we over-approximate the optimal parameters using an abstract operator $\Phi_{exact}^{\sharp} : \boldsymbol{w}^{\sharp} \to \boldsymbol{w}^{\sharp}$ that generates a sequence of abstract elements $\{\boldsymbol{w}^{\sharp\,k}\}_{k=0}^{\infty}$:

$$\boldsymbol{w}^{\sharp\,k+1} = \Phi_{exact}^{\sharp}(\boldsymbol{w}^{\sharp\,k}),$$

where the abstract operator $\Phi_{exact}^{\sharp}$ given by $\Phi_{exact}^{\sharp}(\boldsymbol{w}^{\sharp}) = \boldsymbol{w}^{\sharp} - \eta \nabla L(\boldsymbol{w}^{\sharp})$ captures one gradient descent step in the abstract domain. Specifically, for any loss function $L$ whose gradient $\nabla L$ consists of linear or polynomial expressions, such as the mean squared error (MSE) loss, $\Phi_{exact}^{\sharp}$ is an exact abstract transformer. This follows from the existence of exact abstract transformers for addition and multiplication over polynomial zonotopes [37] and the fact that abstract transformers compose (see App. E, Prop. B.1 and Prop. E.1).

**Proposition 3.1.** *The abstract gradient descent operator $\Phi_{exact}^{\sharp}$ is an exact abstract transformer for the concrete gradient descent operator $\Phi$. Formally, for any abstract $\boldsymbol{w}^{\sharp}$,*

$$\gamma\left(\Phi_{exact}^{\sharp}(\boldsymbol{w}^{\sharp})\right) = \Phi(\gamma(\boldsymbol{w}^{\sharp})),$$

Unfortunately, the sequence $\{\boldsymbol{w}^{\sharp\,k}\}_{k=0}^{\infty}$ does not have a fixed point as the highest-order symbolic terms in $\boldsymbol{w}^{\sharp\,k}$ are multiplied with symbolic terms in the gradient. Thus, $\boldsymbol{w}^{\sharp\,k+1} = \Phi_{exact}^{\sharp}(\boldsymbol{w}^{\sharp\,k})$ has terms of higher orders than any term in $\boldsymbol{w}^{\sharp\,k}$. To identify when the abstract model weights represent all fixed points in the concrete domain, we define abstract fixed points using concretization.

**Definition 3.2** (Fixed Point of Abstract Gradient Descent). *An abstract model weight $\boldsymbol{w}^{\sharp\,*}$ is a fixed point for an abstract transformer $F^{\sharp}$ for the gradient descent operator $\Phi$ if,*

$$\gamma\left(\boldsymbol{w}^{\sharp\,*}, \mathbf{D}^{\sharp}\right) = \gamma\left(F^{\sharp}(\boldsymbol{w}^{\sharp\,*}), \mathbf{D}^{\sharp}\right).$$

*Here, $\gamma\left(\cdot, \cdot\right)$ denotes the joint concretization of $\boldsymbol{w}^{\sharp\,*}$ and $\mathbf{D}^{\sharp}$ (see App. C.2 for more details).*

Any abstract fixed point $\boldsymbol{w}^{\sharp\,*}$ over-approximates all possible optimal model weights $\boldsymbol{w}^{\odot*}$.

**Proposition 3.3.** *Consider an abstract transformer $F^{\sharp}$ for the gradient descent operator $\Phi$. Let $\boldsymbol{w}^{\sharp\,*}$ be a fixed point according to Def. 3.2. Then it holds that $\gamma\left(\boldsymbol{w}^{\sharp\,*}\right) \supseteq \boldsymbol{w}^{\odot*}$.*

As we demonstrate in App. F.3, $\boldsymbol{w}^{\sharp\,*}$ can also be used to compute prediction ranges. In general, an abstract fixed point is not guaranteed to exist for every abstract transformer for $\Phi$. While $\Phi_{exact}^{\sharp}$ as defined above has a fixed point (see Prop. F.1 in the appendix), two significant barriers remain. First, the representation size of the polynomial zonotope $\boldsymbol{w}^{\sharp\,i}$ for abstract model parameters grows exponentially in $i$, the number of the steps of abstract gradient descent. Additionally, testing for convergence is challenging, as checking for containment is already NP-hard for linear zonotopes [39].

## 4 Efficient Sound Approximation for Learning Linear Models

In this section, we present an efficient abstract gradient descent method for ridge regression that guarantees a fixed point and addresses the exponential growth of generated abstract model parameters. The core idea is to use the linearization and order reduction techniques introduced in the following to deal with intractable polynomial zonotopes. We develop a specific order reduction technique that admits a fixed point and enables us to find a closed-form solution for the fixed point.

### 4.1 Gradient Descent With Order Reduction

A *linearization* operator $\mathbf{L}$ maps a polynomial zonotope $\boldsymbol{z}^{\sharp}$ to a linear zonotope $\boldsymbol{\ell}^{\sharp}$ that over-approximates $\boldsymbol{z}^{\sharp}$: $\gamma\left(\mathbf{L}(\boldsymbol{z}^{\sharp})\right) \supseteq \gamma\left(\boldsymbol{z}^{\sharp}\right)$. An *order reduction* operator $\mathbf{R}$ takes a linear zonotope $\boldsymbol{\ell}^{\sharp}$ as input and returns another linear zonotope $\boldsymbol{\ell}^{\sharp}$ of reduced order (representation size) such that: $\gamma\left(\mathbf{R}(\boldsymbol{\ell}^{\sharp})\right) \supseteq \gamma\left(\boldsymbol{\ell}^{\sharp}\right)$. Specific linearization and order reduction operators are discussed in App. G. The graph on the right shows the progression of the zonotope of parameters toward a fixed point from a random initialization by applying 200 iterations of the following abstract gradient descent operator $\Phi^{\sharp}$. The number inside each zonotope indicates its corresponding iteration. After 200 iterations the resulting zonotope is close to a fixed point shown in red and computed using techniques we introduce in the following. This operator first linearizes the gradient zonotope using $\mathbf{L}$, subtracts it from the current abstract model parameters $\boldsymbol{w}^{\sharp}$, and then reduces the order of the resulting zonotope using $\mathbf{R}$:

$$\Phi^{\sharp}(\boldsymbol{w}^{\sharp}) = \mathbf{R}\left(\boldsymbol{w}^{\sharp} - \mathbf{L}\left(\eta\nabla L(\boldsymbol{w}^{\sharp})\right)\right), \tag{1}$$

This operator is an abstract transformer for the gradient descent operator (Prop. H.1). While each step can be computed efficiently by bounding the order of the resulting zonotope, $\Phi^{\sharp}$ may not always converge to a fixed point. The graph on the left illustrates a real example where, despite the existence of a zonotope containing all optimal possible parameters (shown again in red), abstract gradient descent with linearization and order reduction keeps diverging, generating larger and larger zonotopes due to over-approximation error. In the following, we develop an order reduction operator that ensures abstract gradient descent with linearization converges to a fixed point for linear regression with $\ell_2$ regularization. Additionally, we derive a closed-form solution for this fixed point. The red dotted zonotope in the graphs represents the fixed point generated by our method.

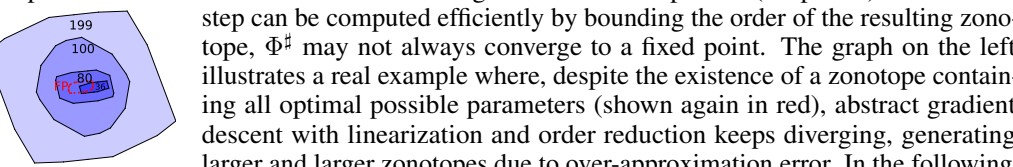

**Decomposition of Gradients.** A core idea in our approach is to decompose the abstract dataset into real and symbolic parts and use an order reduction operator that allows for the decomposition of the abstract gradient descent operator into components processing the real and symbolic parts in a specific evaluation order. Specifically, we observe that the components of the abstract training dataset $\mathbf{D}^\sharp = (\boldsymbol{X}^\sharp, \boldsymbol{y}^\sharp)$ can be decomposed into a sum of a real part (without error symbols) and a symbolic part (containing only symbolic terms). Furthermore, the abstract model weight $\boldsymbol{w}^\sharp$ produced by $\Phi^\sharp$ can be decomposed as shown below. $\boldsymbol{w}_R$ is the real part of the zonotope and does not contain any error symbols, $\boldsymbol{w}_D^\sharp$ contains only symbolic terms with error symbols that also occur in $\mathbf{D}^\sharp$, and $\boldsymbol{w}_N^\sharp$ contains only symbolic terms introduced by linearization and order reduction, hence does not contain any error symbols from $\mathbf{D}^\sharp$ since these methods introduce fresh symbols.

$$\boldsymbol{X}^\sharp = \boldsymbol{X}_R + \boldsymbol{X}_S^\sharp, \qquad \boldsymbol{y}^\sharp = \boldsymbol{y}_R + \boldsymbol{y}_S^\sharp \qquad \boldsymbol{w}^\sharp = \boldsymbol{w}_R + \boldsymbol{w}_D^\sharp + \boldsymbol{w}_N^\sharp$$

The key observation that enables our approach is that one step of abstract gradient descent can now be decomposed into three components:

$$\Phi^\sharp(\boldsymbol{w}^\sharp) = \Phi_R(\boldsymbol{w}_R) + \Phi^\sharp{}_D(\boldsymbol{w}_R, \boldsymbol{w}_D^\sharp) + \Phi^\sharp{}_N(\boldsymbol{w}_R, \boldsymbol{w}_D^\sharp, \boldsymbol{w}_N^\sharp) \tag{2}$$

where

$$\Phi_R(\boldsymbol{w}_R) = \boldsymbol{w}_R - \eta \cdot \left( \frac{2}{n}(\boldsymbol{X}_R^T \boldsymbol{X}_R \boldsymbol{w}_R - \boldsymbol{X}_R^T \boldsymbol{y}_R) + 2\lambda \boldsymbol{w}_R \right)$$

$$\Phi^\sharp{}_D(\boldsymbol{w}_R, \boldsymbol{w}_D^\sharp) = \boldsymbol{w}_D^\sharp - \eta \cdot \left( (2\lambda I + \frac{2}{n}\boldsymbol{X}_R^T \boldsymbol{X}_R)\boldsymbol{w}_D^\sharp + \frac{2}{n}(\boldsymbol{X}_R^T \boldsymbol{X}_S^\sharp + \boldsymbol{X}_S^{\sharp T}\boldsymbol{X}_R)\boldsymbol{w}_R - \frac{2}{n}\boldsymbol{X}_S^\sharp \boldsymbol{y}_R - \frac{2}{n}\boldsymbol{X}_R^T \boldsymbol{y}_S^\sharp \right)$$

$$\Phi^\sharp{}_N(\boldsymbol{w}_R, \boldsymbol{w}_D^\sharp, \boldsymbol{w}_N^\sharp) = \mathbf{R}\left( \boldsymbol{w}_N^\sharp - \eta \cdot \left( (2\lambda I + \frac{2}{n}\boldsymbol{X}_R^T \boldsymbol{X}_R)\boldsymbol{w}_N^\sharp \right.\right.$$
$$\left.\left. + \mathbf{L}\left( \frac{2}{n}\left( (\boldsymbol{X}_R^T \boldsymbol{X}_S^\sharp + \boldsymbol{X}_S^{\sharp T}\boldsymbol{X}_R)(\boldsymbol{w}_D^\sharp + \boldsymbol{w}_N^\sharp) + \boldsymbol{X}_S^{\sharp T}\boldsymbol{X}_S^\sharp(\boldsymbol{w}_R + \boldsymbol{w}_D^\sharp + \boldsymbol{w}_N^\sharp) - \boldsymbol{X}_S^{\sharp T}\boldsymbol{y}_S^\sharp \right) \right) \right) \right)$$

$\Phi_R(\boldsymbol{w}_R)$, the *real part updater*, only relies on and updates $\boldsymbol{w}_R$, the real part of the abstract weight, and coincides with the abstract gradient operator $\Phi$ on the real part of abstract weights. $\Phi^\sharp{}_D(\boldsymbol{w}_R, \boldsymbol{w}_D^\sharp)$, the *symbolic data-dependent updater*, which is a function of $\boldsymbol{w}_D^\sharp$ itself and

$$\boldsymbol{w}_R \Longrightarrow \Phi_R(\boldsymbol{w}_R)$$
$$\boldsymbol{w}_D^\sharp \quad\searrow\quad \Phi^\sharp{}_D(\boldsymbol{w}_R, \boldsymbol{w}_D^\sharp)$$
$$\boldsymbol{w}_N^\sharp \longrightarrow \Phi^\sharp{}_N(\boldsymbol{w}_R, \boldsymbol{w}_D^\sharp, \boldsymbol{w}_N^\sharp)$$

$\boldsymbol{w}_R$, updates $\boldsymbol{w}_D^\sharp$ based on some linear terms and consists of symbols that come from the abstract dataset $\mathbf{D}^\sharp$ and does not include any non-linear terms; hence applying this does not generate higher-order terms, so the output is a linear zonotope, and no linearization is needed. Additionally, no order reduction is needed since the number of error symbols remains constant. $\Phi^\sharp{}_N(\boldsymbol{w}_R, \boldsymbol{w}_D^\sharp, \boldsymbol{w}_N^\sharp)$, the *symbolic data-independent updater*, is a function of $\boldsymbol{w}_N^\sharp$ itself, $\boldsymbol{w}_D^\sharp$, and $\boldsymbol{w}_R$. It consists of the non-linear terms in the gradient, hence performs linearization to over-approximate higher-order terms and performs order reduction to reduce the number of error symbols, which would otherwise grow exponentially with the number of iterations. Therefore, it generates the updated $\boldsymbol{w}_N^\sharp$ as a linear zonotope that does not share any symbol with $\mathbf{D}^\sharp$, i.e., only consists of fresh symbols introduced by linearization and order reduction. See Appendix I for formal statements and proofs for this section.

**Proposition 4.1.** *Any abstract model weights* $\boldsymbol{w}^{\sharp *} = \boldsymbol{w}_R^* + \boldsymbol{w}_D^{\sharp *} + \boldsymbol{w}_N^{\sharp *}$ *is a fixed point for the abstract gradient descent operator* $\Phi^\sharp$ *if the following conditions are satisfied:*

$$\boldsymbol{w}_R^* = \Phi_R(\boldsymbol{w}_R^*), \qquad \boldsymbol{w}_D^{\sharp *} = \Phi^\sharp{}_D(\boldsymbol{w}_R^*, \boldsymbol{w}_D^{\sharp *}) \qquad \boldsymbol{w}_N^{\sharp *} \simeq_\sharp \Phi^\sharp{}_N(\boldsymbol{w}_R^*, \boldsymbol{w}_D^{\sharp *} \boldsymbol{w}_N^{\sharp *}) \tag{3}$$

---

**Algorithm 1:** Abstract Learning

**Input:** abstract dataset $\mathbf{D}^\sharp$, learning rate $\eta$, regularization coeff. $\lambda$, transformation $\boldsymbol{A}$

$\boldsymbol{w}_R^* \leftarrow \texttt{closedFormReal}(\mathbf{D}^\sharp, \lambda)$
$\boldsymbol{w}_D^{\sharp *} \leftarrow \texttt{closedFormSymb}(\mathbf{D}^\sharp, \lambda, \boldsymbol{w}_R^*)$
$\Xi \leftarrow \texttt{genNonDataEq}(\mathbf{D}^\sharp, \lambda, \eta, \boldsymbol{w}_R^*, \boldsymbol{w}_D^{\sharp *})$
$\boldsymbol{w}_N^{\sharp *} \leftarrow \texttt{solveFixedPointEq}(\Xi, \lambda, \eta)$
$\boldsymbol{w}^{\sharp *} \leftarrow \boldsymbol{w}_R^* + \boldsymbol{w}_D^{\sharp *} + \boldsymbol{w}_N^{\sharp *}$
**return** $\boldsymbol{w}^{\sharp *}$

---

We summarize the fixed point construction process in Alg. 1. The algorithm takes as input an abstract dataset $\mathbf{D}^\sharp$, a learning rate $\eta$, a regularization coefficient $\lambda$, and a transformation matrix $\boldsymbol{A}$. The construction order is: first $\boldsymbol{w}_R^*$, then $\boldsymbol{w}_D^{\sharp *}$, and finally $\boldsymbol{w}_N^{\sharp *}$, each step building on the previous results. Closed-form solutions for $\boldsymbol{w}_R^*$ and $\boldsymbol{w}_D^{\sharp *}$ exist, as detailed in Lem. I.4. The fixed point of $\Phi_R$ matches the fixed point of gradient descent on the real part of the abstract data, independent of other parts. Given $\boldsymbol{w}_R^*$, $\boldsymbol{w}_D^{\sharp *}$ is obtained

by solving a system of linear equations from Eq. (3), ensuring zonotope containment by equalizing the coefficients of the same error symbols, which is a sufficient condition for zonotope equivalence. This involves solving a system of $m$ equations, where $m$ is the number of error symbols in $\mathbf{D}^{\sharp}$.

To construct $\boldsymbol{w}_N^{\sharp *}$ in Eq. (3), which involves zonotope equivalence, we encounter a challenge because the RHS and LHS have different error symbols, making symbolic enforcement of equivalence inapplicable. We exploit the properties of the order reduction operator to create a system of linear equations whose solution yields a fixed point for $\Phi^{\sharp}_N$. Re-

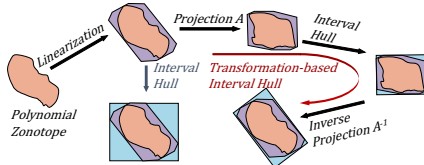

call that this component uses linearization to handle non-linear terms and applies order reduction to efficiently manage the order of the resulting linear zonotope. As shown on the right side, the order reduction operator in $\Phi^{\sharp}_N$ utilizes a transformation matrix $\boldsymbol{A}$ to project the higher-order linear zonotope into a new space and then over-approximates it using an interval hull—a zonotope that is a box enclosing the input higher-order zonotope in the projected space—and finally projects this box back into the original space using $\boldsymbol{A}^{-1}$. The core idea of transformation-based order reduction [38] is that interval hulls provide a better over-approximation in the projected space, where the shape of the input zonotope is closer to a box than in the original space, as shown in the graph on the right.

$$\Phi_N^{\#}(\quad \cdot \quad , \underset{\substack{w_D^{\#*} \\ \text{(known)}}}{\bigcirc} , \underset{w_N^{\#*}}{\diamondsuit}) \simeq_{\#} \diamondsuit$$
$$\underset{\substack{w_R^* \\ \text{(known)}}}{}$$

At a fixed point, the input and output interval hulls of $\Phi^{\sharp}_N$ should be equivalent in the projected space. This equivalence can be effectively enforced by checking for equal edge lengths along each dimension, which is sufficient for two interval hulls to be equivalent. This translates to a system of $k$ equations, where $k$ is the number of model parameters, hence can be done efficiently. We show that this system of equations is guaranteed to have a solution. We are now ready to state our main technical result.

**Theorem 4.2** (Correctness of Alg. 1). *Given a set of possible training datasets $\boldsymbol{D}^{\odot}$ associated with uncertain data $\boldsymbol{D}$, and an appropriate abstraction function $\alpha$ in the zonotope abstract domain, and given a regularization coefficient $\lambda$, Alg. 1, when provided with $\mathbf{D}^{\sharp} = \alpha(\boldsymbol{D}^{\odot})$ as input, computes abstract model parameters $\boldsymbol{w}^{\sharp *}$ such that:*

$$\gamma\left(\boldsymbol{w}^{\sharp *}\right) \supseteq \boldsymbol{w}^{\odot *},$$

*where $\boldsymbol{w}^{\odot *}$ denotes the set of all optimal model parameters corresponding to $\boldsymbol{D}^{\odot}$ for linear regression with $\ell_2$ regularization with regularization coefficient $\lambda$.*

## 5 Experiments

We implement ZORRO using SymPy [45], a Python library for symbolic computations and evaluate the system on two key applications: (1) computing prediction ranges and robustness certification for linear models trained on uncertain data, and (2) robustness of model weights for causal inference using linear models as a case study. We also measured the performance of ZORRO under varying conditions, including varying the degree of training data uncertainty. All our experiments are performed on a single machine with an Apple M1 chip, 8 cores, and 16 GB RAM. Experiments are repeated 5 times with different random seeds, and we report the mean (error bars denote $3\sigma$). The code is shared at https://github.com/lodino/Zorro.

**Datasets, Baselines, and Metrics.** For robustness verification we use regression tasks: for *MPG* [58] (392 instances) we predict fuel consumption based on car features (cylinders, horsepower, weight); for *Insurance* [30] (1338 instances) we predict medical insurance charges based on demographics (age, gender, BMI), habits (smoking), and geographical features. We use a 80:20 train-test split and inject random errors to the training data varying (i) the *Uncertain Data Percentage*, the percentage of instances that have uncertain features / labels, and (ii) the *Uncertainty Radius*, the difference between the minimum and maximum possible value of an uncertain feature expressed as a fraction of the feature's domain. There is no direct baseline capturing prediction ranges for learning linear regression models from uncertain data. Most existing works in robustness certification focuses on test-time robustness (cf. Sec. 1). The exception is [47], which only supports uncertainty in labels using interval arithmetic. We compare ZORRO against this approach for robustness certification, referred to as MEYER in the following, only for training label uncertainty. For robustness verification, a prediction is *robust* if the size of the prediction interval is smaller than a given threshold. We use

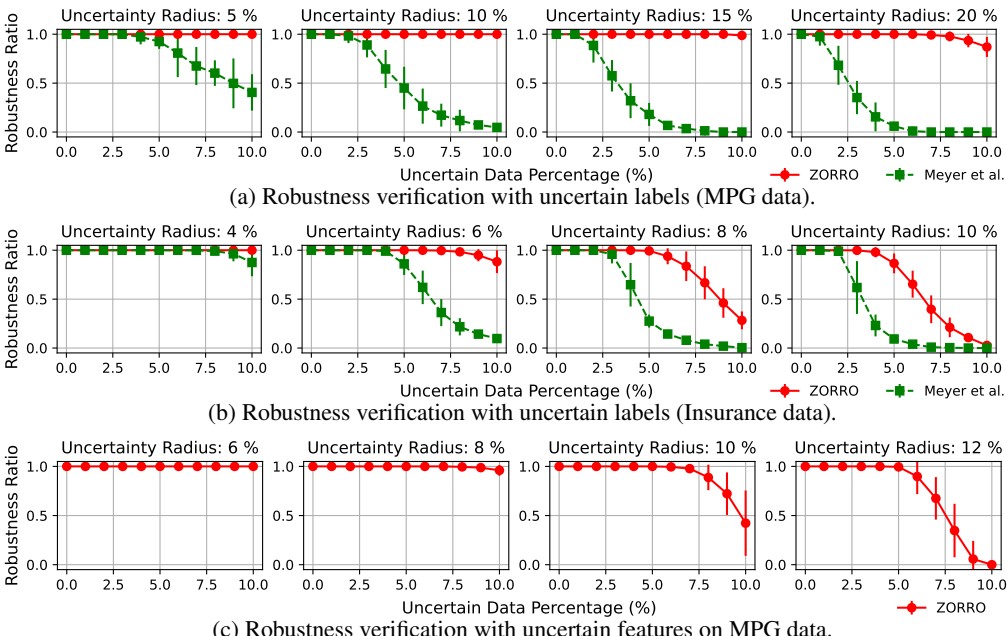

(a) Robustness verification with uncertain labels (MPG data).

(b) Robustness verification with uncertain labels (Insurance data).

(c) Robustness verification with uncertain features on MPG data.

Figure 1: Robustness verification on using intervals (MEYER [47]) and zonotopes (ZORRO).

the *robustness ratio* which is the fraction of the test data receiving robust predictions as a metric in all robustness verification experiments. The robustness threshold is set to 5% of the label range for the *MPG* data, and 0.8% of the label range for the *Insurance* data. Additionally, we assess the *worst-case test loss* using certain test data and uncertain model weights trained from uncertain training data.

## 5.1 Robustness Verification

**Prediction Robustness (Uncertain Labels).** Fig. 1(a)(b) compares ZORRO with the baseline MEYER [47] on a setting where only training labels suffer from uncertainty. We vary the uncertainty radius and uncertain data percentage. As both systems provide sound over-approximations of prediction ranges, they may underestimate a model's robustness. As shown, ZORRO consistently certifies significantly higher robustness ratios than MEYER. This is due to the fact that MEYER uses interval-arithmetic which ignores the correlation between model weights in different dimensions and between the training labels and the weights, leading to overly conservative prediction ranges. Specifically, for higher uncertainty radius values, MEYER fails to certify robustness for most of the data while ZORRO still can certify robustness for 100% of the instances.

**Prediction Robustness (Uncertain Features).** We also evaluate the impact of training feature uncertainty (not supported by MEYER). Specifically, we introduce uncertainty into the vehicle weight column for the MPG dataset. As shown in Fig. 1(c), uncertainty in the features results in relatively less robust predictions compared to uncertain labels for a similar uncertainty radius (Fig. 1(a)). This is primarily because uncertain features result in more high-order terms in the closed form solution than uncertain labels which in turn leads to larger over-approximation errors during linearization.

In all experiments the standard deviation of the robustness ratio, calculated by repeating experiments with different random seeds, is large when the average robustness ratio is close to 0.5. This is because when the uncertain data percentage is low, the model will be robust no matter which training instances are selected to be uncertain. Likewise, when the uncertain data percentage is high, then most predictions will be uncertain no matter which training data points are uncertain.

**Parameter Robustness.** Next, we apply ZORRO for robustness certification of parameters in linear regression models, crucial for statistical estimation and causal analysis. We compare the ground truth coefficients for a treatment variable with the results obtained by ZORRO and through KNN imputation on a dataset with injected missing data. Fig. 2 shows the treatment variable coefficient and

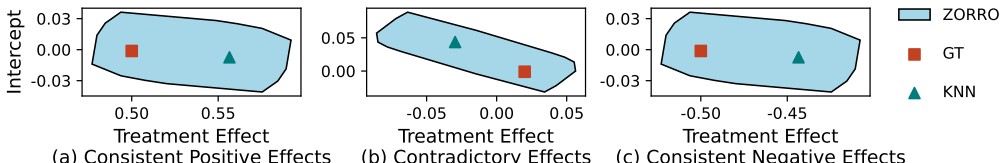

Figure 2: Applying ZORRO to causal inference. The intercept (y-axis) is the model's bias term, the treatment effect (x-axis) is the coefficient for the treatment variable.

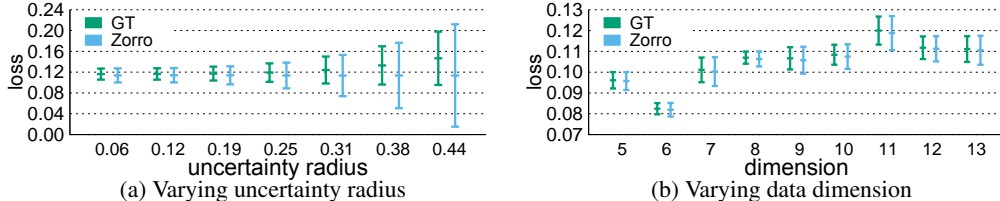

Figure 3: Range of the loss, through enumeration of all possible worlds (GT) and ZORRO.

the regression model's intercept. The intercept captures the baseline level of the outcome variable when all predictors are zero, highlighting how baseline values can shift under uncertainty. While the model trained after KNN imputation sometimes correctly identifies the directionality of the treatment effect, this is not always the case as shown in Fig. 2(b). This highlights the needs for techniques like ZORRO which guarantee that the true treatment effect is within certain bounds.

## 5.2 Solution Quality with Varying Uncertainty and Hyperparameters

We evaluate ZORRO's effectiveness by testing the tightness of the over-approximation and the accuracy of possible models. Specifically, we examine how the over-approximation quality and worst-case loss are influenced by the level of data uncertainty and the regularization coefficient.

**Varying Data Uncertainty.** To evaluate how specific characteristics of the data affect the effectiveness of ZORRO, we injected errors into real datasets, varying uncertain data percentage and uncertainty range. We compare ZORRO with the ground truth range of the loss computed by enumerating all possible worlds (GT). The results shown in Fig. 3(a) demonstrate that ZORRO tightly over-approximates the ground truth loss range, especially for smaller uncertainty radius values. As uncertainty increases, the over-approximation gap widens due to the increased coefficient of higher-order terms in the gradient, which are linearized, leading to higher linearization errors. As shown in Fig. 3(b), the tightness of ZORRO's over-approximation is not affected by the dimension of the data.

**Effect of Regularization.** We investigate the impact of the regularization coefficient on the robustness of predictions and the worst-case loss of possible models. Following a similar approach to Sec. 5.1, we introduce uncertainty in both features and labels in the MPG dataset and use zonotopes with varying levels of uncertainty to over-approximate the training data uncertainty. The results, shown in Fig. 4, indicate that a higher regularization coefficient leads to more robust predictions, as regularization tends to "compress" all possible model weights towards the origin. Interestingly, the worst-case loss shows that $\lambda = 0$ is not optimal across all scenarios, especially when the fraction uncertain instances is high. Instead, a small, positive $\lambda$ (e.g., 0.02 or 0.025) generally yields the best worst-case losses. Combining these results, the optimal regularization coefficient should enhance robustness (i.e., a higher robustness ratio) while maintaining an acceptable worst-case loss. Therefore, the regularization coefficient should be tuned based on a validation dataset to achieve a small range of accurate possible models.

## 6 Conclusions, limitations and broader impacts

We introduce an approach for propagating uncertainty through model training and inference for linear models. Given an abstract uncertain training dataset that over-approximates the possible worlds of a training dataset, we develop abstract interpretation techniques to over-approximate the set of possible models and inference results for this set of models using zonotopes. This is challenging, as we need to compute fixed points of gradient descent in the abstract domain. Our main technical

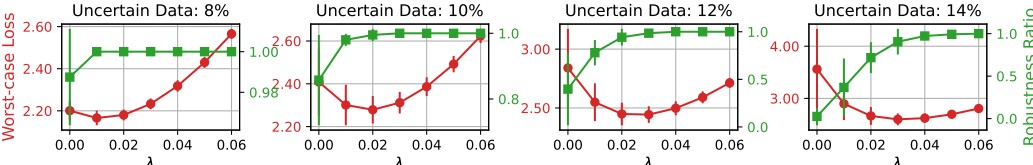

Figure 4: Varying regularization coefficient $\lambda$: Robustness ratio (green) and worst-case test loss (red).

contribution is the development of closed-form solutions for such fixed points that can be solved efficiently. Our techniques efficiently over-approximate models and inference for several use cases, including robustness verification, uncertainty management in causal reasoning, and improving the interpretability and reliability of predictions and inferences. This framework can be particularly valuable in critical applications where data quality and robustness are paramount. While we propose an effective method for abstract learning of linear models, extending our approach to more complex models is challenging. Non-linear models, such as neural networks, would require more advanced linearization and order reduction techniques, as well as parallelization, to manage the increased complexity of the involved symbolic operations. In addition, adapting our method to a broader classes of models through efficiently approximating the fixed points remains a challenging and promising future direction.

## 7  Acknowledgment

This research was supported by NSF awards IIS-2340124, IIS-2420691, and IIS-2420577, as well as by NIH grant U54HG012510. The views, opinions, and findings presented are those of the authors and do not necessarily represent those of the NSF or NIH.

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

## A  Linear Regression and Ridge Regression

In this section, we review standard loss functions for linear regression, specifically mean squared error (MSE) and the loss function used in ridge regression. can be trained via either the closed-form solution or gradient descent. Specifically, the closed-form solution requires computing the inversion of the covariance matrix, while the gradient descent only involves matrix addition and multiplication. Since the linear regression model is convex, gradient descent is guaranteed to converge to the global optimum.

Suppose we have a training data $\boldsymbol{D} = (\boldsymbol{X}, \boldsymbol{y})$ with $n$ i.i.d. samples , where the feature matrix $\boldsymbol{X} = [\boldsymbol{x}_1 \ \cdots \ \boldsymbol{x}_n]^T \in \mathbb{R}^{n \times d}$, and the labels $\boldsymbol{y} = [y_1, \cdots, y_n]^T \in \mathbb{R}^n$. Given an input $\boldsymbol{x}$ and the model weight $\boldsymbol{w}$, the prediction of the linear regression model $\hat{\boldsymbol{y}} = \boldsymbol{w}^T \boldsymbol{x}$ (the bias term can be integrated into $\boldsymbol{w}$, corresponding to an added column in $\boldsymbol{X}$ with constant 1's).

**Loss Functions**   The mean squared error (MSE) loss on $\boldsymbol{D}$ is defined as shown below.

$$L(\boldsymbol{X}, \boldsymbol{y}, \boldsymbol{w}) = \frac{1}{n} \sum_{i=1}^{n} (y_i' - y_i)^2 = \frac{1}{n} \sum_{i=1}^{n} (\boldsymbol{w}^T \boldsymbol{x_i} - y_i)^2 = \frac{1}{n}(\boldsymbol{X}\boldsymbol{w} - \boldsymbol{y})^T (\boldsymbol{X}\boldsymbol{w} - \boldsymbol{y}). \quad (4)$$

In practice, regularization terms, e.g., based on the $l_p$-norm of the model parameters, are often added to the original MSE loss to prevent overfitting by penalizing large weights. Using $l_2$-regularization with a regularization coefficient $\lambda$ which determines the strength of regularization is often called *ridge regression*. The loss function for ridge regression is:

$$L(\boldsymbol{X}, \boldsymbol{y}, \boldsymbol{w}) = \frac{1}{n}(\boldsymbol{X}\boldsymbol{w} - \boldsymbol{y})^T (\boldsymbol{X}\boldsymbol{w} - \boldsymbol{y}) + \lambda \cdot \boldsymbol{w}^T \boldsymbol{w}. \quad (5)$$

**Gradient Descent for Linear Regression**   Due to the convexity of linear models, the locally optimal point, which can be obtained by gradient descent with an appropriate learning rate $\eta$, is globally optimal. In gradient descent, the model weights $\boldsymbol{w}$ are iteratively updated (with some learning rate $\eta$) towards the reverse direction of the gradient $\frac{\partial L(\boldsymbol{X}, \boldsymbol{y}, \boldsymbol{w})}{\partial \boldsymbol{w}}$. Thus, for ridge regression:

$$\frac{\partial L(\boldsymbol{X}, \boldsymbol{y}, \boldsymbol{w})}{\partial \boldsymbol{w}} = \frac{2}{n}(\boldsymbol{X}^T \boldsymbol{X} \boldsymbol{w} - \boldsymbol{X}^T \boldsymbol{y}) + 2\lambda \boldsymbol{w}. \quad (6)$$

Thus, one step of gradient descent $\Phi$ is:

$$\boldsymbol{w}^{i+1} = \boldsymbol{w}^i - \eta \frac{2}{n}(\boldsymbol{X}^T \boldsymbol{X} \boldsymbol{w} - \boldsymbol{X}^T \boldsymbol{y}) + 2\lambda \boldsymbol{w} \quad (7)$$

**Closed-form Solution for Linear Regression**   The convex nature of linear models ensures that the optimal weight $\boldsymbol{w} = \boldsymbol{w}^*$, which minimizes the loss $L(\boldsymbol{X}, \boldsymbol{y}, \boldsymbol{w})$, can be computed by establishing $\frac{\partial L(\boldsymbol{X}, \boldsymbol{y}, \boldsymbol{w})}{\partial \boldsymbol{w}} = 0$:

$$\begin{aligned} \frac{\partial L(\boldsymbol{X}, \boldsymbol{y}, \boldsymbol{w}^*)}{\partial \boldsymbol{w}^*} &= \frac{2}{n}(\boldsymbol{X}^T \boldsymbol{X} \boldsymbol{w}^* - \boldsymbol{X}^T \boldsymbol{y}) + 2\lambda \boldsymbol{w}^* = 0 \\ \Rightarrow \quad \boldsymbol{X}^T \boldsymbol{y} &= (\boldsymbol{X}^T \boldsymbol{X} + \lambda n I)\boldsymbol{w}^* \\ \Rightarrow \quad \boldsymbol{w}^* &= (\boldsymbol{X}^T \boldsymbol{X} + \lambda n I)^{-1} \boldsymbol{X}^T \boldsymbol{y} \end{aligned} \quad (8)$$

## B  Background on Abstract Interpretation

Abstract interpretation [13] is a technique for over-approximating the results of computations over a set of inputs. This is achieved by associating sets of elements of a concrete domain $\mathbb{D}$ with elements from an abstract domain $\mathbb{D}^\sharp$. In this context, various abstract domains are employed: interval domains represent variables as ranges of possible values [13], octagon domains allow for constraints between

pairs of variables within a specific bound [48], and polynomial zonotopes which represent convex polytopes using polynomial constraints [14]. Abstract interpretation, while originally designed for static program analysis such as strictness analysis, has also found applications in wide range of other domains including reachability analysis [1, 5, 2, 10, 36, 4], robustness verification for neural networks [32, 64], learning robust models by providing bounds on the loss for a set of inputs [51, 63, 49, 22], and many others.

We now state some important, but well-known, facts about abstract transformers that we utilize in our derivations.

## B.1 Abstract Transformers

In Def. 2.5, we presented the standard definition of abstract transformers as functions over abstract domains that over-approximate the application of functions in the concrete domains to sets of elements. To clarify the connection between abstract transformers and possible world semantics observe that both take a concrete function $F : \mathbb{D} \to \mathbb{D}'$ and lift it to sets of inputs $S \in \mathcal{P}(\mathbb{D})$ through point-wise applications:

$$F(S) = \{F(e) \mid e \in S\}$$

Thus, abstract interpretation is a natural fit for over-approximating PWS by over-approximating sets of possible worlds using abstract elements and then over-approximates computations with PWS for a function $F$ using abstract transformers for such a function.

## B.2 Abstract Transformers Compose

Importantly, abstract transformers compose. This enables us to decompose a complex computation into simpler operations and build an abstract transformer for the computation by composing abstract transformations for these simpler operations.

**Proposition B.1** (Abstract Transformers Compose). *Consider (exact) abstract transformers $f^\sharp$ and $g^\sharp$ for functions $f$ and $g$, then $g^\sharp \circ f^\sharp$ is an (exact) abstract transformer for $g \circ f$.*

# C Symbolic and Standard Representation of Zonotopes

We now provide a more detailed account of the correspondence between the symbolic and standard geometric representation of zonotopes and polynomial zonotopes and discuss why matrices and sets of symbolic matrices as used in our abstract domain can equivalently be thought of as (polynomial) zonotopes.

## C.1 Symbolic vs. Geometric Representation

We defined zonotopes $z^\sharp$ as vectors $\Psi^d$ where $\Psi$ denotes polynomials over variables (called error symbols) $\mathcal{E}$. The concretization of such a symbolic representation of a zonotope is the set of vectors in $\mathbb{R}^d$ that can be derived from $z^\sharp$ by assigning values from $[-1, 1]$ to each variable $\epsilon \in \mathcal{E}$. We encode such variable assignments as vectors $[-1, 1]^{|\mathcal{E}|}$.

**Definition C.1** (Polynomial zonotopes - Symbolic Representation). *A $d$-dimensional polynomial zonotope is a vector $z^\sharp \in \Psi^d$ [10]. Let $m = |\mathcal{E}|$. The concretization of $z^\sharp$ is defined as:*

$$\gamma\left(z^\sharp\right) = \left\{ z^\sharp(e) \mid e \in [-1, 1]^{|\mathcal{E}|} \right\}$$

A common measure of the representation size of a zonotope is its order. The order of a $d$-dimensional (polynomial) $z^\sharp$ in symbolic representation is the total number of distinct monomials in $z^\sharp$ divided by $d$:

$$\text{ORD}(z^\sharp) = \frac{\#\mathcal{M}(z^\sharp)}{d}$$

where $\#\mathcal{M}(\psi)$ denotes the cardinality of the set of distinct monomials in polynomial $\psi$.

A more common way to represent $d$-dimensional zonotopes is by fixing a set $\mathcal{S}$ of monomials over $\mathcal{E}$ and writing the zonotope as a central point $c \in \mathbb{R}^d$ and a sum of generator vectors $g_i \in \mathbb{R}^d$ multiplied with the monomials from $\mathcal{S}$. The generator vector $g_i$ assigns a coefficient to monomial $\mathcal{S}[i]$ for each of the $d$ dimensions.

**Definition C.2** (Polynomial zonotopes - Geometric Representation). *The geometric representation of a $d$-dimensional polynomial zonotope is a sum of a center point $c \in \mathbb{R}^d$ and the monomials over error symbols in $\mathcal{E}$ from a set $\mathcal{S}$ multiplied by coefficients encoded in set of generator vectors $g_i \in \mathbb{R}^d$:*

$$z^\sharp = c + \sum_{i=1}^{|\mathcal{S}|} g_i \mathcal{S}[i]$$

For the geometric representation, the order of a $d$-dimensional (polynomial) $z^\sharp$ over monomials $\mathcal{S}$ is defined as:

$$\mathrm{ORD}(z^\sharp) = \frac{|\mathcal{S}|}{d}$$

Note that the order of the symbolic and geometric representation of a zonotope are the same. As an example consider the 4-dimensional polynomial zonotope $z^\sharp$ shown in symbolic representation (left) and standard representation (right).

$$\begin{bmatrix} 1 + 1\epsilon_2{}^3 + 1\epsilon_1{}^2\epsilon_2\epsilon_3 \\ 4 + 2\epsilon_2{}^3 + 2\epsilon_1{}^2\epsilon_2\epsilon_3 \\ 0 + 2\epsilon_2{}^3 + 3\epsilon_1{}^2\epsilon_2\epsilon_3 \\ 2 + 1\epsilon_2{}^3 + 4\epsilon_1{}^2\epsilon_2\epsilon_3 \end{bmatrix} \qquad \underbrace{\begin{bmatrix} 1 \\ 4 \\ 0 \\ 2 \end{bmatrix}}_{c} + \underbrace{\begin{bmatrix} 1 \\ 2 \\ 2 \\ 1 \end{bmatrix}}_{g_1} \cdot \underbrace{\epsilon_2^3}_{\mathcal{S}[1]} + \underbrace{\begin{bmatrix} 1 \\ 2 \\ 3 \\ 4 \end{bmatrix}}_{g_2} \cdot \underbrace{\epsilon_1{}^2\epsilon_2\epsilon_3}_{\mathcal{S}[2]}$$

## C.2 Symbolic Matrices and Sets of Symbolic Matrices

We use matrices over symbolic expressions and sets of heterogeneous abstract matrices to represent the state of a computation and allow matrices from such a set to share variables to encode relationships between the elements of such matrices. The semantics we associate with such a set is that of a *joint concretization*. For $S^\sharp = \{M_i{}^\sharp\}$, we define:

$$\gamma\left(S^\sharp\right) = \{M_i{}^\sharp(e) \mid M_i{}^\sharp \in S^\sharp \wedge e \in [-1, 1]^n\}$$

zonotopes encode convex sets of points in $\mathbb{R}^d$. We can think of a symbolic matrix $M^\sharp \in \Psi^{n \times m}$ as a $n \cdot m$-dimensional polynomial zonotope. Similarly, a set of symbolic matrices $S^\sharp = \{M_i{}^\sharp \in \mathbb{R}^{n_i \times m_i}\}$ can be thought of as a $l$-dimensional polynomial zonotope $\mathbb{R}^l$ where $l = \sum_i n_i \cdot m_i$. If every symbolic expression in a matrix or set of matrices is a linear combination of error symbols (an affine form), then such objects can equivalently be represented as zonotopes.

For instance, below on the left we show a zonotope matrix $W^\sharp$ (all expressions are linear) and two possible worlds in its concretization (for assignments $[-1, -1]$ and $[0, 0.5]$).

$$W^\sharp = \begin{bmatrix} \epsilon_1 + 3 & \epsilon_2 \\ 15 & 22\epsilon_1 \end{bmatrix} \qquad W^\sharp\left(\begin{bmatrix} -1 \\ -1 \end{bmatrix}\right) = \begin{bmatrix} 2 & -1 \\ 15 & -22 \end{bmatrix} \qquad W^\sharp\left(\begin{bmatrix} 0 \\ 0.5 \end{bmatrix}\right) = \begin{bmatrix} 3 & 0.5 \\ 15 & 11 \end{bmatrix}$$

## C.3 Abstract Training Data and Model Weights

Using possible world semantics to compute all possible model weights $w^{\odot*}$ given a set of possible training datasets $D^\odot$, there will be a natural correspondence between a model weight $w \in w^{\odot*}$ and the dataset $D \in D^\odot$ from which is was derived. When training in the abstract domain such correlations can be preserved by sharing error symbols between the abstract training data $\mathbf{D}^\sharp$ and corresponding model weights $w^\sharp$. When such sharing occurs, then it is critical to reason about the joint concretization when determining whether an abstract fixed point as been achieved. Testing equivalence (equal concretization) of $w^\sharp$ alone can lead to false positives, as $w^\sharp{}_1 \simeq_\sharp w^\sharp{}_2$ does not in general imply $(\mathbf{D}^\sharp, w^\sharp{}_1) \simeq_\sharp (\mathbf{D}^\sharp, w^\sharp{}_2)$. This is due to the fact that a particular concrete model weight $w \in \gamma\left(w^\sharp{}_1\right) = \gamma\left(w^\sharp{}_2\right)$ may be associated with different datasets in $(\mathbf{D}^\sharp, w^\sharp{}_1)$ and $\mathbf{D}^\sharp, w^\sharp{}_2)$ because of shared error symbols between $\mathbf{D}^\sharp$ and the abstract model weights. Thus,

concretization equivalence between model weights only does not imply equivalent results after application of a gradient decent step.

To further illustrate this consider, two pairs of abstract model weights $\boldsymbol{w}^{\sharp}{}_1$ and $\boldsymbol{w}^{\sharp}{}_2$.

$$\left( \boldsymbol{w}^{\sharp}{}_1 = \begin{bmatrix} 2 + 2\epsilon_1 \\ 3 + \epsilon_2 \end{bmatrix}, \mathbf{D}^{\sharp} = \begin{bmatrix} \epsilon_1 \\ 5 \end{bmatrix} \right)$$

$$\left( \boldsymbol{w}^{\sharp}{}_2 = \begin{bmatrix} 2 + 2\epsilon_3 \\ 3 + \epsilon_2 \end{bmatrix}, \mathbf{D}^{\sharp} = \begin{bmatrix} \epsilon_1 \\ 5 \end{bmatrix} \right)$$

The only difference between $\boldsymbol{w}^{\sharp}{}_1$ and $\boldsymbol{w}^{\sharp}{}_2$ is that $\boldsymbol{w}^{\sharp}{}_1$ shares a variable ($\epsilon_1$) with $\mathbf{D}^{\sharp}$ while $\boldsymbol{w}^{\sharp}{}_2$ does not. Observe that $\gamma\left(\boldsymbol{w}^{\sharp}{}_1\right) = \gamma\left(\boldsymbol{w}^{\sharp}{}_2\right)$. However, applying a step of gradient decent to $\boldsymbol{w}^{\sharp}{}_1$ and $\boldsymbol{w}^{\sharp}{}_2$ may lead to abstract models that are not equivalent. For sake of this example, consider a hypothetical gradient operator $\Phi^{\sharp}_{dummy}$ that subtracts the data from the current model weight.

$$\left( \Phi^{\sharp}_{dummy}(\boldsymbol{w}^{\sharp}{}_1) = \begin{bmatrix} 2 + \epsilon_1 \\ -2 + \epsilon_2 \end{bmatrix}, \mathbf{D}^{\sharp} = \begin{bmatrix} \epsilon_1 \\ 5 \end{bmatrix} \right)$$

$$\left( \Phi^{\sharp}_{dummy}(\boldsymbol{w}^{\sharp}{}_2) = \begin{bmatrix} 2 + 2\epsilon_3 - \epsilon_1 \\ -2 + \epsilon_2 \end{bmatrix}, \mathbf{D}^{\sharp} = \begin{bmatrix} \epsilon_1 \\ 5 \end{bmatrix} \right)$$

Note that $\Phi^{\sharp}_{dummy}(\boldsymbol{w}^{\sharp}{}_1) \not\simeq_{\sharp} \Phi^{\sharp}_{dummy}(\boldsymbol{w}^{\sharp}{}_2)$. Thus, even if two abstract model weights have equal concretization this does not guarantee that a fixed point has been reached when also taking the data into account. This is important as otherwise it is not possible that the abstract model weights contain all possible optimal model weights $\boldsymbol{w}^{\odot *}$. Instead, we need to consider the *joint concretization* of model weight and data, s.t. if $\gamma\left(S^{\sharp}{}_1\right) = \gamma\left(S^{\sharp}{}_2\right)$ then concretization equivalence is guaranteed to hold for all subsequent iterations.

## D  Examples Of Training Data Uncertainty and Abstraction

In this section, we discuss several causes of training (and test) data uncertainty, how to encode them as possible worlds, and abstraction functions for approximating such uncertainty in our symbolic model.

### D.1  Measurement Uncertainty

Sensors typically have some measurement uncertainty. If the measurement error $\epsilon$ is know, e.g., provided by the instrument manufacturer or estimated through repeated measurements and calibration, then for a set of sensor readings used in training $\boldsymbol{D} = \{(\boldsymbol{x}_i, y_i)\}$, then each possible world in $\boldsymbol{D}^{\odot}$ is derived from $\boldsymbol{D}$ by replacing values $\boldsymbol{x}_i[j]$ with values in $\boldsymbol{x}_i[j] \pm \epsilon$.

### D.2  Missing Values and Imputation

Consider a training dataset $\boldsymbol{D}$ where some of the features are missing for some of the datapoints in $\boldsymbol{D}$. If each feature $\mathcal{X}_i$'s domain is an interval $[l_i, u_i]$ and assuming that missing values can be represented as independent random variables, then the set of possible worlds of the training data $\boldsymbol{D}^{\odot}$ are all training datasets that can be derived from $\boldsymbol{D}$ by replacing each missing value in a feature $\mathcal{X}_i$ with a value from $[l_i, u_i]$.

We can use an abstraction function $\alpha_{missing}$ that represents each missing value as an interval $[l_i, u_i]$. In the symbolic representation, this is encoded as the central point of the interval of an error symbol $\epsilon$ with a coefficient half of the interval's length. We associate a separate error symbol with each missing value. For $\boldsymbol{D} \in \mathbb{R}^{n \times m}$, we define $\alpha_{missing}$ for $i \in [1, n]$ and $j \in [1, m]$ as:

$$\alpha_{missing}(\boldsymbol{D}_{ij}) = \begin{cases} \frac{u_j + l_j}{2} + \frac{u_j - l_j}{2}\epsilon_{i,j} & \text{if } \boldsymbol{D}_{ij} = \perp \\ \boldsymbol{D}_{ij} & \text{otherwise} \end{cases}$$

For instance, consider the training dataset $\boldsymbol{D} \in \mathbb{R}^{3 \times 2}$ shown below where $\perp$ denotes a missing value.

$$\boldsymbol{D} = \begin{pmatrix} 3 & \perp \\ 1 & 6 \\ \perp & 9 \end{pmatrix}$$

Assuming that both features have a domain $[0, 10]$, we get:

$$\alpha_{missing}(\boldsymbol{D}) = \begin{pmatrix} 3 & 5 + 5\epsilon_{1,2} \\ 1 & 6 \\ 5 + 5\epsilon_{3,1} & 9 \end{pmatrix}$$

If the independence assumption on missing values holds, then $\gamma\left(\alpha_{missing}(\boldsymbol{D})\right) = \boldsymbol{D}^{\odot}$. If the independence assumption does not hold, then $\boldsymbol{D}^{\odot}$ would be a subset of the worlds described above and $\alpha_{missing}(\boldsymbol{D})$ is a still a valid abstraction function, albeit an over-approximating one:

$$\gamma\left(\alpha_{missing}(\boldsymbol{D})\right) \supseteq \boldsymbol{D}^{\odot}$$

**Imputation** If we make the stronger assumption the unknown ground truth value corresponding to a missing value is in a set of estimations $\{a_1, \ldots, a_m\}$ returned by a set of imputation methods, then we can use $[min(\{a_i\}), max(\{a_i\})]$ instead of $[l_i, u_i]$.

**Training Label Uncertainty** The abstract transformer $\alpha_{missing}$ can also be used if training data labels are missing.

## E Abstract Transformers for Zonotopes

In this section we introduce (exact) abstract transformers for (polynomial) zonotopes that have been introduced in related work and discuss their computation complexity and the space requirements for the output zonotope $z^{\sharp}$ in terms of its order $\text{ORD}(z^{\sharp})$ (see Sec. 2.2). Kochdumper et al. [37, Table 1] shows an overview of which operations are exact for linear and polynomial zonotopes (and other set representations). Relevant to for our purpose is that exact transformers exist for polynomial zonotopes for all operations used in the learning algorithms for linear models we consider.

**Proposition E.1** (Exact Transformers for Polynomial Zonotopes). *There exist exact abstract transformers for scalar addition and multiplication as well as for matrix addition and multiplication for polynomial zonotopes [3]. There exist exact transformers for scalar addition and matrix addition for zonotopes. Abstract transformers for multiplication and matrix multiplication for zonotopes exist, but are not exact.*

We present the details of these operations in the following.

### E.1 Arithmetic Operations

**Addition.** Scalar and matrix addition are exact in both zonotopes and polynomial zonotopes. Given two matrix (polynomial) zonotopes $\boldsymbol{V}^{\sharp}$ and $\boldsymbol{W}^{\sharp}$ in $\Psi^{n \times m}$, their addition $\boldsymbol{Z}^{\sharp} = \boldsymbol{V}^{\sharp} + \boldsymbol{W}^{\sharp}$ is defined by adding entries. For each $i \in [1, n]$ and $j \in [1, m]$:

$$\boldsymbol{Z}^{\sharp}{}_{ij} = \boldsymbol{V}^{\sharp}{}_{ij} + \boldsymbol{W}^{\sharp}{}_{ij}$$

The order of $\boldsymbol{Z}^{\sharp}$ is the sum of the orders of $\boldsymbol{V}^{\sharp}$ and $\boldsymbol{W}^{\sharp}$:

$$\text{ORD}(\boldsymbol{Z}^{\sharp}) = \text{ORD}(\boldsymbol{V}^{\sharp}) + \text{ORD}(\boldsymbol{W}^{\sharp})$$

**Scalar multiplication.** Multiplying a (polynomial) zonotope matrix $\boldsymbol{W}^{\sharp}$ with a scalar $c$ is exact using the abstract transformer $\cdot_{poly}$ defined below. We simply multiply each entry in the matrix by $c$. For each $i \in [1, n]$ and $j \in [1, m]$:

$$(c \cdot_{poly} \boldsymbol{W}^{\sharp})_{ij} = c \cdot \boldsymbol{W}^{\sharp}{}_{ij}$$

Multiplication with a scalar abstract value $d \in \Psi$ is exact for polynomial zonotope matrices, but increases the order of the input zonotope:

$$\text{ORD}(d \cdot \boldsymbol{W}) = \text{ORD}(d) \cdot \text{ORD}(\boldsymbol{W})$$

Here we define the order of a scalar abstract value $d \in \Psi$ to be the number of monomials in the representation of $d$. For a linear zonotope $\boldsymbol{W}$, $d \cdot \boldsymbol{W}^\sharp{}_{ij}$ is in general not linear as it contains higher-order terms. Thus, matrix multiplication for linear zonotopes requires application of linearization (see App. G):

$$d \cdot_{Lin} \boldsymbol{W} = \mathbf{L}(d \cdot_{poly} \boldsymbol{W})$$

**Matrix multiplication.**  Matrix multiplication is defined using scalar multiplication and addition. As discussed above, addition is exact for both linear and polynomial zonotopes. However, scalar multiplication of symbolic expressions is only exact for polynomial zonotopes, but requires linearization and, thus, over-approximation, for linear zonotopes. That is, matrix multiplication is exact for polynomial zonotopes only. Consider two matrices $\boldsymbol{V}^\sharp \in \Psi^{n \times m}$ and $\boldsymbol{V}^\sharp \in \Psi^{m \times k}$, then matrix multiplication is defined as usual, but using abstract transformers for scalar addition and multiplication. Each symbolic entry in the matrix $\boldsymbol{V}^\sharp \cdot \boldsymbol{W}^\sharp$ is a sum of $m$ elements, each the multiplication of one entry of $\boldsymbol{V}^\sharp$ with one entry of $\boldsymbol{W}^\sharp$. Thus,

$$\text{ORD}(\boldsymbol{V}^\sharp \cdot \boldsymbol{W}^\sharp) = m \cdot \text{ORD}(\boldsymbol{V}^\sharp) \cdot \text{ORD}(\boldsymbol{W}^\sharp)$$

For linear zonotopes, we have to again apply linearization to make sure that the output is a linear zonotope.

# F  Abstract Transformers for Gradient Descent

## F.1  Abstract Fixed Points Over-Approximate Possible Fixed Points

*Proof of Prop. 3.3.* Initially, we will assume that $\boldsymbol{w}^{\sharp j}$ is computed through repeated application of $F^\sharp$. Let $n$ be the smallest number such that $\boldsymbol{w}^{\sharp *} = \boldsymbol{w}^{\sharp n} = F^\sharp(\boldsymbol{w}^{\sharp n-1}, \mathbf{D}^\sharp)$ for iteration with abstract transformer $F^\sharp$. As $F^\sharp$ is an abstract transformer for $\Phi$ and abstract transformers compose (Prop. B.1), we know that for every $\boldsymbol{D}_i \in \boldsymbol{D}^\odot$ and $n \in \mathbb{N}$, we have $(\boldsymbol{w}_i^n, \boldsymbol{D}_i) \in \gamma\left(\boldsymbol{w}^{\sharp n}, \mathbf{D}^\sharp\right)$. To prove the claim it is sufficient to show that for every such $\boldsymbol{D}_i$ we have $(\boldsymbol{w}_i^*, \boldsymbol{D}_i) \in \gamma\left(\boldsymbol{w}^{\sharp n}, \mathbf{D}^\sharp\right)$. WLOG consider some $\boldsymbol{D}_i \in \boldsymbol{D}^\odot$ and $(\boldsymbol{w}_i^n, \boldsymbol{D}_i) \in \gamma\left(\boldsymbol{w}^{\sharp n}, \mathbf{D}^\sharp\right)$. Let $m \in \mathbb{N}$ be the smallest number such that $\boldsymbol{w}_i^* = \boldsymbol{w}_j^m$. If $m \leq n$, then based on the fact that $F^\sharp$ is an abstract transformer (over-approximates $\Phi$), the result holds as $(\boldsymbol{w}_i^n, \boldsymbol{D}_i) \in \gamma\left(\boldsymbol{w}^{\sharp n}, \mathbf{D}^\sharp\right)$. Now consider the case where $m > n$. We will show through induction that for all $j \in [n+1, m]$, $(\boldsymbol{w}_i^j, \boldsymbol{D}_i) \in \gamma\left(\boldsymbol{w}^{\sharp n}, \mathbf{D}^\sharp\right)$ and, thus $(\boldsymbol{w}_i^*, \boldsymbol{D}_i) \in \gamma\left(\boldsymbol{w}^{\sharp n}, \mathbf{D}^\sharp\right)$.

*Induction start*. For $j = n$ the result trivially holds based on the definition of abstract fixed points.

*Induction step*. Assume that $(\boldsymbol{w}_i^j, \boldsymbol{D}_i) \in \gamma\left(\boldsymbol{w}^{\sharp n}, \mathbf{D}^\sharp\right)$ for $j \in [n, m-1]$, we have to show that this implies that $(\boldsymbol{w}_i^{j+1}, \boldsymbol{D}_i) \in \gamma\left(\boldsymbol{w}^{\sharp n}, \mathbf{D}^\sharp\right)$. By definition, we have

$$\boldsymbol{w}_i^{j+1} = \Phi(\boldsymbol{w}_i^j)$$

As $F^\sharp$ is an abstract transformer for $\Phi$, we have, $(\boldsymbol{w}_i^{j+1}, \boldsymbol{D}_i) \in \gamma\left(\boldsymbol{w}^{\sharp n+1}, \mathbf{D}^\sharp\right)$. Now based on the fact that $\boldsymbol{w}^{\sharp n}$ is an abstract fixed point according to Def. 3.2 and, thus, $\gamma\left(\boldsymbol{w}^{\sharp n}, \mathbf{D}^\sharp\right) \supseteq \gamma\left(\boldsymbol{w}^{\sharp n+1}, \mathbf{D}^\sharp\right)$, it follows that $(\boldsymbol{w}_i^{j+1}, \boldsymbol{D}_i) \in \gamma\left(\boldsymbol{w}^{\sharp n}, \mathbf{D}^\sharp\right)$.

So far we have demonstrated that a fixed point $\boldsymbol{w}^{\sharp *}$ that appears in the iteration sequence $\{\boldsymbol{w}^{\sharp j}\}_{j=0}^\infty$ contains all optimal model weights $\boldsymbol{w}^{\odot *}$. We now prove the stronger result that as long as $\boldsymbol{w}^{\sharp *}$ fulfills the condition of Def. 3.2, no matter it is the result of an iteration sequence using $F^\sharp$ or not, its concretization encloses $\boldsymbol{w}^{\odot *}$. Consider one $\boldsymbol{D}_i \in \gamma\left(\mathbf{D}^\sharp\right)$ and as above let $\boldsymbol{w}_i^*$ denote its optimal model weight. We will demonstrate that $(\boldsymbol{w}_i^*, \boldsymbol{D}_i) \in \gamma\left(\boldsymbol{w}^{\sharp *}, \mathbf{D}^\sharp\right)$. First note that given that gradient descent for linear models is convex, for any initial model weight $\boldsymbol{w}_i^0$, the sequence $\{\boldsymbol{w}_i^j\}_{j=0}^\infty$ converges to $\boldsymbol{w}_i^*$. Specifically, let $\boldsymbol{w}$ denote the model weight such that $(\boldsymbol{w}, \boldsymbol{D}_i) \in \gamma\left(\boldsymbol{w}^{\sharp *}, \mathbf{D}^\sharp\right)$

and let $n_i$ denote the smallest integer such that $\boldsymbol{w}_i^{n_i} = \boldsymbol{w}_i^*$ for the sequence generated starting from $\boldsymbol{w}_i^0 = \boldsymbol{w}$. However, now we can apply the same proof by induction shown above to demonstrate that $(\boldsymbol{w}_i^{n_i}, \boldsymbol{D}_i) \in \gamma\left(\boldsymbol{w}^{\sharp\,*}, \mathbf{D}^\sharp\right)$. This concludes the proof. $\qquad\square$

## F.2 Exact Abstract Transformer for Gradient Descent

In this section we show that the abstract transformer for gradient descent introduced in sec. 3 has a fixed point.

**Proposition F.1.** *The abstract gradient descent operator $\Phi^\sharp$ has a fixed point $\boldsymbol{w}^{\sharp\,*}$.*

*Proof of Prop. F.1.* The existence of a fixed point is implied by the fact that $\Phi^\sharp$ is an exact abstract transformer of the concrete gradient descent operator $\Phi$. Consider $\gamma\left(\boldsymbol{w}^{\sharp\,0}, \mathbf{D}^\sharp\right) = \{(\boldsymbol{w}_i^0, \boldsymbol{D}_i)\}$. Let $n_i$ be the smallest integer such that $\boldsymbol{w}_i^{n_i} = \boldsymbol{w}_i^*$ and let $n = \max_i n_i$, i.e., at iteration $n$, the concrete model weights have converged in every possible world in the concretization of $(\boldsymbol{w}^{\sharp\,0}, \mathbf{D}^\sharp)$. As $\Phi^\sharp$ is an exact abstract transformer, we can show by induction that $\gamma\left(\boldsymbol{w}^{\sharp\,j}, \mathbf{D}^\sharp\right) = \{(\boldsymbol{w}_i^j, \boldsymbol{D}_i)\}$ for any $j$. As all computations in the concretization have converged at $n$, we know that $\boldsymbol{w}_i^{n+1} = \Phi(\boldsymbol{w}_i^n) = \boldsymbol{w}_i^n$ and, thus, $\{(\boldsymbol{w}_i^n, \boldsymbol{D}_i)\} = \{(\boldsymbol{w}_i^{n+1}, \boldsymbol{D}_i)\}$. As $\Phi^\sharp$ is exact, we get the desired result: $\gamma\left(\boldsymbol{w}^{\sharp\,n}, \mathbf{D}^\sharp\right) = \{(\boldsymbol{w}_i^n, \boldsymbol{D}_i)\} = \{(\boldsymbol{w}_i^{n+1}, \boldsymbol{D}_i)\} = \gamma\left(\boldsymbol{w}^{\sharp\,n+1}, \mathbf{D}^\sharp\right)$. $\qquad\square$

## F.3 Prediction with Abstract Model Weight Fixed Points

We now discuss how to use the abstract model weights $\boldsymbol{w}^{\sharp\,*}$ returned by our abstract transformer for learning linear models during inference to over-approximate the possible set of predictions for a test data point. We start by discussing test data that is not uncertain and then extend the discussion to the case where the test data is also uncertain.

**Deterministic Test Data**   For now let us assume that the test data is not uncertain. The following corollary then enables us to use abstract gradient descent for inference and for over-approximating the prediction ranges.

**Corollary F.2.** *Let $f_{\boldsymbol{w}}(\boldsymbol{x})$ denote the linear model for parameters $\boldsymbol{w}$. Given an incomplete training dataset $\boldsymbol{D}$ associated with a set of possible worlds $\boldsymbol{D}^\odot$, the prediction range $V(\boldsymbol{x})$ for a test data point $\boldsymbol{x}$ can be over-approximated by:*

$$V^\sharp(\boldsymbol{x}) = \left[\min_{\boldsymbol{w} \in \gamma(\boldsymbol{w}^{\sharp\,*})} f_{\boldsymbol{w}}(\boldsymbol{x}), \max_{\boldsymbol{w} \in \gamma(\boldsymbol{w}^{\sharp\,*})} f_{\boldsymbol{w}}(\boldsymbol{x})\right],$$

*where $\boldsymbol{w}^{\sharp\,*}$ is the fixed point of the abstract gradient descent operator $\Phi^\sharp$ applied to $\mathbf{D}^\sharp = \alpha(\boldsymbol{D}^\odot)$, the abstract representation of $\boldsymbol{D}^\odot$ in the zonotope domain.*

The prediction returned by a linear model with parameters $\boldsymbol{w}$ for a data point $\boldsymbol{x}$ is $\boldsymbol{w}^T\boldsymbol{x}$. As $\boldsymbol{x}$ does not contain any symbolic term this is the sum of linear terms multiplied by constants, i.e., the result is a 1-dimensional linear zonotope which is a linear expression of the form:

$$c_0 + \sum_{i=1}^m c_i \epsilon_i$$

where $c_i \in R$ and $\epsilon_i \in \mathcal{E}$. The minimum and maximum value of a 1-d linear zonotope can be determined efficiently as shown below:

$$\left[c_0 - \sum_{i=1}^m |c_i|, c_0 + \sum_{i=1}^m |c_i|\right]$$

**Uncertain Test Data**   In Sec. 2 we modeled uncertainty in the training data as sets of possible worlds $\boldsymbol{D}^\odot$. As mentioned in that section, our techniques also supports uncertain test data, i.e., both the training and test data may be uncertain.

In the most general case, the training and test data may be correlated.[1] We model this as a set of possible worlds $(\boldsymbol{D}^{\odot}, \mathbf{X}_{\text{test}}{}^{\odot})$ where each world is a pair of a training dataset $\boldsymbol{D}_i$ and a test dataset $\mathbf{X}_{\text{test}i}$:

$$(\boldsymbol{D}, \mathbf{X}_{\text{test}})^{\odot} = \{(\boldsymbol{D}_1, \mathbf{X}_{\text{test}1}), \ldots, (\boldsymbol{D}_m, \mathbf{X}_{\text{test}m})\}$$

If training and test data are independent, then we can specify their worlds separately ($\boldsymbol{D}^{\odot}$ and $\mathbf{X}_{\text{test}}{}^{\odot}$) and assume the worlds of $(\boldsymbol{D}, \mathbf{X}_{\text{test}})^{\odot}$ to be their cross product. Uncertainty propagation for inference then requires us to compute the set of possible predictions:

$$\boldsymbol{y}^{\odot} = \{f(\mathbf{X}_{\text{test}i}) \mid \exists (\boldsymbol{D}_i, \mathbf{X}_{\text{test}i}) \in (\boldsymbol{D}, \mathbf{X}_{\text{test}})^{\odot} : f_i = \mathcal{A}(\boldsymbol{D}_i)\}$$

For inference in the abstract domain we first have to select an appropriate abstraction function for the test data, e.g., using the same abstraction function $\alpha$ we use for the training data. The only difference to the case discussed above is that a test data point is now also an abstract element $\boldsymbol{x}^{\sharp}$ and the prediction $\boldsymbol{w}^{\sharp}\boldsymbol{x}^{\sharp}$ is a polynomial zonotope as it contains higher order terms that are the result of multiplying linear terms. To efficiently determine the minimum and maximum we can, e.g., employ linearization to map the polynomial zonotope into a linear one and apply the solution described above for finding the minimum and maximum of a linear zonotope.

# G  Linearization and Order Reduction Techniques

## G.1  Linearization

The purpose of linearization is to over-approximate an input polynomial zonotope with a linear zonotope.

**Definition G.1** (Linearization Operator $\mathbf{L}$). *A linearization operator $\mathbf{L}$ maps a polynomial zonotope $\boldsymbol{z}^{\sharp}$ to a linear zonotope $\boldsymbol{\ell}^{\sharp}$. It replaces high-order polynomial terms with new error symbols, ensuring that all expressions are linear while maintaining an over-approximation:*

$$\gamma \left( \mathbf{L}(\boldsymbol{z}^{\sharp}) \right) \supseteq \gamma \left( \boldsymbol{z}^{\sharp} \right).$$

During inference, computing the prediction intervals using the linear zonotope representation can be done efficiently with linear programming. However, more importantly, in our construction of abstract fixed points we use a specific order reduction technique that requires prior linearization in each gradient descent step to enforce the existence of a fixed point. For a $d$-dimensional polynomial zonotope $\boldsymbol{z}^{\sharp}$ with a set of monomials $\mathcal{S}$:

$$\boldsymbol{z}^{\sharp} = \boldsymbol{c} + \sum_{i=1}^{|\mathcal{S}|} \boldsymbol{g}_i \mathcal{S}[i],$$

we have its linearization:

$$\mathbf{L}(\boldsymbol{z}^{\sharp}) = \boldsymbol{c} + \underbrace{\sum_{i \in \sigma_l} \boldsymbol{g}_i \mathcal{S}[i]}_{\textbf{Linear monomials}} + \underbrace{\sum_{i \notin \sigma_l} \boldsymbol{g}_i \epsilon_i'}_{\textbf{Replaced with linear monomials}}$$

where $\sigma_l$ denotes the set of indices of all linear terms in $\boldsymbol{z}^{\sharp}$. Here, the third part over-approximates each high-order term in $\boldsymbol{z}^{\sharp}$ by replacing it with a new error symbol.

## G.2  Order Reduction Operators

*Order reduction operators* are used to reduce the representation size of a zonotope. That is, an order reduction operator takes as input a linear zonotope $\boldsymbol{\ell}^{\sharp}$ and return a linear zonotope of smaller order

---

[1]Note that this does not necessarily imply a violation of the i.i.d. assumption. For instance, consider a dataset with a textual feature race that is first translated into a categorical feature that is then one-hot encoded into multiple binary attributes. If we are uncertain about the meaning of a particular value of the original attribute, then this leads to a correlation between the uncertainty of both datasets as the interpretation of this value affects all data points with that particular value in the race feature before preprocessing.

(smaller representation size) that over-approximates $\ell^\sharp$. For linear zonotopes this means that order reduction operators reduce the number of distinct error symbols that occur in a zonotope by merging error symbols.

**Definition G.2** (Order Reduction Operator $\mathbf{R}$). *An order reduction operator $\mathbf{R}$ takes a linear zonotope $\ell^\sharp$ as input and returns another linear zonotope of reduced order:*

$$\gamma\left(\mathbf{R}(\ell^\sharp)\right) \supseteq \gamma\left(\ell^\sharp\right) \qquad\qquad \text{ORD}(\mathbf{R}(\ell^\sharp)) < \text{ORD}(\ell^\sharp)$$

We now present details about two commonly adopted order reduction techniques: *Interval Hull* (*IH*) and *transformation-based Interval Hull* (*TIH*) in App. G. In this section, we use the geometric representation of zonotopes (see Def. C.2). Note that for a linear zonotope, each $\mathcal{S}[i]$ consists of a single error term $\epsilon_i$. Thus, we can write such a zonotope $\ell^\sharp$ as:

$$\ell^\sharp = \boldsymbol{c} + \sum_{i=1}^{|\mathcal{S}|} \boldsymbol{g}_i \epsilon_i$$

IH merges a selected subset $\delta_s$ of the error symbols of a zonotope and their corresponding generator vectors. For a $d$-dimensional zonotope $\ell^\sharp$:

$$\mathbf{R}_{IH}(\delta_s, \ell^\sharp) = \mathbf{R}_{IH}\left(\delta_s, \boldsymbol{c} + \sum_{i=1}^{|\mathcal{S}|} \boldsymbol{g}_i \epsilon_i\right) = \boldsymbol{c} + \underbrace{\sum_{i \notin \delta_s} \boldsymbol{g}_i \epsilon_i}_{\text{Retained error symbols}} + \underbrace{\begin{bmatrix} \left(\sum_{i \in \delta_s} |\boldsymbol{g}_i[1]|\right)\epsilon_1' \\ \vdots \\ \left(\sum_{i \in \delta_s} |\boldsymbol{g}_i[d]|\right)\epsilon_d' \end{bmatrix}}_{\text{Over-approximated with } d\text{-dimensional box}}$$

The selected terms $\delta_s$ are merged into a $d$-dimensional box described by $d$ new error symbols $\{\epsilon_1', \cdots, \epsilon_d'\}$. We will drop $\delta_s$ if all error symbols of the input zonotope are selected.

The error symbols getting merged ($\delta_s$) are often determined based on some heuristic, e.g., symbols with lowest coefficients. TIH $\mathbf{R}_{TIH}$ first projects the zonotope to another space using an invertible linear transformation matrix $\boldsymbol{A} \in \mathbb{R}^{d \times d}$, then applies IH, and finally projects the resulting zonotope back with $\boldsymbol{A}^{-1}$. IH is a special case of TIH where $\boldsymbol{A} = I$.

$$\mathbf{R}_{TIH}(\delta_s, \boldsymbol{A}, \boldsymbol{z}^\sharp) = \boldsymbol{A}^{-1}\mathbf{R}_{IH}(\boldsymbol{A}\boldsymbol{z}^\sharp).$$

Intuitively, the purpose of the linear transformation $\boldsymbol{A}$ is to project the zonotope to a space where its shape is closer to a box, to reduce the loss of precision brought by order reduction [2, 38]. One of the most widely used TIH is PCA-based order reduction, whose transformation is obtained from the PCA of the set of all generator vectors of the input zonotope [38, 51].

**Order Reduction for PTIME Zonotope Training**  With linearization, the number of error symbols in the model weights zonotope $\boldsymbol{w}^\sharp$ still grows exponentially with rate $\mathcal{O}(p^2)$, leading to exponential time complexity for gradient descent. We can overcome this challenge through *order reduction*, which enforces the maximum number of terms in the symbolic expressions of model weights zonotopes [38]. Note that for a linear zonotope, each $\mathcal{S}[i]$ consists of a single error term $\epsilon_i$. Thus, we can write such a zonotope $\ell^\sharp$ as:

$$\ell^\sharp = \boldsymbol{c} + \sum_{i=1}^{|\mathcal{S}|} \boldsymbol{g}_i \epsilon_i$$

Order reduction $\mathbf{R}$ reduces the order (representation size) of a linear zonotope $\ell^\sharp$. This is achieved by *merging* error symbols in $\mathcal{S}$, while ensuring that the result over-approximation the input zonotope, i.e.,

$$\gamma\left(\mathbf{R}(\ell^\sharp)\right) \supseteq \gamma\left(\ell^\sharp\right)$$

Two commonly adopted order reduction approaches [38] are *Interval Hull* (*IH*), denoted as $\mathbf{R}_{IH}$, and *Transformation-based Interval Hull* (*TIH*), denoted as $\mathbf{R}_{TIH}$ [2, 38, 64]. Specifically, IH merges a set of error symbols $\delta_s \subseteq \{\epsilon_i\}$ and their corresponding generator vectors (here $\delta_k = \{\epsilon_i\} - \delta_s$):

$$\mathbf{R}_{IH}(\boldsymbol{z}^\sharp) = \mathbf{R}_{IH}\left(\boldsymbol{c} + \sum_{i=1}^{|\mathcal{S}|} \boldsymbol{g}_i \mathcal{S}[i]\right) = \boldsymbol{c} + \sum_{i \in \delta_k} \boldsymbol{g}_i \mathcal{S}[i] + diag(\sum_{i \in \delta_s} |\boldsymbol{g}_i[1]|, \cdots, \sum_{i \in \delta_s} |\boldsymbol{g}_i[d]|) \begin{bmatrix} \epsilon'_1 \\ \vdots \\ \epsilon'_d \end{bmatrix}$$

The selected terms $\delta_s$ are merged into a $d$-dimensional box described by $d$ new error symbols $\{\epsilon'_1, \cdots, \epsilon'_d\}$. The error symbols getting merged ($\delta_s$) are often determined based on some heuristic [38], e.g., the symbols with lowest coefficients. Similar to IH, TIH also merges error terms using IH, but in some projected space. TIH first projects the zonotope to another space using an invertible linear transformation matrix $A \in \mathbb{R}^{d \times d}$, then conducts IH, and finally projects the resulting zonotope back into the original space with $A^{-1}$:

$$\mathbf{R}_{TIH}(\boldsymbol{z}^\sharp) = A^{-1}\mathbf{R}_{IH}(A\boldsymbol{z}^\sharp).$$

Intuitively, the linear transformation $A$ aims to project the zonotope to a space where its shape is closer to a box, to reduce the loss of precision brought by order reduction [2, 38]. One of the most widely used TIH is PCA-based order reduction, whose transformation is obtained from the PCA of all generator vectors [38, 51].

# H Efficient Abstract Gradient Descent with Order Reduction

As mentioned in sec. 4.1, to address the tractability issues with abstract gradient descent, we employ two key techniques: linearization and order reduction. We employ linearization at each step of gradient descent to ensure that the resulting abstract representation of model parameters remains a linear zonotope.

Given a linearization operator $\mathbf{L}$ and order reduction operator $\mathbf{R}$, we construct an abstract gradient descent operator, $\Phi^\sharp$:

$$\Phi^\sharp(\boldsymbol{w}^\sharp) = \mathbf{R}\Big(\boldsymbol{w}^\sharp - \mathbf{L}\big(\eta \nabla L(\boldsymbol{w}^\sharp)\big)\Big), \tag{9}$$

which ensures that the abstract representation size remains bounded while providing efficient over-approximation. This operator is an abstract transformer for the concrete gradient descent operator.

**Proposition H.1.** *For any linearization $\mathbf{L}$ and order reduction $\mathbf{R}$, the abstract gradient descent $\Phi^\sharp$ is an abstract transformer for the concrete gradient descent operator $\Phi$. Formally, for any abstract $\boldsymbol{w}^\sharp$,*

$$\gamma\left(\Phi^\sharp(\boldsymbol{w}^\sharp)\right) \supseteq \Phi(\gamma\left(\boldsymbol{w}^\sharp\right)),$$

.

*Proof.* Given that abstract transformers compose (Prop. B.1), we can decompose $\Phi$ into separate steps and construct an abstract transformer for $\Phi$ by composing abstract transformers for the individual steps. We can inject identity functions anywhere into the computation without changing its result. Let $ident$ represent an identify function of an appropriate type and $F^\sharp$ be the abstract operator resulting from injecting identity functions as shown below, then:

$$F^\sharp(\boldsymbol{w}^\sharp) = ident\Big(\boldsymbol{w}^\sharp - ident\big(\eta \nabla L(\boldsymbol{w}^\sharp)\big)\Big) = \boldsymbol{w}^\sharp - \big(\eta \nabla L(\boldsymbol{w}^\sharp)\big) = \Phi^\sharp_{exact}(\boldsymbol{w}^\sharp)$$

Since $\Phi^\sharp_{exact}$ is an abstract transformer for $\Phi$, so is $F^\sharp$. Now observe that both $\mathbf{L}$ and $\mathbf{R}$ are abstract transformers for $ident$ as

$$\gamma\left(\mathbf{L}(\boldsymbol{w}^\sharp)\right) \supseteq \gamma\left(\boldsymbol{w}^\sharp\right) = ident(\gamma\left(\boldsymbol{w}^\sharp\right))$$

Thus, $\Phi^\sharp$ is an abstract transformer for $\Phi$. $\square$

Note that Prop. 3.3 then implies that a fixed point $\boldsymbol{w}^{\sharp*}$ for $\Phi^\sharp$ is an over-approximation of all possible model weights $\boldsymbol{w}^{\circ*}$:

$$\gamma\left(\boldsymbol{w}^{\sharp*}\right) \supseteq \boldsymbol{w}^{\circ*}$$

### H.1 Abstract Gradient Descent With Order Reduction And Fixed Points

While $\Phi^\sharp$ ensures that every step of gradient descent can be computed efficiently as we bound the order of the resulting zonotope in each step, this operator typically does not have a fixed point and even if it does, we still would have to solve an NP-hard problem [39] to detected that we have converged. The reason for the lack of a fixed point is that both linearization and order reduction results in over-approximation and the over-approximation error may grow in each iteration. Recall that we did show a real example of where this operator diverges in Sec. 4.1.

## I   An Efficient Approximate Abstract Transformer for Ridge Regression

In this section and the next section we present the proof of our main technical result: Thm. 4.2. Recall that Thm. 4.2 states that Alg. 1 computes abstract model parameters $\boldsymbol{w}^{\sharp *}$ that are a fixed point for the abstract transformer $\Phi^\sharp$ using a closed form solution. Because $\Phi^\sharp$ was shown to be an abstract transformer for gradient decent this then implies that $\boldsymbol{w}^{\sharp *}$ over-approximates all possible model weights $\boldsymbol{w}^{\odot *}$. We start by presenting additional details of the decomposition we employ to force a fixed points, formally prove that the condition on $\boldsymbol{w}^{\sharp *}$ from Prop. 4.1 is a sufficient for $\boldsymbol{w}^{\sharp *}$ being a fixed point for $\Phi^\sharp$, and then develop a closed form solution that requires solving a system of linear equations. For $\boldsymbol{w}_N^\sharp$, the parts of the abstract model weights that exclusively contains symbols that do not appear in the abstract training dataset $\mathbf{D}^\sharp$, the equation system only has a solution if the regularization coefficient $\lambda$ is larger then or equal to a constant $\beta$ that depends on $\mathbf{D}^\sharp$. Based on our experience and extensive experimental evaluation, we typically have $\beta = 0$, i.e., the closed form solution exists for any regularization coefficient $\lambda$. Nonetheless, we present a technique for achieving any desired $\beta \geq 0$ by splitting zonotopes into smaller parts with lower $\beta$, computing fixed points for each split individually, and merging the final result. These techniques will be presented in app. J.

### I.1   Decomposing Fixed Point Equations For Abstract Gradient Descent

We now present additional details about our decomposition of the gradient from Sec. 4.1 for linear regression with $\ell_2$ regularization (ridge regression). The loss function $L$ for ridge regression an $\ell_2$ penalty, is given by:

$$L(\boldsymbol{X}, \boldsymbol{y}, \boldsymbol{w}) = \frac{1}{n}(\boldsymbol{X}\boldsymbol{w} - \boldsymbol{y})^T(\boldsymbol{X}\boldsymbol{w} - \boldsymbol{y}) + \lambda \cdot \boldsymbol{w}^T\boldsymbol{w},$$

where $\lambda$ is the regularization coefficient (see App. A for details).

Recall that we observed that an abstract model weight zonotope $\boldsymbol{w}^\sharp$ can be decomposed into parts that can be dealt with separately in the sense that we will show that a sufficient condition for achieving a fixed point $\boldsymbol{w}^{\sharp *}$ is that each component has a fixed point. Given an abstract training dataset $\mathbf{D}^\sharp$, we use $\boldsymbol{X}^\sharp$ and $\boldsymbol{y}^\sharp$ to denote its feature matrix and labels. We decompose them into real (concrete) and symbolic components:

$$\boldsymbol{X}^\sharp = \boldsymbol{X}_R + \boldsymbol{X}_S^\sharp$$
$$\boldsymbol{y}^\sharp = \boldsymbol{y}_R + \boldsymbol{y}_S^\sharp$$

where $\boldsymbol{X}_R \in \mathbb{R}^{n \times d}$ and $\boldsymbol{y}_R \in \mathbb{R}^n$ represent the real centers of zonotopes $\boldsymbol{X}^\sharp$ and $\boldsymbol{y}^\sharp$, while $\boldsymbol{X}_S^\sharp \in \Lambda^{n \times d}$ and $\boldsymbol{y}_S^\sharp \in \Lambda^n$ contain the symbolic terms.

Similarly, we decompose the abstract model weights at iteration $i$ into real and symbolic components:

$$\boldsymbol{w}^{\sharp i} = \boldsymbol{w}_R^i + \boldsymbol{w}_S^{\sharp i},$$

where $\boldsymbol{w}_R^i \in \mathbb{R}^d$ represents the real center, and $\boldsymbol{w}_S^{\sharp i} \in \Lambda^d$ contains the symbolic terms. The symbolic terms are further decomposed into those containing data symbols ($\boldsymbol{w}_D^{\sharp i}$), i.e., symbols that are shared with $\mathbf{D}^\sharp$, and those introduced via linearization and order reduction in linear abstract gradient descent ($\boldsymbol{w}_N^{\sharp i}$), i.e., that do not appear in $\mathbf{D}^\sharp$.

$$\boldsymbol{w}_S^{\sharp i} = \boldsymbol{w}_D^{\sharp i} + \boldsymbol{w}_N^{\sharp i}.$$

Accordingly, we decompose the abstract gradient presented in Eq. (9) into several distinct components: real numbers ($\mathcal{G}_R$), linear symbolic expressions that share symbols with $\mathbf{D}^\sharp$ ($\mathcal{G}_L^D$), linear symbolic expressions that do not share symbols with $\mathbf{D}^\sharp$ ($\mathcal{G}_L^N$), and high-order symbolic expressions ($\mathcal{G}_H$). Using the loss function for ridge regression (see Eq. (5) in App. A):

$$\Phi^\sharp(\boldsymbol{w}^\sharp) = \mathbf{R}\Big(\boldsymbol{w}^\sharp - \mathbf{L}\big(\eta\nabla L(\boldsymbol{w}^\sharp)\big)\Big) = \mathbf{R}\Big(\boldsymbol{w}^\sharp - \eta\mathbf{L}\big(\tfrac{2}{n}(\boldsymbol{X}^{\sharp T}\boldsymbol{X}^\sharp\boldsymbol{w}^\sharp - \boldsymbol{X}^{\sharp T}\boldsymbol{y}^\sharp) + 2\lambda\boldsymbol{w}^\sharp\big)\Big)$$

$$= \mathbf{R}\Big(\boldsymbol{w}_R + \boldsymbol{w}_D^\sharp + \boldsymbol{w}_N^\sharp - \eta\mathbf{L}\big(\tfrac{2}{n}\big((\boldsymbol{X}_R + \boldsymbol{X}_S^\sharp)^T(\boldsymbol{X}_R + \boldsymbol{X}_S^\sharp)(\boldsymbol{w}_R + \boldsymbol{w}_D^\sharp + \boldsymbol{w}_N^\sharp)$$
$$- (\boldsymbol{X}_R + \boldsymbol{X}_S^\sharp)^T(\boldsymbol{y}_R + \boldsymbol{y}_S^\sharp)\big) + 2\lambda(\boldsymbol{w}_R + \boldsymbol{w}_D^\sharp + \boldsymbol{w}_N^\sharp)\big)\Big) \tag{10}$$

$$= \mathbf{R}\Big(\boldsymbol{w}_R - \eta\cdot\mathcal{G}_R + \boldsymbol{w}_D^\sharp - \eta\cdot\mathcal{G}_L^D + \boldsymbol{w}_N^\sharp - \eta\cdot\mathcal{G}_L^N - \eta\cdot\mathbf{L}(\mathcal{G}_H)\Big) \tag{11}$$

where

$$\mathcal{G}_R = \frac{2}{n}(\boldsymbol{X}_R{}^T\boldsymbol{X}_R\boldsymbol{w}_R - \boldsymbol{X}_R{}^T\boldsymbol{y}_R) + 2\lambda\boldsymbol{w}_R \tag{12}$$

$$\mathcal{G}_L^D = (2\lambda I + \frac{2}{n}\boldsymbol{X}_R{}^T\boldsymbol{X}_R)\boldsymbol{w}_D^\sharp + \frac{2}{n}(\boldsymbol{X}_R{}^T\boldsymbol{X}_S^\sharp + \boldsymbol{X}_S^\sharp{}^T\boldsymbol{X}_R)\boldsymbol{w}_R - \frac{2}{n}\boldsymbol{X}_S^\sharp\boldsymbol{y}_R - \frac{2}{n}\boldsymbol{X}_R{}^T\boldsymbol{y}_S^\sharp \tag{13}$$

$$\mathcal{G}_L^N = (2\lambda I + \frac{2}{n}\boldsymbol{X}_R{}^T\boldsymbol{X}_R)\boldsymbol{w}_N^\sharp \tag{14}$$

$$\mathcal{G}_H = \frac{2}{n}\Big((\boldsymbol{X}_R{}^T\boldsymbol{X}_S^\sharp + \boldsymbol{X}_S^\sharp{}^T\boldsymbol{X}_R)(\boldsymbol{w}_D^\sharp + \boldsymbol{w}_N^\sharp) + \boldsymbol{X}_S^\sharp{}^T\boldsymbol{X}_S^\sharp(\boldsymbol{w}_R + \boldsymbol{w}_D^\sharp + \boldsymbol{w}_N^\sharp) - \boldsymbol{X}_S^\sharp{}^T\boldsymbol{y}_S^\sharp\Big) \tag{15}$$

In Eq. (10) we substitute definitions which are further expanded in Eq. (11) using new notation for rearranged parts of the gradient (Eq. (12) to (15)). Furthermore, in this step we also use the fact that linearization only affects higher order terms and $\mathcal{G}_H$ is the only component of the gradient with non-linear terms. Now consider the relationship of Eq. (12) to (15) to

**Enforcing Fixed Points With Parameterized Order Reduction.** We now introduce an order reduction operator $\mathbf{R}_A$ that is a specific type of TIH (see App. G) which only merges non-data symbols and is parameterized by its transformation matrix $A$, identify a sufficient condition for achieving an abstract fixed point for gradient descent with this order reduction operator, and demonstrate that the fixed point exists for any coefficient $\lambda$ of $l_2$-regularization that is larger than a data-dependent constant $\beta$. Furthermore, we show how such a fixed point $\boldsymbol{w}^{\sharp *}$ can be computed using closed form solutions for $\boldsymbol{w}_R$, $\boldsymbol{w}_D^\sharp$ and $\boldsymbol{w}_N^\sharp$.

Recall from App. G that the TIH order reduction operator $\mathbf{R}_{TIH}(\delta_s, A)$ is parameterized by $\delta_s$ (the set of error symbols to merge) and a transformation matrix $A$. Consider the input to order reduction as shown in Eq. (10). We use $\mathcal{E}_D$ to denote the error symbols that this input shares with the data (that occur in $\mathbf{D}^\sharp$) and $\mathcal{E}_N$ to denote those that only occur in the input to order reduction. Then we define $\mathbf{R}_A = \mathbf{R}_{TIH}(\mathcal{E}_N, A)$, i.e., we only merge symbols that are not shared with the data to ensure that the correspondence between model weights and possible worlds is preserved by letting $\boldsymbol{w}^\sharp$ share symbols in linear expressions with $\mathbf{D}^\sharp$. The linear part that does not share symbols with $\mathbf{D}^\sharp$ has $O(p^2 q)$ monomials (where $p = |\mathcal{E}_D|$ and $q$ is the number of error symbols in $\boldsymbol{w}^\sharp$). Now expanding the definition of abstract gradient descent with order reduction using $\mathbf{R}_A$ as the order reduction operator (for some given transformation matrix $A$), we get:

$$\Phi_{(\mathbf{L},\mathbf{R}_A)}^\sharp(\boldsymbol{w}^\sharp) = \mathbf{R}_A(\boldsymbol{w}_R - \eta\mathcal{G}_R + \boldsymbol{w}_D^\sharp - \eta\mathcal{G}_L^D + \boldsymbol{w}_N^\sharp - \eta(\mathcal{G}_L^N + \mathbf{L}(\mathcal{G}_H))) \tag{16}$$

$$= \boldsymbol{w}_R - \eta\mathcal{G}_R + \boldsymbol{w}_D^\sharp - \eta\mathcal{G}_L^D + \mathbf{R}_A(\boldsymbol{w}_N^\sharp - \eta(\mathcal{G}_L^N + \mathbf{L}(\mathcal{G}_H))) \tag{17}$$

$$= \underbrace{\boldsymbol{w}_R - \eta\mathcal{G}_R}_{\Phi_R(\boldsymbol{w}_R)} + \underbrace{\boldsymbol{w}_D^\sharp - \eta\mathcal{G}_L^D}_{\Phi^\sharp{}_D(\boldsymbol{w}_R,\boldsymbol{w}_D^\sharp)} + \underbrace{A^{-1}\big(\mathbf{R}_{IH}(A(\boldsymbol{w}_N^\sharp - \eta(\mathcal{G}_L^N + \mathbf{L}(\mathcal{G}_H))))\big)}_{\Phi^\sharp{}_N(\boldsymbol{w}_R,\boldsymbol{w}_D^\sharp,\boldsymbol{w}_N^\sharp)} \tag{18}$$

Eq. (18) shows the components of $\Phi_{(\mathbf{L},\mathbf{R}_A)}^\sharp(\boldsymbol{w}^\sharp)$ and related them to the notation $\Phi_R(\boldsymbol{w}_R)$, $\Phi^\sharp{}_D(\boldsymbol{w}_R,\boldsymbol{w}_D^\sharp)$, and $\Phi^\sharp{}_N(\boldsymbol{w}_R,\boldsymbol{w}_D^\sharp,\boldsymbol{w}_N^\sharp)$ we introduced in Eq. (2). Specifically, $\Phi_R(\boldsymbol{w}_R) = (\boldsymbol{w}_R - \eta\mathcal{G}_R)$ does not contain any error symbols and, thus, corresponds to the real number part of the updated model weight. Similarly, $\Phi^\sharp{}_D(\boldsymbol{w}_R,\boldsymbol{w}_D^\sharp) = (\boldsymbol{w}_D^\sharp - \eta\mathcal{G}_L^D)$ only contains linear symbolic expression with symbols that are from $\mathbf{D}^\sharp$. The remaining part, $\Phi^\sharp{}_N(\boldsymbol{w}_R,\boldsymbol{w}_D^\sharp,\boldsymbol{w}_N^\sharp) =$

$A\big(\mathbf{R}_{IH}(A^{-1}(\boldsymbol{w}_N^\sharp - \eta(\mathcal{G}_L^N + \mathbf{L}(\mathcal{G}_H)))))\big)$ does not share symbols with $\mathbf{D}^\sharp$, as it consists of new symbols generated from order reduction.

## I.2 Constructing Fixed Points For Abstract Gradient Descent

Having defined $\Phi_{(\mathbf{L},\mathbf{R}_A)}^\sharp$ which was denoted by $\Phi_{(\mathbf{L},\mathbf{R})}^\sharp$ in app. H, we are now ready to state the first part of our main technical result: under some mild assumptions on $\lambda$ being larger than some data-dependent constant $\beta$, $\Phi_{(\mathbf{L},\mathbf{R}_A)}^\sharp$ has a fixed point and, importantly, this fixed point can be computed efficiently. In practice, we often have $\beta = 0$. However, if this is not the case, we demonstrate later that at the cost of reduced computational efficiency we can reduce $\beta$ by splitting $\mathbf{D}^\sharp$ into several zonotopes and solving the problem independently for each zonotope as we will details in App. J.

**Theorem I.1** (Existence of Abstract Fixed Points). *Consider an abstract training dataset $\mathbf{D}^\sharp$. There exists a constant $\beta$ specific to $\mathbf{D}^\sharp$ such that $\Phi_{(\mathbf{L},\mathbf{R}_A)}^\sharp$ has a fixed point for any $\lambda \geq \beta$.*

In the remainder of this section, we prove Thm. I.1 by demonstrating how to construct such a fixed point efficiently. We start by identifying a sufficient condition for $\boldsymbol{w}^{\sharp*}$ to be a fixed point, then demonstrate that the problem of finding a $\boldsymbol{w}^{\sharp*}$ that fulfills this sufficient condition can be decomposed based on our decomposition $\boldsymbol{w}^{\sharp*} = \boldsymbol{w}_R^* + \boldsymbol{w}_D^{\sharp*} + \boldsymbol{w}_N^{\sharp*}$ presented earlier. Specifically, certain components are independent of other components suggesting an evaluation order where we find a fixed point for one component treating the previously computed fixed points for other components as constants. Then we proceed to prove closed form solutions for $\boldsymbol{w}_R^*$ and for $\boldsymbol{w}_D^{\sharp*}$ given $\boldsymbol{w}_R^*$. Finally, given $\boldsymbol{w}_R^*$ and $\boldsymbol{w}_D^{\sharp*}$ we construct a system of equations whose solution gives a fixed point for $\boldsymbol{w}_N^{\sharp*}$ and demonstrate that this system of equations has a closed form solution for any $\lambda \geq \beta$.

**A Sufficient Condition for Abstract Fixed Points.** Recall from Def. 3.2 that the fixed point $\boldsymbol{w}^{\sharp*}$ must satisfy $\gamma\left(\boldsymbol{w}^{\sharp*}, \mathbf{D}^\sharp\right) \supseteq \gamma\left(\Phi_{(\mathbf{L},\mathbf{R}_A)}^\sharp(\boldsymbol{w}^{\sharp*}), \mathbf{D}^\sharp\right)$. We now present a sufficient condition for this to hold. Intuitively this condition requires the **invariance** of different components of $\Phi_{(\mathbf{L},\mathbf{R}_A)}^\sharp(\boldsymbol{w}^\sharp)$ when doing joint concretization with $\mathbf{D}^\sharp$.

**Proposition I.2.** *If an abstract model weight $\boldsymbol{w}^{\sharp*} = \boldsymbol{w}_R^* + \boldsymbol{w}_D^{\sharp*} + \boldsymbol{w}_N^{\sharp*}$ satisfies the following three conditions, then it is an abstract fixed point of the abstract gradient descent operator $\Phi_{(\mathbf{L},\mathbf{R}_A)}^\sharp$:*

$$\boldsymbol{w}_R^* = \boldsymbol{w}_R^* - \eta\mathcal{G}_R \tag{19}$$

$$\boldsymbol{w}_D^{\sharp*} = \boldsymbol{w}_D^{\sharp*} - \eta\mathcal{G}_L^D \tag{20}$$

$$\boldsymbol{w}_N^{\sharp*} \simeq_\sharp \mathbf{R}_A\left(\boldsymbol{w}_N^\sharp - \eta(\mathcal{G}_L^N + \mathbf{L}(\mathcal{G}_H))\right). \tag{21}$$

*Proof.* Consider the definition of an abstract fixed point $\boldsymbol{w}^{\sharp*}$ $\left(\text{Def. 3.2 applied to } \Phi_{(\mathbf{L},\mathbf{R}_A)}^\sharp\right)$:

$$\gamma\left(\boldsymbol{w}^{\sharp*}, \mathbf{D}^\sharp\right) \supseteq \gamma\left(\Phi_{(\mathbf{L},\mathbf{R}_A)}^\sharp(\boldsymbol{w}^{\sharp*}), \mathbf{D}^\sharp\right)$$

This can certainly be achieved if

$$\boldsymbol{w}^{\sharp*}, \mathbf{D}^\sharp \simeq_\sharp \Phi_{(\mathbf{L},\mathbf{R}_A)}^\sharp(\boldsymbol{w}^{\sharp*}), \mathbf{D}^\sharp$$

This requires equal joint concretization of the fixed point the pairs $(\boldsymbol{w}^{\sharp*}, \mathbf{D}^\sharp)$ and $(\Phi_{(\mathbf{L},\mathbf{R}_A)}^\sharp(\boldsymbol{w}^{\sharp*}), \mathbf{D}^\sharp)$. To understand the need for joint concretization here note that certain model weights $\boldsymbol{w}$paired with certain datasets $\boldsymbol{D}$ in the joint concretization. If we would only consider equivalence of the abstract model weight, two zonotopes $\boldsymbol{w}^\sharp$ and $\boldsymbol{w}^{\sharp'}$ may be equivalent $\boldsymbol{w}^\sharp \simeq_\sharp \boldsymbol{w}^{\sharp'}$, but pair a particular model weight $\boldsymbol{w} \in \gamma\left(\boldsymbol{w}^\sharp\right) = \gamma\left(\boldsymbol{w}^{\sharp'}\right)$ with different datasets $\boldsymbol{D}_1 \in \gamma\left(\mathbf{D}^\sharp\right)$ and $\boldsymbol{D}_2 \in \gamma\left(\mathbf{D}^\sharp\right)$. Thus, it is possible that $\boldsymbol{w}^\sharp \simeq_\sharp \Phi_{(\mathbf{L},\mathbf{R}_A)}^\sharp(\boldsymbol{w}^\sharp)$ holds but $\gamma\left(\boldsymbol{w}^\sharp\right) = \gamma\left(\boldsymbol{w}^{\sharp'}\right)$ does not contain all model weights from $\boldsymbol{w}^{\odot*}$.

First observe that requiring equality of abstract elements is a stricter condition than equivalence wrt. $\simeq_\sharp$ as

$$\boldsymbol{w}^\sharp{}_1 = \boldsymbol{w}^\sharp{}_2 \Rightarrow \boldsymbol{w}^\sharp{}_1 \simeq_\sharp \boldsymbol{w}^\sharp{}_2$$

Also note that $\boldsymbol{w}_D^\sharp$ is the only component of $\boldsymbol{w}^{\sharp*}$ that contains symbols. Thus, requiring equality in Eq. (20) is sufficient for ensuring equivalent join concretization with $\mathbf{D}^\sharp$ as long as Eq. (19) and Eq. (21) also hold. Furthermore, Eq. (19) and Eq. (20) imply:

$$\mathcal{G}_R = 0 \qquad\qquad\qquad \mathcal{G}_L^D = 0$$

Expanding $\boldsymbol{w}^{\sharp*} = \boldsymbol{w}_R^* + \boldsymbol{w}_D^{\sharp*} + \boldsymbol{w}_N^{\sharp*}$ and substituting into Eq. (18) we get

$$\Phi_{(\mathbf{L},\mathbf{R}_A)}^\sharp(\boldsymbol{w}^{\sharp*}) = \boldsymbol{w}_R^* - \eta\mathcal{G}_R + \boldsymbol{w}_D^{\sharp*} - \eta\mathcal{G}_L^D + \mathbf{R}_A\big(\boldsymbol{w}_N^{\sharp*} - \eta(\mathcal{G}_L^N + \mathbf{L}(\mathcal{G}_H))\big)$$

$$= \boldsymbol{w}_R^* + \boldsymbol{w}_D^{\sharp*} + \mathbf{R}_A\big(\boldsymbol{w}_N^{\sharp*} - \eta(\mathcal{G}_L^N + \mathbf{L}(\mathcal{G}_H))\big)$$

Thus, using Eq. (21) and the fact that $\boldsymbol{w}_N^\sharp$ does not share any symbols with $\mathbf{D}^\sharp$, we get the desired result:

$$(19) \wedge (20) \wedge \boldsymbol{w}_N^{\sharp*} \simeq_\sharp \mathbf{R}_A\left(\boldsymbol{w}_N^\sharp - \eta(\mathcal{G}_L^N + \mathbf{L}(\mathcal{G}_H))\right) \Rightarrow (\boldsymbol{w}^{\sharp*}, \mathbf{D}^\sharp) \simeq_\sharp (\Phi_{(\mathbf{L},\mathbf{R}_A)}^\sharp(\boldsymbol{w}^{\sharp*}), \mathbf{D}^\sharp)$$

$$\square$$

**Independence of Fixed Point Components.** Recall that we observed that some components of the gradient depend only on some components of $\boldsymbol{w}^{\sharp*}$. Specifically, $\mathcal{G}_R$ only depends on $\boldsymbol{w}_R^*$ and $\mathcal{G}_L^D$ only depends on $\boldsymbol{w}_R^*$ and $\boldsymbol{w}_D^{\sharp*}$. This implies that we can find a fixed point by first computing $\boldsymbol{w}_R^*$, then finding $\boldsymbol{w}_D^{\sharp*}$ treating $\boldsymbol{w}_R^*$ as a constant, and finally determine $\boldsymbol{w}_N^{\sharp*}$ treating both $\boldsymbol{w}_R^*$ and $\boldsymbol{w}_D^{\sharp*}$ as a constant.

**Lemma I.3** (Independence of Fixed Point Components). *The following independence relationships hold:*

*(i)* $\mathcal{G}_R$ *does not depend on* $\boldsymbol{w}_D^{\sharp*}$ *nor* $\boldsymbol{w}_N^{\sharp*}$          *(ii)* $\mathcal{G}_L^D$ *does not depend on* $\boldsymbol{w}_N^{\sharp*}$

*Proof.* The definition of $\mathcal{G}_R$ is shown in Eq. (12). The only component of $\boldsymbol{w}^{\sharp*}$ that appears in the definition of $\mathcal{G}_R$ is $\boldsymbol{w}_R^*$. Thus, the equations defining $\boldsymbol{w}_R^*$ only depend on $\boldsymbol{w}_R^*$ and are independent of $\boldsymbol{w}_D^{\sharp*}$ and $\boldsymbol{w}_N^{\sharp*}$. Using a similar argument we can show that $\mathcal{G}_L^D$ does not depend on $\boldsymbol{w}_N^{\sharp*}$. $\square$

**Closed Form Solutions for Real And Linear Data Components of Abstract Model Weights.** Before explaining how to determine $\boldsymbol{w}_N^{\sharp*}$, utilizing the lemma above we derive closed form solutions for $\boldsymbol{w}_R^*$ and $\boldsymbol{w}_D^{\sharp*}$.

**Lemma I.4** (Closed Form Solutions for $\boldsymbol{w}_R^*$ and $\boldsymbol{w}_D^{\sharp*}$). *Given* $\mathbf{D}^\sharp$, $\boldsymbol{w}_R^*$ *as defined below fulfills Eq. (19)*

$$\boldsymbol{w}_R^* = (\boldsymbol{X}_R{}^T\boldsymbol{X}_R + \lambda nI)^{-1}\boldsymbol{X}_R{}^T\boldsymbol{y}_R \tag{22}$$

*Given* $\mathbf{D}^\sharp$ *and* $\boldsymbol{w}_R^*$, $\boldsymbol{w}_D^{\sharp*}$ *as defined below fulfills Eq. (20).*

$$\boldsymbol{w}_D^{\sharp*} = (\boldsymbol{X}_R{}^T\boldsymbol{X}_R + \lambda nI)^{-1}\Big(\boldsymbol{X}_S^\sharp \boldsymbol{y}_R + \boldsymbol{X}_R{}^T\boldsymbol{y}_S^\sharp - (\boldsymbol{X}_R{}^T\boldsymbol{X}_S^\sharp + \boldsymbol{X}_S^\sharp{}^T\boldsymbol{X}_R)\boldsymbol{w}_R^*\Big) \tag{23}$$

*Proof.* *Fixed Point for* $\boldsymbol{w}_R^*$. Substituting the definition of $\mathcal{G}_R$ from Eq. (12) into Eq. (21):

$$\boldsymbol{w}_R^* = \boldsymbol{w}_R^* - \eta \cdot \left(\frac{2}{n}(\boldsymbol{X}_R{}^T\boldsymbol{X}_R\boldsymbol{w}_R^* - \boldsymbol{X}_R{}^T\boldsymbol{y}_R) + 2\lambda\boldsymbol{w}_R^*\right)$$

$$\Leftrightarrow \frac{2\eta}{n} \cdot \left(\boldsymbol{X}_R{}^T\boldsymbol{X}_R\boldsymbol{w}_R^* - \boldsymbol{X}_R{}^T\boldsymbol{y}_R + \lambda n\boldsymbol{w}_R^*\right) = 0$$

$$\Leftrightarrow \boldsymbol{X}_R{}^T\boldsymbol{X}_R\boldsymbol{w}_R^* - \boldsymbol{X}_R{}^T\boldsymbol{y}_R + \lambda n\boldsymbol{w}_R^* = 0 \qquad (\eta > 0)$$

$$\Leftrightarrow (\boldsymbol{X}_R{}^T\boldsymbol{X}_R + \lambda nI)\boldsymbol{w}_R^* = \boldsymbol{X}_R{}^T\boldsymbol{y}_R$$

$$\Leftrightarrow \boldsymbol{w}_R^* = (\boldsymbol{X}_R{}^T\boldsymbol{X}_R + \lambda nI)^{-1}\boldsymbol{X}_R{}^T\boldsymbol{y}_R \tag{24}$$

The last step relies on the fact that $\boldsymbol{X_R}^T\boldsymbol{X_R} + \lambda nI$ is invertible which is guaranteed for $\lambda > 0$ [26]. Note that, as expected since the fixed point equation for $\boldsymbol{w}_R^*$ does not contain any symbolic expressions, the closed form for $\boldsymbol{w}_R^*$ is equal to the well-known closed form for ridge regression.

_Fixed Point for $\boldsymbol{w}_D^{\sharp*}$._ Note that given the independence of $\boldsymbol{w}_R^*$ from $\boldsymbol{w}_D^{\sharp*}$, we can assume that $\boldsymbol{w}_R^*$ has been determined using the closed form solution shown above. Thus, we can treat $\boldsymbol{w}_R^*$ as a constant in the following derivation. We start by substituting the definition or Eq. (13) into Eq. (20).

$$\boldsymbol{w}_D^{\sharp*} = \boldsymbol{w}_D^{\sharp*} - \eta\Big((2\lambda I + \frac{2}{n}\boldsymbol{X_R}^T\boldsymbol{X_R})\boldsymbol{w}_D^{\sharp*} + \frac{2}{n}(\boldsymbol{X_R}^T\boldsymbol{X}_S^\sharp + \boldsymbol{X}_S^{\sharp T}\boldsymbol{X_R})\boldsymbol{w}_R^*$$
$$- \frac{2}{n}(\boldsymbol{X}_S^\sharp \boldsymbol{y}_R + \boldsymbol{X_R}^T\boldsymbol{y}_S^\sharp)\Big)$$
$$\Leftrightarrow 0 = \frac{2\eta}{n}\Big((\boldsymbol{X_R}^T\boldsymbol{X_R} + \lambda nI)\boldsymbol{w}_D^{\sharp*} + (\boldsymbol{X_R}^T\boldsymbol{X}_S^\sharp + \boldsymbol{X}_S^{\sharp T}\boldsymbol{X_R})\boldsymbol{w}_R^* - \boldsymbol{X}_S^\sharp\boldsymbol{y}_R - \boldsymbol{X_R}^T\boldsymbol{y}_S^\sharp\Big)$$
$$\Leftrightarrow 0 = (\lambda nI + \boldsymbol{X_R}^T\boldsymbol{X_R})\boldsymbol{w}_D^{\sharp*} + (\boldsymbol{X_R}^T\boldsymbol{X}_S^\sharp + \boldsymbol{X}_S^{\sharp T}\boldsymbol{X_R})\boldsymbol{w}_R^* - \boldsymbol{X}_S^\sharp\boldsymbol{y}_R - \boldsymbol{X_R}^T\boldsymbol{y}_S^\sharp \quad (\eta > 0)$$
$$\Leftrightarrow (\lambda nI + \boldsymbol{X_R}^T\boldsymbol{X_R})\boldsymbol{w}_D^{\sharp*} = -\Big((\boldsymbol{X_R}^T\boldsymbol{X}_S^\sharp + \boldsymbol{X}_S^{\sharp T}\boldsymbol{X_R})\boldsymbol{w}_R^* - \boldsymbol{X}_S^\sharp\boldsymbol{y}_R - \boldsymbol{X_R}^T\boldsymbol{y}_S^\sharp\Big)$$
$$\Leftrightarrow \boldsymbol{w}_D^{\sharp*} = (\boldsymbol{X_R}^T\boldsymbol{X_R} + \lambda nI)^{-1}\Big(\boldsymbol{X}_S^\sharp\boldsymbol{y}_R + \boldsymbol{X_R}^T\boldsymbol{y}_S^\sharp - (\boldsymbol{X_R}^T\boldsymbol{X}_S^\sharp + \boldsymbol{X}_S^{\sharp T}\boldsymbol{X_R})\boldsymbol{w}_R^*\Big) \quad (25)$$

Note that matrix inversion is applied to $(\boldsymbol{X_R}^T\boldsymbol{X_R} + \lambda nI)$ which does not contain any symbolic terms and as discussed above this matrix is guaranteed to be invertible. $\qquad\square$

**Analysis of the Fixed Point Equations for $w_N^{\sharp*}$.** Using the closed form solutions established in Lem. I.4, we can compute $\boldsymbol{w}_R^*$ and $\boldsymbol{w}_D^{\sharp*}$ fulfilling Eq. (19) and Eq. (20). In the remainder of this subsection we show how to compute a solution $\boldsymbol{w}_N^{\sharp*}$ to Eq. (21) for a given $\lambda \geq \beta$ (regularization coefficient), i.e., we prove Thm. I.1 that postulated the existence of such fixed points $\boldsymbol{w}^{\sharp*}$ constructively.

To find a fixed point $\boldsymbol{w}_N^{\sharp*}$, we are searching for an abstract model weight that fulfills Eq. (21):

$$\boldsymbol{w}_N^{\sharp*} \simeq_\sharp \mathbf{R}_{\boldsymbol{A}}\left(\boldsymbol{w}_N^\sharp - \eta(\mathcal{G}_L^N + \mathbf{L}(\mathcal{G}_H))\right) \tag{26}$$

This implies that $\boldsymbol{w}_N^{\sharp*}$ needs to be equivalent to the result of $\mathbf{R}_{\boldsymbol{A}}$ on some input zonotope $\boldsymbol{w}_N^\sharp - \eta(\mathcal{G}_L^N + \mathbf{L}(\mathcal{G}_H))$. Recall the definition of $\mathbf{R}_{\boldsymbol{A}}$ for a linear input zonotope $\ell^\sharp$:

$$\mathbf{R}_{\boldsymbol{A}}(\ell^\sharp) = \boldsymbol{A}^{-1}\mathbf{R}_{IH}(\boldsymbol{A}\ell^\sharp) \tag{27}$$

Note that $\mathbf{R}_{\boldsymbol{A}}$ applies $\mathbf{R}_{IH}$ to all error symbols in its input. Thus, $\mathbf{R}_{IH}(\cdot)$ is a $d$-dimensional box in the projected space into which the input $\ell^\sharp$ is mapped into by $\boldsymbol{A}$. Furthermore, as $\boldsymbol{w}_N^\sharp - \eta(\mathcal{G}_L^N + \mathbf{L}(\mathcal{G}_H))$ does only contain symbolic terms, the interval hull (the box computed by $\mathbf{R}_{IH}$) is centered at $\boldsymbol{0}$.

**Lemma I.5.** _Any solution $\boldsymbol{w}_N^\sharp$ of Eq. (26) is equivalent to the image of a box $\boldsymbol{b}^\sharp$ centered at $\boldsymbol{0}$ produced by $\boldsymbol{A}^{-1}$, i.e.,_

$$\boldsymbol{w}_N^\sharp \simeq_\sharp \boldsymbol{A}^{-1}\boldsymbol{b}^\sharp$$

_Proof._ IH does not change the center of a zonotope and $\boldsymbol{A}$ is a linear map. Now observe that all terms in Eq. (14) and (15) are symbolic (contain error symbols). Thus, the RHS of Eq. (26) that is the input to order reduction is a zonotope with center $\boldsymbol{0}$. $\qquad\square$

A linear zonotope that is a box in $d$-dimensional space with center $\boldsymbol{0}$ is, up to equivalence, uniquely determined by the diameter of the box in each dimension. That is, if we use $k_i \geq 0$ to denote the diameter of the box in the $i^{th}$ dimension and $\boldsymbol{k} \in \mathbb{R}^d$ to denote the vector of these elements, then we can write any such zonotope $\boldsymbol{b}^\sharp$ as:

$$\boldsymbol{b}^\sharp = \begin{bmatrix} k_1\epsilon_1 \\ \vdots \\ k_d\epsilon_d \end{bmatrix} = \begin{bmatrix} k_1 & \cdots & 0 \\ \vdots & \ddots & \vdots \\ 0 & \cdots & k_d \end{bmatrix}\begin{bmatrix} \epsilon_1 \\ \vdots \\ \epsilon_d \end{bmatrix} = diag\{\boldsymbol{k}\}\begin{bmatrix} \epsilon_1 \\ \vdots \\ \epsilon_d \end{bmatrix}$$

Combining Eq. (26) with Eq. (27), we know that $\boldsymbol{w}_N^\sharp$ is a zonotope that is equivalent to the image of $\boldsymbol{A}^{-1}$ on a box $\boldsymbol{b}^\sharp$ parameterized by $\boldsymbol{k}$. We will only consider solutions of this form:[2]

$$\boldsymbol{w}_N^{\sharp\,*} = \boldsymbol{A}^{-1} diag\{\boldsymbol{k}\} \begin{bmatrix} \epsilon_1 \\ \vdots \\ \epsilon_d \end{bmatrix} = \sum_{i=1}^d k_i \boldsymbol{a}_i \epsilon_i \tag{28}$$

where $\boldsymbol{a}_1, \cdots, \boldsymbol{a}_d$ are the column vectors of $\boldsymbol{A}^{-1}$. As the naming of error symbols is irrelevant, finding a solution to Eq. (26) now amounts to finding a vector $\boldsymbol{k}$ as the error symbols are now treated as fixed.

Now we can substitute the formula for $\boldsymbol{w}_N^{\sharp\,*}$ from Eq. (28) into the RHS of Eq. (26) as shown below. Recall from app. G.2 that IH merges different error symbols by adding up the absolute values of their coefficients. For the $i^{th}$ dimension, each $k_j$ will appear with a certain coefficient. For some terms, the symbolic expression for the $i^{th}$ dimension may contain error symbols that do not have any $k_l$ in their coefficients. Thus, without considering for now the precise values for each coefficient we know that the box produced by IH in the RHS is of the form shown below where $\boldsymbol{C} \in \mathbb{R}^{d\times d}$ is a matrix with non-negative entries $c_{i,j} \geq 0$ (at position $(i,j)$), and $\boldsymbol{c}_0 \in \mathbb{R}^d$ is a vector with non-negative elements $c_{i,0} \geq 0$ (at dimension $i$). Intuitively, $c_{i,j}$ is the coefficient of $k_j$ in the $i^{th}$ dimension and $c_{i,0}$ is the sum of the coefficients of error symbols that do not have any $k_j$ in their coefficients (e.g., the fresh symbols introduced by linearization). Note since IH does calculate coefficients of the error symbols in the output as a sum of absolute values, we know that all entries of $\boldsymbol{C}$ and $\boldsymbol{c}_0$ are positive. Furthermore, recall that we assumed that $k_j \geq 0$.

$$\mathbf{R}_{\boldsymbol{A}}(\boldsymbol{w}_N^\sharp - \eta(\mathcal{G}_L^N + \mathbf{L}(\mathcal{G}_H))) = \boldsymbol{A}^{-1}\mathbf{R}_{IH}(\boldsymbol{A}(\boldsymbol{w}_N^\sharp - \eta(\mathcal{G}_L^N + \mathbf{L}(\mathcal{G}_H))))$$

$$= \boldsymbol{A}^{-1} \begin{bmatrix} (c_{1,0} + \sum_{j=1}^d c_{1,j}k_j)\epsilon_1' \\ \vdots \\ (c_{d,0} + \sum_{j=1}^d c_{d,j}k_j)\epsilon_d' \end{bmatrix} = \boldsymbol{A}^{-1}diag\{\boldsymbol{c}_0 + \boldsymbol{C}\boldsymbol{k}\} \begin{bmatrix} \epsilon_1' \\ \vdots \\ \epsilon_d' \end{bmatrix}. \tag{29}$$

Substituting the LHS and RHS of Eq. (21) with Eq. (28) and (29), we get:

$$\boldsymbol{A}^{-1}diag\{\boldsymbol{k}\} \begin{bmatrix} \epsilon_1 \\ \vdots \\ \epsilon_d \end{bmatrix} \simeq_\sharp \boldsymbol{A}^{-1}diag\{\boldsymbol{c}_0 + \boldsymbol{C}\boldsymbol{k}\} \begin{bmatrix} \epsilon_1' \\ \vdots \\ \epsilon_d' \end{bmatrix}$$

$$\Leftrightarrow diag\{\boldsymbol{k}\} \begin{bmatrix} \epsilon_1 \\ \vdots \\ \epsilon_d \end{bmatrix} \simeq_\sharp diag\{\boldsymbol{c}_0 + \boldsymbol{C}\boldsymbol{k}\} \begin{bmatrix} \epsilon_1' \\ \vdots \\ \epsilon_d' \end{bmatrix} \tag{30}$$

$$\Leftrightarrow \boldsymbol{k} = \boldsymbol{c}_0 + \boldsymbol{C}\boldsymbol{k} \tag{31}$$

$$\Leftrightarrow (\boldsymbol{I} - \boldsymbol{C})\boldsymbol{k} = \boldsymbol{c}_0 \quad s.t.\ \boldsymbol{k} \succcurlyeq \boldsymbol{0} \tag{32}$$

$$\Leftrightarrow (1 - c_{i,i})k_i - \sum_{\substack{j=1 \\ j\neq i}}^d (c_{i,j}k_j) = c_{i,0} \quad s.t.\ k_i \geq 0\ \forall i \in [1,d] \tag{33}$$

Intuitively, the equivalence of boxes from Eq. (30) is equivalent to the equality of their diameters which yields Eq. (31). Therefore, any $\boldsymbol{k} \in \mathbb{R}^d$ satisfying Eq. (31) gives us a fixed point $\boldsymbol{w}_N^{\sharp\,*}$. In Eq. (32) we make explicit our assumption that $k_i \geq 0$ and finally in Eq. (33) we write Eq. (32) as a system of linear equations.

---

[2]This does not restrict the possible concretization of any such zonotope we can achieve. As $\boldsymbol{A}$ is invertible, $\boldsymbol{A}^{-1}$ has full rank and thus, its column vectors $\boldsymbol{a}_1, \cdots, \boldsymbol{a}_d$ form a basis of a vector space. Given $\boldsymbol{A}^{-1}$, the concretization of a zonotope of this form is uniquely determined by $\boldsymbol{k}$ (recall that we assume that all $k_i$ are positive).

**Existence of a Closed Form Solution for the Fixed Point Equations for $w_N^{\sharp\,*}$** What remains to be shown to prove Thm. I.1 is that Eq. (33) is guaranteed to have a solution. For that we will investigate the structure of $C$ and $c_0$. Recall that the matrix $C$ and $c_0$ are obtained from IH order reduction on input (see Eq. (29))

$$A(w_N^\sharp - \eta(\mathcal{G}_L^N + \mathbf{L}(\mathcal{G}_H))).$$

$w_N^{\sharp\,*}$ and $\mathcal{G}_L^N$ share the same set of symbols, while they do not share any error symbols with $\mathbf{L}(\mathcal{G}_H)$, as linearization always generates new error symbols. This implies that the terms of $A(w_N^\sharp - \eta\mathcal{G}_L^N)$ and $-\eta A\mathbf{L}(\mathcal{G}_H)$ will not cancel out. In other words, they contribute separately to the diameter $k$ of the merged box produced by IH. Therefore, we will look into their contributions separately.

For $A(w_N^\sharp - \eta\mathcal{G}_L^N)$, we substitute the definitions of $\mathcal{G}_L^N$ and $w_N^{\sharp\,*}$ to get:

$$A(w_N^\sharp - \eta\mathcal{G}_L^N) = A\Big((1-2\eta\lambda)I - \frac{2\eta}{n}X_R{}^T X_R\Big)w_N^\sharp \tag{34}$$

$$= A\left((1-2\eta\lambda)I - \frac{2\eta}{n}X_R{}^T X_R\right)A^{-1}\begin{bmatrix}k_1\epsilon_1\\ \vdots\\ k_d\epsilon_d\end{bmatrix} \qquad (\text{expand } w_N^\sharp)$$

$$= A\left(A(1-2\eta\lambda)A^{-1} - \frac{2\eta}{n}AX_R{}^T X_R A^{-1}\right)A^{-1}\begin{bmatrix}k_1\epsilon_1\\ \vdots\\ k_d\epsilon_d\end{bmatrix}$$

$$(I = AA^{-1} \text{ and } X_R{}^T X_R = AX_R{}^T X_R A^{-1})$$

$$= \left(A(1-2\eta\lambda)A^{-1} - \frac{2\eta}{n}AX_R{}^T X_R A^{-1}\right)\begin{bmatrix}k_1\epsilon_1\\ \vdots\\ k_d\epsilon_d\end{bmatrix} \tag{35}$$

This contributes $\left(|1 - 2\eta\lambda - \frac{2\eta}{n}q_{i,i}|k_i + \frac{2\eta}{n}\sum_{\substack{j=1\\j\neq i}}^d |q_{i,j}|k_j\right)$ to the diameter of the merged box along dimension $i$, where $q_{i,j}$ is the element in $AX_R{}^T X_R A^{-1}$ at position $(i,j)$. By choosing a small learning rate $\eta$ to let $1 - 2\eta\lambda - \frac{2\eta}{n}q_{i,i} \geq 0$ the contribution is equal to $\Big((1 - 2\eta\lambda - \frac{2\eta}{n}q_{i,i})k_i + \frac{2\eta}{n}\sum_{\substack{j=1\\j\neq i}}^d |q_{i,j}|k_j\Big)$. From Eq. (34) it is obvious that this component has a shared term $w_N^{\sharp\,*}$ containing $k$ and as $w_N^{\sharp\,*}$ is centered at $0$, we know that this component has only symbolic terms. This in turn implies that this component only contributes to the coefficient matrix $C$ and does not contribute to the constant part $c_0$ of the system of linear equations.

Now consider the other component

$$-\eta A\mathbf{L}(\mathcal{G}_H) \tag{36}$$

$$= -\eta A\mathbf{L}\left(\frac{2}{n}\Big((X_R{}^T X_S^\sharp + X_S^\sharp{}^T X_R)(w_D^\sharp + w_N^\sharp) + X_S^\sharp{}^T X_S^\sharp(w_R + w_D^\sharp + w_N^\sharp) - X_S^\sharp{}^T y_S^\sharp\Big)\right)$$

$$= -\eta A\mathbf{L}\Bigg(\frac{2}{n}\Big(\underbrace{(X_R{}^T X_S^\sharp + X_S^\sharp{}^T X_R + X_S^\sharp{}^T X_S^\sharp)w_N^\sharp}_{\text{contributes to } C}$$

$$+ \underbrace{(X_R{}^T X_S^\sharp + X_S^\sharp{}^T X_R)w_D^\sharp + X_S^\sharp{}^T X_S^\sharp(w_R + w_D^\sharp) - X_S^\sharp{}^T y_S^\sharp}_{\text{contributes to } c_0}\Big)\Bigg) \tag{37}$$

Note that the first part of the equation is the only part that contains $w_N^\sharp$ which in turn contains the variables $k$ for which we want to solve. As every element of this part of the equation contains some $k_i$ it only contributes to $C$. In contrast the second part of the equation does not contain any variable $k_i$ and, thus, is the only part of both components that contribute to $c_0$. Let us use $\frac{2\eta}{n}C'k$ to denote the matrix of coefficients that is the sum of absolute values of coefficients of

$\frac{2}{n}(\boldsymbol{X_R}^T \boldsymbol{X_S^\sharp} + \boldsymbol{X_S^\sharp}^T \boldsymbol{X_R} + \boldsymbol{X_S^\sharp}^T \boldsymbol{X_S^\sharp})\boldsymbol{w_N^\sharp}$ that will be projected by $\boldsymbol{A}$ stemming from the first part of the equation in the expanded version of $\mathbf{L}(\mathcal{G}_H)$. We use $c'_{i,j} \geq 0$ to denote the element of $\boldsymbol{C'}$ at $(i,j)$. As mentioned before this is the only part of this component that contains terms containing unknowns $k_i$. We use $\boldsymbol{c'_0}$ to denote the contribution from the second part of Eq. (37) towards the output of linearization $\mathbf{L}$. Then based on the fact that the second part of this component is the only part of both components that contributes to $\boldsymbol{c}_0$, we know that the whole contribution of the component from Eq. (36) towards $\boldsymbol{C} + \boldsymbol{c}_0$ is:

$$\eta \boldsymbol{c'_0} + \frac{2\eta}{n} \boldsymbol{C'} \boldsymbol{k} \tag{38}$$

In summary, the system of linear equations from Eq. (33) can be written as:

$$\left(2\eta\lambda + \frac{2\eta}{n} q_{i,i} - \frac{2\eta}{n} c'_{i,i}\right) k_i - \frac{2\eta}{n} \sum_{\substack{j=1 \\ j \neq i}}^{d} (|q_{i,j}| + c'_{i,j}) k_j = \eta c_{i,0} \quad s.t.\ k_i \geq 0 \quad \forall i \in [1, d]$$

$$\Leftrightarrow (\lambda n + q_{i,i} - c'_{i,i}) k_i - \sum_{\substack{j=1 \\ j \neq i}}^{d} (|q_{i,j}| + c'_{i,j}) k_j = \frac{n}{2} c_{i,0} \quad s.t.\ k_i \geq 0 \quad \forall i \in [1, d]. \tag{39}$$

In the expanded system of linear equations, the diagonal elements of the coefficient matrix are $(\lambda n + q_{i,i} - c'_{i,i})$, while the off-diagonal elements are $-(|q_{i,j}| + c'_{i,j})$.

**Lemma I.6.** *If for all $i \in [1, d]$,*

$$(\lambda n + q_{i,i} - c'_{i,i}) \geq \sum_{\substack{j=1 \\ j \neq i}}^{d} (|q_{i,j}| + c'_{i,j}),$$

*then the system of linear equations from Eq. (39) has a solution.*

*Proof.* When this condition holds, the coefficient matrix of this system of linear equations is diagonally dominant. Also considering all diagonal elements are non-negative ($(\lambda n + q_{i,i} - c'_{i,i}) \geq \sum_{\substack{j=1 \\ j \neq i}}^{d} (|q_{i,j}| + c'_{i,j}) \geq 0$) and all off-diagonal elements are non-positive, the coefficient matrix is an M-matrix[3]. For any M-matrix $\boldsymbol{M}$, the inverse $\boldsymbol{M}^{-1}$ exists and is has all non-negative entries. Furthermore, for any system of linear equations represented as matrix equation $\boldsymbol{M}\boldsymbol{x} = \boldsymbol{b}$ where $\boldsymbol{x}$ are the variables of the equation system, $\boldsymbol{M}^{-1}\boldsymbol{b}$ is a solution. As our matrix $I - \boldsymbol{C}$ is an M-matrix, we know that its inverse exists and has only positive entries and we can compute a solution $\boldsymbol{k}$ as:

$$\boldsymbol{k} = (I - \boldsymbol{C})^{-1} \boldsymbol{c}_0$$

Given that all elements $c_{i,0}$ are positive and that $(I - \boldsymbol{C})^{-1}$ is positive, the solution $\boldsymbol{k}$ for Eq. (39) is also positive. $\qquad\square$

The correctness of Thm. I.1 follows immediately based on the sufficient condition for the existence of solution stated in Lem. I.6, .

**Corollary I.7** (Proof of Thm. I.1). *Given an input abstract training dataset $\mathbf{D}^\sharp$, let $\beta = \frac{1}{n} \max_i \left( \sum_{\substack{j=1 \\ j \neq i}}^{d} (|q_{i,j}| + c'_{i,j}) + c'_{i,i} - q_{i,i} \right)$ For any regularization coefficient $\lambda \geq \beta$, we can compute an abstract fixed point $\boldsymbol{w}^{\sharp *}$ for the model weights according to Def. 3.2.*

*Proof.* To find an abstract fixed point of the model weights

$$\boldsymbol{w}^{\sharp *} = \boldsymbol{w}_R^* + \boldsymbol{w}_D^{\sharp *} + \boldsymbol{w}_N^{\sharp *}$$

---

[3]There are over 40 sufficient conditions for a matrix to be an M-matrix [56]. The one we use here requires that the matrix fulfills two conditions: (1) all off-diagonal elements are non-positive, and (2) each diagonal element is no less than the sum of absolute values of the off-diagonal elements at the same row (also known as diagonally dominant).

that fulfills the sufficient condition for an abstract fixed point from Prop. I.2, we first use the closed form solutions from Lem. I.4 to compute $\boldsymbol{w}_R^*$ and then using $\boldsymbol{w}_R^*$ compute $\boldsymbol{w}_D^{\sharp *}$. These are guaranteed to fulfill the first two conditions of Prop. I.2. As the regularization coefficient $\lambda \geq \beta = \frac{1}{n} \max_i \left( \sum_{\substack{j=1 \\ j \neq i}}^d (|q_{i,j}| + c'_{i,j}) + c'_{i,i} - q_{i,i} \right)$, the condition from Lem. I.6 holds. Thus, the linear system of equations for a $\boldsymbol{w}_N^{\sharp *}$ that fulfills the last condition of Prop. I.2 has a solution that we can compute using any techniques for solving a system of linear equation. It follows that $\boldsymbol{w}^{\sharp *} = \boldsymbol{w}_R^* + \boldsymbol{w}_D^{\sharp *} + \boldsymbol{w}_N^{\sharp *}$ fulfills all conditions of Prop. I.2 and is an abstract fixed point according to Def. 3.2. □

## J   Weakening the Requirements on Regularization

The existence of a fixed point $\boldsymbol{w}^{\sharp *}$ as postulated in Thm. I.1 only holds for sufficiently large regularization $\lambda \geq \beta$ where $\beta$ depends on $\mathbf{D}^\sharp$. The majority of real world examples without multicolinearity in the real part of the training data that we have investigated have $\beta = 0$. However, we still provide a technique to reduce $\beta$ if the need should arise at the cost of increasing the runtime of our approach. Specifically, we will make use of *splitting* [2] which divides an input zonotope into multiple zonotopes of smaller expand. We apply splitting to $\mathbf{D}^\sharp$, then compute the fixed point for each zonotope in the result of splitting, and finally combine these fixed point using a *join* operation for zonotopes that over-approximates the union of the concretizations of two zonotopes (allows us to merge multiple zonotope such that the merged zonotope has a concretization that over-approximates the concretizations of all of the inputs). We will demonstrate that this approach enables us to reduce $\beta$ to any value $\geq 0$. Intuitively, this works because we can reduce the extend of the input zonotope for the fixed point computation to the extend necessary to ensure $\beta = 0$. Albeit in worst-case this can require a splitting operator that generates a number of zonotopes that is exponential in the dimension.

### J.1   Splitting and Join for zonotopes

Intuitively, a splitting operator divides an input zonotope into two or more zonotopes such that the union of their concretization is the same as the concretization of the input zonotope.

**Definition J.1** (Split Operator). *An operator $\mathbf{S}$ that maps a zonotope $\ell^\sharp$ to a set of zonotopes $\{\ell_1^\sharp, \cdots, \ell_m^\sharp\}$ is for any input zonotope $\ell^\sharp$ we have:*

$$\mathbf{S}(\ell^\sharp) = \{\ell_1^\sharp, \cdots, \ell_m^\sharp\} \quad s.t. \bigcup_i \gamma\left(\ell_i^\sharp\right) = \gamma\left(\ell^\sharp\right).$$

As an example of a split operator consider the binary split (2-split) at dimension $i$ denoted by $\mathbf{S}_{2,i}$ that splits the input zonotope into two parts by scaling the $i^{th}$ generator $\boldsymbol{g}_i$ by $\frac{1}{2}$ and shifting the center of the zonotope by $\frac{1}{2}\boldsymbol{g}_i$ ($-\frac{1}{2}\boldsymbol{g}_i$):

$$\mathbf{S}_{2,i}(\ell^\sharp) = \left\{\ell_1^\sharp, \ell_2^\sharp\right\} = \left\{\boldsymbol{c} + \underbrace{\sum_j \boldsymbol{g}_j \epsilon_j}_{\substack{\epsilon_i \in [-1,0] \\ \text{other } \epsilon_j \in [-1,1]}}, \boldsymbol{c} + \underbrace{\sum_j \boldsymbol{g}_j \epsilon_j}_{\substack{\epsilon_i \in [0,1] \\ \text{other } \epsilon_j \in [-1,1]}}\right\}$$

$$= \left\{\boldsymbol{c} - \frac{1}{2}\boldsymbol{g}_i + \frac{1}{2}\boldsymbol{g}_i\epsilon_i + \sum_{j \neq i} \boldsymbol{g}_j\epsilon_j, \quad \boldsymbol{c} + \frac{1}{2}\boldsymbol{g}_i + \frac{1}{2}\boldsymbol{g}_i\epsilon_i + \sum_{j \neq i} \boldsymbol{g}_j\epsilon_j\right\} \quad (40)$$

As $\gamma\left(\ell_1^\sharp\right) \cup \gamma\left(\ell_2^\sharp\right) = \gamma\left(\ell^\sharp\right)$, $\mathbf{S}_{2,i}(\cdot)$ is a split operator according to Def. J.1. We can generalize 2-split to an $(m, i)$-split $\mathbf{S}_{m,i}$ that divides the zonotope evenly into $m$ parts along the $i^{th}$ dimension. We will call the composition of $(m, i)$-splits across all dimensions as a $\mu$-split. Given $\mu = \frac{1}{m}$ for $m \in \mathbb{N}$ as input, the effect of the $\mu$-split $\mathbf{S}_\mu$ is the generators of each zonotope in the result of splitting are the generators of the input zonotope scaled by $\mu$. Thus, $\mu$-splitting allows us to downscale the generators of a zonotope. For a $d$-dimensional zonotope and $\mu = \frac{1}{m}$, $\mathbf{S}_\mu$ returns $d^m$ zonotopes.

**Definition J.2** ($\mu$-split). *For $d$-dimensional zonotopes and $\mu = \frac{1}{m}$ for $m \in \mathbb{N}$, the $\mu$-split $\mathbf{S}_\mu$ is defined as:*

$$\mathbf{S}_{m,d} \circ \ldots \circ \mathbf{S}_{m,1}$$

*where the $(m, i)$-split $\mathbf{S}_{m,i}$ is defined as shown below and the application of a split operator to a set of zonotopes is defined as applying the split operator to every element in the set.*

$$\mathbf{S}_{m,i}(\boldsymbol{\ell}^\sharp) = \{\boldsymbol{\ell}^\sharp{}_i\}_{j=1}^m$$

*for*

$$\boldsymbol{\ell}^\sharp{}_j = \boldsymbol{c} + \frac{-m + 2j - 1}{m}\boldsymbol{g}_i + \frac{1}{m}\boldsymbol{g}_i\epsilon_i + \sum_{j \neq i}\boldsymbol{g}_j\epsilon_j$$

For example, $\mathbf{S}_2(1 + \epsilon_1) = \{0.5 + 0.5\epsilon_1, 1.5 + 0.5\epsilon_1\}$. It is easy to see that $\mathbf{S}_\mu$ is indeed a split according to Def. J.1.

**Proposition J.3** ($\mathbf{S}_\mu$ is a Split Operator). $\mathbf{S}_\mu$ *is a split operator for any* $\mu = \frac{1}{m}$ *were* $m \in \mathbb{N}$.

Next we introduce join operators for zonotopes and then demonstrate that for any abstract transformer $F^\sharp$ for a function $F$, we can construct another abstract transformer for $F$ by (i) splitting the input zonotope using some split operator $\mathbf{S}$, (ii) applying $F^\sharp$ separately on each zonotope returned by the split, and (iii) merge the results using a join that we introduce next.[4]

**Definition J.4** (Join). *An operator $\sqcup$ is a join for zonotopes if it over-approximates union of concretization. That is, for any set $S^\sharp$ of zonotopes:*

$$\gamma\left(\bigsqcup S^\sharp\right) \supseteq \bigcup_{\boldsymbol{z}^\sharp \in S^\sharp} \gamma\left(\boldsymbol{z}^\sharp\right).$$

Several join operators have been proposed for zonotopes, e.g., [24]. Next we establish that new abstract transformers can be constructed by wrapping existing transformers with split and join. This ensures that our approach of combining our abstract fixed point calculation with splitting yields valid fixed points.

**Lemma J.5** (Abstract Transformers Are Sound on Splits). *Consider a zonotope $\boldsymbol{z}^\sharp$ and let $F^\sharp$ be an abstract transformer for a function $F$. Furthermore, let $\sqcup$ be a join operator, $\mathbf{S}$ be a split operator, and let $\mathbf{S}(\boldsymbol{\ell}^\sharp) = \{\boldsymbol{\ell}^\sharp_1, \ldots, \boldsymbol{\ell}^\sharp_m\}$. Then, $F_{\mathbf{S},\sqcup}{}^\sharp = \sqcup \circ F^\sharp \circ \mathbf{S}$ is an abstract transformer for $F$.*

*Proof.* We have to show that

$$\gamma\left(F_{\mathbf{S},\sqcup}{}^\sharp(\boldsymbol{\ell}^\sharp)\right) \supseteq F(\gamma\left(\boldsymbol{\ell}^\sharp\right)).$$

We have

$$
\begin{aligned}
& F(\gamma\left(\boldsymbol{\ell}^\sharp\right)) \\
=& F(\bigcup_{\boldsymbol{\ell}^\sharp{}' \in \mathbf{S}(\boldsymbol{\ell}^\sharp)} \gamma\left(\boldsymbol{\ell}^{\sharp'}\right)) && \text{(Def. J.1 for } \mathbf{S}) \\
=& F(\bigcup_{i=1}^m \gamma\left(\boldsymbol{\ell}^\sharp{}_i\right)) \\
=& \bigcup_{i=1}^m F(\gamma\left(\boldsymbol{\ell}^\sharp{}_i\right)) && \text{(function application on sets of concrete elements is point-wise application)} \\
\subseteq& \bigcup_{i=1}^m \gamma\left(F^\sharp(\boldsymbol{\ell}^\sharp{}_i)\right) && (F^\sharp \text{ is an abstract transformer}) \\
\subseteq& \gamma\left(\bigsqcup_{i=1}^m F^\sharp(\boldsymbol{\ell}^\sharp{}_i)\right) && (\gamma\left(\boldsymbol{z}_1^\sharp \sqcup \boldsymbol{z}_2^\sharp\right) \supseteq \gamma\left(\boldsymbol{z}_1^\sharp\right) \cup \gamma\left(\boldsymbol{z}_2^\sharp\right)) \\
=& \gamma\left(F_{\mathbf{S},\sqcup}{}^\sharp(\boldsymbol{\ell}^\sharp)\right)
\end{aligned}
$$

$\square$

---

[4]Technically, join is normally defined as an over-approximation of the least upper bound of two abstract elements wrt. a partial order of the abstract domain. We order abstract elements based on set inclusion of their concretizations here and only define join regarding to this partial order.

## J.2 Weakening Regularization Requirements

Equipped with the $\mu$-split and a join operator, we are ready to analyze how the introduction of a $\mu$-split can be used to achieve an arbitrarily small $\beta$. Given the input data $\mathbf{D}^\sharp$, the learning rate $\eta \geq \beta$, the regularization coefficient $\lambda$, the transformation matrix $\boldsymbol{A}$, we use $\Gamma(\mathbf{D}^\sharp, \eta, \lambda, \boldsymbol{A})$ to denote the function that computes the fixed point $\boldsymbol{w}_S^{\sharp *}$ for these inputs using the process outlined in Corollary I.7. From Lem. J.5 follows that $\Gamma_{\mathbf{S}_\mu, \sqcup}$ is an abstract transformer for gradient descent and, thus, according to Prop. 3.3 over-approximates $\boldsymbol{w}^{\odot *}$

Next, we show that given an abstract training dataset $\mathbf{D}^\sharp$ and a desired regularization coefficient $\lambda_{target}$, we can find a value of $\mu$ such that the abstract fixed point construction with $\mu$-splitting computes a fixed point. Recall that $\Gamma$ can construct a fixed point if $\lambda \geq \beta$ where $\beta$ is a constant that depends on $\mathbf{D}^\sharp$. We will show that by choosing $\mu$ carefully, we can achieve $\beta \leq \lambda_{target}$ for each zonotope in the result of the split and, thus, $\Gamma$ will return a valid fixed point for each zonotope in the split result.

**Lemma J.6.** *For any abstract training dataset $\mathbf{D}^\sharp$ and desired regularization coefficient $\lambda$, there exists $m \in \mathbb{N}$ such that for $\mu = \frac{1}{m}$, $\Gamma_{\mathbf{S}_\mu, \sqcup}$ returns a fixed point $\boldsymbol{w}^{\sharp *}$.*

*Proof.* Consider an abstract training dataset $\mathbf{D}^\sharp = (\boldsymbol{X}^\sharp, \boldsymbol{y}^\sharp)$ with $\boldsymbol{X}^\sharp = \boldsymbol{X}_R + \boldsymbol{X}_S^\sharp$. WLOG consider a single abstract training dataset $\mathbf{D}^\sharp_i$ in the result of $\mu$-split on $\mathbf{D}^\sharp$:

$$\mathbf{D}^\sharp_i \in \mathbf{S}_\mu(\mathbf{D}^\sharp)$$

We will show that using the regularization coefficient $\lambda$, we can find $\mu$ such that the precondition $\lambda \geq \beta$ for $\Gamma$ to compute a fixed point holds. Then applying Lem. J.5 we get the desired result.

First observe that since $\mathbf{D}^\sharp_i$ is in the result of $\mu$-splitting, we know that it has the following shape where the symbolic component $\boldsymbol{X}_S^\sharp$ is scaled by $\mu$ and the real component $\boldsymbol{X}_{Ri}$ of $\mathbf{D}^\sharp_i$ typically differs from $\boldsymbol{X}^\sharp$ as $\mu$-splitting changes the center of the zonotope.

$$\boldsymbol{X}^\sharp_i = \boldsymbol{X}_{Ri} + \mu \boldsymbol{X}_S^\sharp$$

Recall that $\frac{2\eta}{n} \boldsymbol{C}' \boldsymbol{k}$ denotes the matrix of coefficients that is the sum of absolute values of coefficients of $\frac{2}{n} \boldsymbol{A}(\boldsymbol{X}_R^T \boldsymbol{X}_S^\sharp + \boldsymbol{X}_S^{\sharp T} \boldsymbol{X}_R + \boldsymbol{X}_S^{\sharp T} \boldsymbol{X}_S^\sharp) \boldsymbol{w}_N^\sharp$. Plugging in the definition of $\boldsymbol{X}^\sharp_i$ from above, we get a matrix $\frac{2\eta}{n} \boldsymbol{C}' \boldsymbol{k}$ that is the sum of absolute values of coefficients of

$$\frac{2}{n} \boldsymbol{A}(\boldsymbol{X}_{Ri}^T \mu \boldsymbol{X}_S^\sharp + \mu \boldsymbol{X}_S^{\sharp T} \boldsymbol{X}_{Ri} + \mu \boldsymbol{X}_S^{\sharp T} \mu \boldsymbol{X}_S^\sharp) \boldsymbol{w}_N^\sharp \tag{41}$$

$$= \frac{2}{n} \boldsymbol{A}(\mu \boldsymbol{X}_{Ri}^T \boldsymbol{X}_S^\sharp + \mu \boldsymbol{X}_S^{\sharp T} \boldsymbol{X}_{Ri} + \mu^2 \boldsymbol{X}_S^{\sharp T} \boldsymbol{X}_S^\sharp) \boldsymbol{A}^{-1} \begin{bmatrix} k_1 \epsilon_1 \\ \vdots \\ k_d \epsilon_d \end{bmatrix} \tag{42}$$

$$= \Big( \underbrace{\frac{2\mu}{n} \boldsymbol{A}(\boldsymbol{X}_{Ri}^T \boldsymbol{X}_S^\sharp + \boldsymbol{X}_S^{\sharp T} \boldsymbol{X}_{Ri}) \boldsymbol{A}^{-1}}_{\text{contributes to } c_{i,j}^1} + \underbrace{\frac{2\mu^2}{n} \boldsymbol{A}(\boldsymbol{X}_S^{\sharp T} \boldsymbol{X}_S^\sharp) \boldsymbol{A}^{-1}}_{\text{contributes to } c_{i,j}^2} \Big) \begin{bmatrix} k_1 \epsilon_1 \\ \vdots \\ k_d \epsilon_d \end{bmatrix}. \tag{43}$$

Recall that $c'_{i,j}$ denotes entry of $\boldsymbol{C}'$ at row $i$ and column $j$. Here, $\boldsymbol{X}_S^{\sharp T} \boldsymbol{X}_S^\sharp$ has higher order than $(\boldsymbol{X}_R^T \boldsymbol{X}_S^\sharp + \boldsymbol{X}_S^{\sharp T} \boldsymbol{X}_R)$, thus their terms will not cancel out. Therefore, we can decompose the contribution to $c'_{i,j}$ into the above two parts, denoted as

$$c'_{i,j} = c_{i,j}^1 + c_{i,j}^2.$$

Given scalar $\mu$, let

$$c'_\mu = \max_{u,v} c'_{u,v}$$

i.e., $c'_\mu$ denotes the largest entry $c'_{i,j}$ in $\mathbf{D}^\sharp_i$.

Let $i$ and $j$ such that $c'_\mu = c'_{i,j}$, we have:

$$c'_\mu = \mu c^1_{i,j} + \mu^2 c^2_{i,j} \leq \mu c^1_{i,j} + \mu c^2_{i,j} = \mu c'_{max},$$

where $c'_{max}$ is a constant that is equal to the maximum $c'_{i,j}$ before scaling $\boldsymbol{X}^\sharp_S$ through $\mu$-splitting.

Next, we prove the lemma for a specific transformation $\boldsymbol{A} = V^T$ during order reduction derived through singular value decomposition (SVD). Using SVD, we can decompose the covariance matrix $\boldsymbol{X}_{Ri}^T \boldsymbol{X}_{Ri}$ of the real number part of features $\boldsymbol{X}_{Ri}$ as shown below

$$\boldsymbol{X}_{Ri}^T \boldsymbol{X}_{Ri} = V\Sigma V^T.$$

For $\boldsymbol{A} = V^T$, and using the fact that $V$ is a rotation in SVD which implies $V^{-1} = V^T$, we get:

$$\boldsymbol{A}\boldsymbol{X}_{Ri}^T \boldsymbol{X}_{Ri} \boldsymbol{A}^{-1}$$
$$=\boldsymbol{A}(V\Sigma V^T)\boldsymbol{A}^{-1}$$
$$=V^T(V\Sigma V^T)V^{T^{-1}}$$
$$=(V^{-1}V)\Sigma(V^{-1}V^{-1^{-1}})$$
$$=\Sigma$$

Thus, the matrix $\boldsymbol{A}\boldsymbol{X}_{Ri}^T \boldsymbol{X}_{Ri} \boldsymbol{A}^{-1} = \Sigma$ is a diagonal matrix, meaning that all off-diagonal elements of it ($|q_{i,j}|$ for all $(i \neq j)$ in Corollary I.7) are zero. In addition, the diagonal element $\Sigma[i,i]$ ($q_{i,i}$ in Corollary I.7) is the $i^{th}$ eigenvalue of the covariance matrix $\boldsymbol{X}_{Ri}^T \boldsymbol{X}_{Ri}$, which must be positive when assuming no multicolinearity.

To sum up, $\beta$ from Corollary I.7 can be re-written as:

$$\beta = \frac{1}{n} \max_i \left( c'_{i,i} + \sum_{\substack{j=1 \\ j \neq i}}^{d} (|q_{i,j}| + c'_{i,j}) - q_{i,i} \right) \tag{44}$$

$$\leq \frac{1}{n} \max_i \left( c'_\mu + \sum_{\substack{j=1 \\ j \neq i}}^{d} c'_\mu - q_{i,i} \right) \tag{45}$$

$$\leq \frac{1}{n} \max_i \left( \mu c'_{max} + \sum_{\substack{j=1 \\ j \neq i}}^{d} \mu c'_{max} - q_{i,i} \right) \tag{46}$$

$$= \frac{1}{n} \max_i \left( d\mu c'_{max} - q_{i,i} \right) = \frac{1}{n} d\mu c'_{max} - \frac{1}{n} \min_i (q_{i,i}) \tag{47}$$

where $\frac{1}{n} \min_i(q_{i,i})$ is a positive constant. Therefore, when setting $\mu = \frac{\min_i(q_{i,i})}{dc'_{max}} > 0$, we have $\beta \leq \frac{1}{n} d\mu c'_{max} - \frac{1}{n} \min_i(q_{i,i}) = 0$. $\qquad\square$

## K   An Approximate Abstract Transformer for Ridge Regression

Alg. 2 is the detailed version of Alg. 1 we presented in the main paper. Like Alg. 1, Alg. 2 takes as input an abstract dataset $\mathbf{D}^\sharp$ that over-approximates $\boldsymbol{D}^\odot$, a learning rate $\eta$, a regularization coefficient $\lambda$, and a transformation matrix $\boldsymbol{A}$ used for order reduction (e.g., the SVD-based transformation discussed in App. J.2). In Line 1, we first use function constructFPEquations to compute fixed points for $\boldsymbol{w}_R^*$ and $\boldsymbol{w}_D^\sharp$ using the closed form solution from Lem. I.4 and construct the equation system $\Xi$ for $\boldsymbol{w}_N^{\sharp*}$ (Eq. (39)). Furthermore, this function also computes the threshold $\beta$ on the regularization coefficient for this system of equations to have a solution. If $\beta$ is smaller or equal to the desired regularization coefficient $\lambda$ (Line 10), then we solve the linear equations $\Xi$ (Line 11) and then returns the fixpoint $\boldsymbol{w}^{\sharp*}$. Otherwise, we determine a sufficiently small splitting factor $\mu$ that ensures that $\beta \leq \lambda$ for each $\boldsymbol{w}^\sharp_i$ in the result of the split (Line 3), compute the fixed point for each such $\boldsymbol{w}^\sharp_i$ (Lines 6 and 8), and merge these fixed points using join to compute the final result (Line 9).

**Algorithm 2:** Constructing a Fixed Point for Abstract Gradient Descent

---

**Input:** abstract dataset $\mathbf{D}^\sharp = (\boldsymbol{X}^\sharp, \boldsymbol{y}^\sharp)$, learning rate $\eta$, Regularization coefficient $\lambda$, transformation matrix $\boldsymbol{A}$

**Output:** abstract fixed point $\boldsymbol{w}^{\sharp *}$

1   $\boldsymbol{w}_R^*, \boldsymbol{w}_D^{\sharp *}, \Xi, \beta \leftarrow \texttt{constructFPequations}(\mathbf{D}^\sharp, \lambda, \eta)$

2   **if** $\beta > \lambda$ **then**                        `// Is splitting necessary?`

3      $\mu \leftarrow \texttt{determineNumSplits}(\Xi, \lambda)$

4      $\{\mathbf{D}^\sharp{}_i\}_{i=1}^m \leftarrow \mathbf{S}_\mu(\mathbf{D}^\sharp)$                    `// Apply` $\mu$`-splitting`

5      **for** $\mathbf{D}^\sharp{}_i \in \{\mathbf{D}^\sharp{}_i\}_{i=1}^m$ **do**         `// Construct FP for each` $\mathbf{D}^\sharp{}_i$ `in the split`

6          $\boldsymbol{w}_R^*, \boldsymbol{w}_D^{\sharp *}, \Xi, \beta \leftarrow \texttt{constructFPequations}(\mathbf{D}^\sharp{}_i, \lambda, \eta)$

7          $\boldsymbol{w}_N^{\sharp *} \leftarrow \texttt{solveFixedPointEquations}(\Xi, \lambda, \eta)$

8          $\boldsymbol{w}^\sharp{}_i \leftarrow \boldsymbol{w}_R^* + \boldsymbol{w}_D^{\sharp *} + \boldsymbol{w}_N^{\sharp *}$

9      **return** $\bigsqcup_{i=1}^m \boldsymbol{w}^\sharp{}_i$                      `// Merge FPs using Join`

10   **else**                                   `// No splitting required`

11      $\boldsymbol{w}_N^{\sharp *} \leftarrow \texttt{solveFixedPointEquations}(\Xi, \lambda, \eta)$

12      $\boldsymbol{w}^{\sharp *} \leftarrow \boldsymbol{w}_R^* + \boldsymbol{w}_D^{\sharp *} + \boldsymbol{w}_N^{\sharp *}$

13   **return** $\boldsymbol{w}^{\sharp *}$

14   **def** $\texttt{constructFPEquations}(\mathbf{D}^\sharp, \lambda, \eta)$**:**

15      $\boldsymbol{w}_R^* \leftarrow \texttt{evalClosedFormReal}(\mathbf{D}^\sharp, \lambda)$                     `// Eq. (22)`

16      $\boldsymbol{w}_D^{\sharp *} \leftarrow \texttt{evalClosedFormSymbolic}(\mathbf{D}^\sharp, \lambda, \boldsymbol{w}_R^*)$         `// Eq. (23)`

17      $\Xi \leftarrow \texttt{constructNonDataFixedPointEquations}(\mathbf{D}^\sharp, \lambda, \eta, \boldsymbol{w}_R^*, \boldsymbol{w}_D^{\sharp *})$     `// Eq. (39)`

18      $\beta \leftarrow \texttt{determineMinRegularization}(\Xi)$

19      **return** $\boldsymbol{w}_R^*, \boldsymbol{w}_D^{\sharp *}, \Xi, \beta$

---

# L   Infeasibility of Symbolically Evaluating the Closed-form Solution for Ridge Regression

In this section, we demonstrate why evaluating the closed-form solution for linear regression with MSE is not feasible. It produces symbolic expressions with many variables and large symbolic terms (their representation size) that include the fractions with denominators and enumerators that are polynomial expressions. Apart from the size of these expressions, computing viable prediction intervals from such a symbolic representation of model weights is computationally hard. We use MSE loss for simplicity, but the same arguments also apply to ridge regression.

We illustrate the issues using a randomly-generated toy dataset with 10 samples and 3 uncertain data cells in the third column (corresponding to 3 error symbols $\epsilon_0, \epsilon_1, \epsilon_2$)[5]:

$$\boldsymbol{X}^\sharp = \begin{bmatrix} 1.0 & 1.6 & 1.3\epsilon_0 + 1.4 \\ 1.0 & 0.8 & 1.5 \\ 1.0 & -1.7 & -1.9 \\ 1.0 & -0.57 & 1.1 \\ 1.0 & -0.39 & 0.36 \\ 1.0 & 0.035 & 1.2 \\ 1.0 & -0.34 & -0.73 \\ 1.0 & 0.038 & 0.3\epsilon_1 + 0.44 \\ 1.0 & 1.5 & 1.2\epsilon_2 + 1.3 \\ 1.0 & -0.98 & -0.66 \end{bmatrix} \qquad \boldsymbol{y}^\sharp = \begin{bmatrix} 4.6 \\ -1.5 \\ 0.39 \\ -5.7 \\ -3.0 \\ -3.6 \\ -0.44 \\ 0.46 \\ 2.2 \\ -3.0 \end{bmatrix}$$

---

[5]The first column of $\boldsymbol{X}^\sharp$ consisting of 1's corresponds to the bias term or the intercept term in the weights.

Then, symbolically evaluating the closed-form solution for linear regression with MSE loss, we obtain the symbolic expression representing all possible model weights:

$$w^{\sharp *} = (X^{\sharp^T} X^{\sharp})^{-1} X^{\sharp^T} y^{\sharp}$$

$$= \begin{bmatrix} \frac{1.5\cdot10^6\epsilon_0^2+8.7\cdot10^4\epsilon_0\epsilon_1-4.5\cdot10^5\epsilon_0\epsilon_2-1.2\cdot10^6\epsilon_0+9.2\cdot10^4\epsilon_1^2+4.0\cdot10^3\epsilon_1\epsilon_2-2.4\cdot10^5\epsilon_1+1.0\cdot10^6\epsilon_2^2-2.1\cdot10^6\epsilon_2-1.8\cdot10^6}{-1.1\cdot10^6\epsilon_0^2+8.4\cdot10^4\epsilon_0\epsilon_1+1.1\cdot10^6\epsilon_0\epsilon_2+1.1\cdot10^6\epsilon_0-8.3\cdot10^4\epsilon_1^2+7.7\cdot10^4\epsilon_1\epsilon_2+6.1\cdot10^3\epsilon_1-9.9\cdot10^5\epsilon_2^2+9.5\cdot10^5\epsilon_2-4.0\cdot10^6} \\ \frac{-9.5\cdot10^4\epsilon_0^2+2.3\cdot10^4\epsilon_0\epsilon_1+2.6\cdot10^5\epsilon_0\epsilon_2+2.3\cdot10^5\epsilon_0-1.4\cdot10^4\epsilon_1^2+2.0\cdot10^4\epsilon_1\epsilon_2+3.6\cdot10^4\epsilon_1-1.4\cdot10^5\epsilon_2^2-6.0\cdot10^2\epsilon_2-1.8\cdot10^6}{-1.1\cdot10^5\epsilon_0^2+8.4\cdot10^3\epsilon_0\epsilon_1+1.1\cdot10^5\epsilon_0\epsilon_2+1.1\cdot10^5\epsilon_0-8.3\cdot10^3\epsilon_1^2+7.7\cdot10^3\epsilon_1\epsilon_2+6.1\cdot10^2\epsilon_1-9.9\cdot10^4\epsilon_2^2+9.5\cdot10^4\epsilon_2-4.0\cdot10^5} \\ \frac{-3.7\cdot10^3\epsilon_0-4.1\cdot10^2\epsilon_1-9.2\cdot10^2\epsilon_2+1.3\cdot10^4}{-1.1\cdot10^3\epsilon_0^2+84.0\epsilon_0\epsilon_1+1.1\cdot10^3\epsilon_0\epsilon_2+1.1\cdot10^3\epsilon_0-83.0\epsilon_1^2+77.0\epsilon_1\epsilon_2+6.1\epsilon_1-9.9\cdot10^2\epsilon_2^2+9.5\cdot10^2\epsilon_2-4.0\cdot10^3} \end{bmatrix}.$$

Next, we consider one test sample $x_t = [1, -1, 1]$ and apply this closed-form model weight expression to infer its prediction range $V^{\sharp}(x_t)$ (cf. Corollary F.2). With *merely* 3 uncertain data points in the training data, the prediction of this test data point

$$\hat{y_t} = w^{\sharp *} x_t$$

$$= -\frac{3.7\cdot10^3\epsilon_0+4.1\cdot10^2\epsilon_1+9.2\cdot10^2\epsilon_2-1.3\cdot10^4}{-1.1\cdot10^3\epsilon_0^2+84.0\epsilon_0\epsilon_1+1.1\cdot10^3\epsilon_0\epsilon_2+1.1\cdot10^3\epsilon_0-83.0\epsilon_1^2+77.0\epsilon_1\epsilon_2+6.1\epsilon_1-9.9\cdot10^2\epsilon_2^2+9.5\cdot10^2\epsilon_2-4.0\cdot10^3}$$
$$+\frac{9.5\cdot10^4\epsilon_0^2-2.3\cdot10^4\epsilon_0\epsilon_1-2.6\cdot10^5\epsilon_0\epsilon_2-2.3\cdot10^5\epsilon_0+1.4\cdot10^4\epsilon_1^2-2.0\cdot10^4\epsilon_1\epsilon_2-3.6\cdot10^4\epsilon_1+1.4\cdot10^5\epsilon_2^2+6.0\cdot10^2\epsilon_2+1.8\cdot10^6}{-1.1\cdot10^5\epsilon_0^2+8.4\cdot10^3\epsilon_0\epsilon_1+1.1\cdot10^5\epsilon_0\epsilon_2+1.1\cdot10^5\epsilon_0-8.3\cdot10^3\epsilon_1^2+7.7\cdot10^3\epsilon_1\epsilon_2+6.1\cdot10^2\epsilon_1-9.9\cdot10^4\epsilon_2^2+9.5\cdot10^4\epsilon_2-4.0\cdot10^5}$$
$$+\frac{1.5\cdot10^6\epsilon_0^2+8.7\cdot10^4\epsilon_0\epsilon_1-4.5\cdot10^5\epsilon_0\epsilon_2-1.2\cdot10^6\epsilon_0+9.2\cdot10^4\epsilon_1^2+4.0\cdot10^3\epsilon_1\epsilon_2-2.4\cdot10^5\epsilon_1+1.0\cdot10^6\epsilon_2^2-2.1\cdot10^6\epsilon_2-1.8\cdot10^6}{-1.1\cdot10^6\epsilon_0^2+8.4\cdot10^4\epsilon_0\epsilon_1+1.1\cdot10^6\epsilon_0\epsilon_2+1.1\cdot10^6\epsilon_0-8.3\cdot10^4\epsilon_1^2+7.7\cdot10^4\epsilon_1\epsilon_2+6.1\cdot10^3\epsilon_1-9.9\cdot10^5\epsilon_2^2+9.5\cdot10^5\epsilon_2-4.0\cdot10^6}$$

$$(48)$$

already consists of fractions of complex polynomial expressions. Finding the viable prediction range for this data point, i.e. the minimum and maximum possible prediction for the point across all models in the concretization of the symbolic expression, using this expression is infeasible, as it is harder than finding an extrema for multivariate polynomials under linear constraints[6], which is known to be NP-hard. In practice, expressions will be significantly larger as they will involve more error symbols. Assume each column has $p$ uncertain cells, and $q$ is the number of uncertain labels, the number of distinct monomials is $O(p^d d^{2d} q)$, where $d$ is the number of dimensions, and the $d$ in exponents mainly comes from the matrix inversion when computing the determinant.

In summary, the symbolic expressions obtained from symbolically evaluating the closed-form solution for linear regression are not suitable for representing the space of possible model weights.

## M  Additional Experiments

### M.1  Robustness Verification Additional Results

For a better comparison with MEYER [47], we use heatmaps to visualize the robustness ratios averaged over 5 repeated experiments with different random seeds. In App. M.1, ZORRO exhibits much less yellow regions (representing higher uncertainty) compared to MEYER. Especially when the uncertainty is high (bottom-right part), there are many cases that ZORRO returns high robustness ratio but MEYER does not. Recall that both approach gives sound over-approaximation of prediction robustness, thus the robustness ratios returned by both are lower bounds of the ground truth prediction robustness ratio. Therefore, the aforementioned cases correspond to highly robust ground truth (no less than the result of ZORRO), where ZORRO gives tight over-approximations but MEYER does not.

### M.2  Micro Benchmark

We use range of loss as the metric to measure the quality (tightness) of uncertain training result. range of loss for samples are calculated by max loss - min loss across all sampled results. Loss range for symbolic representations are calculated my measuring the range of the interval concretization of the loss function result. Notice that samplings return a subset of all possible result which has no soundness guarantees, as a result, produces an under-approximation of the loss range. The symbolic fixed point approach produces an over-approximated loss range. We take 1,000 (1k) and 10,000 (10k) samples from all possible worlds as under approximations to the concrete range when number of possible worlds are to large to compute the ground truth range.

---

[6]The expression is a sum of fractions in which both numerator and denominators are multivariate polynomial expressions. The range $[-1, 1]$ for each error symbol can be formulated as linear constraints over the input space.

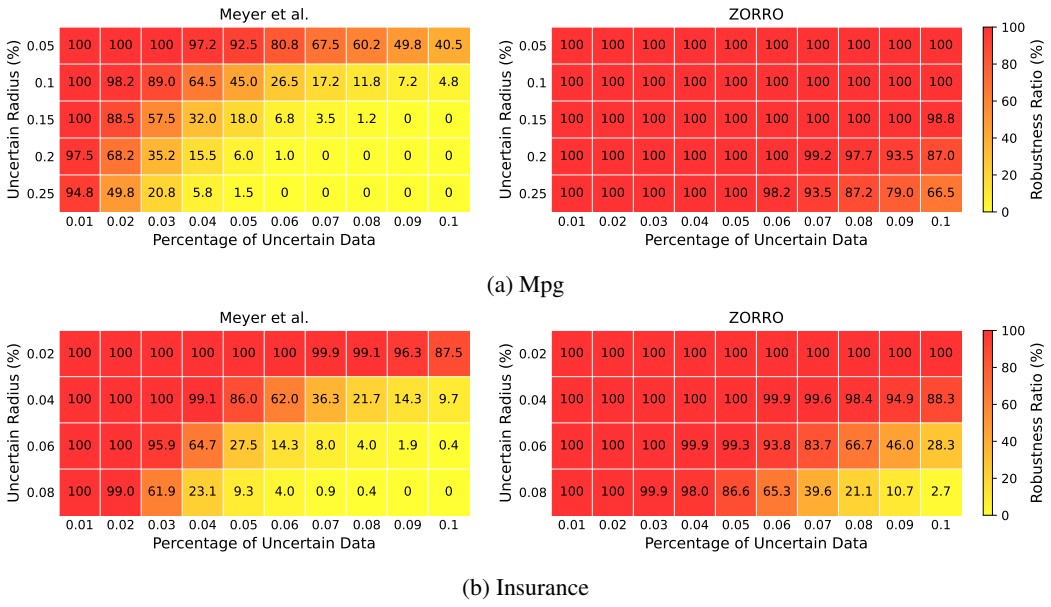

Figure 5: Robustness verification under label errors using intervals (MEYER) or zonotopes (ZORRO).

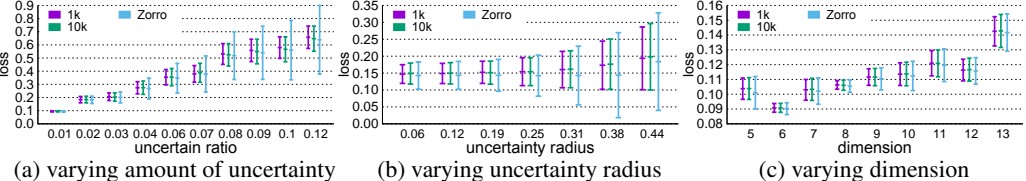

Figure 6: Fixed point versus samplings

**Varying ratio of tuples contains uncertain data**   Figure 6a shows the loss range results while updating the ratio of uncertain rows from 0.01 to 0.12. ZORRO has a tight range considering all samples are under approximations of ground truth range especially in . As uncertainty increases, the over-approximation gap widens due to the increased coefficient of higher order terms in the gradient, which are linearized, leading to higher linearization errors.

**Increasing uncertainty radius**   Figure 6b shows the result by increasing the radius of uncertain data (imputation results) by multiplier of 0.06 to 0.44 where 0.06 means the range for each uncertain data is increased by 6%. Similar to amount of uncertainty,

**Varying dimensions**   Figure 6c Shows the result by change number of dimensions to the training data. Result indicates tightness of ZORRO 's over-approximation is not affected by the dimension of the data.

Figure 7 added sampling result to Figure 3 shows additionally that sampling result is an under approximation of the ground truth result.

## M.3   Additional Experiments on Varying Regularization Coefficients

Fig. 8 shows the effect of varying regularization coefficients on the worst-case loss and prediction robustness ratio. To avoid much overlapping in the plots, we used one standard deviation as the error bar. Similar to the conclusions in Sec. 5.2, $\lambda = 0$ is often not the optimal regularization coefficient in terms of accuracy or robustness. In fact, a small, positive $\lambda$ could result in higher accuracy and better robustness, and this optimal $\lambda$ varies across different settings. In general, higher data uncertainty requires higher $\lambda$, which coincides with the intuition of using regularization in traditional settings of linear regressions, which aims to mitigate the effect of data noises or errors.

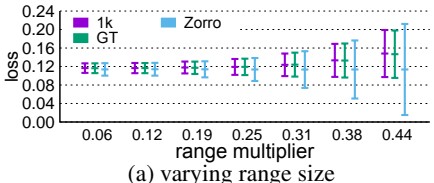
(a) varying range size

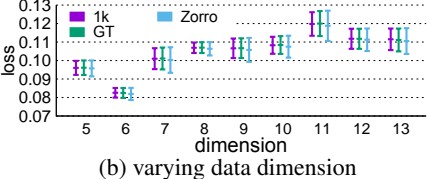
(b) varying data dimension

Figure 7: The range of losses obtained by enumerating all possible worlds (GT), sampling 1000 possible worlds (1k), and ZORRO.

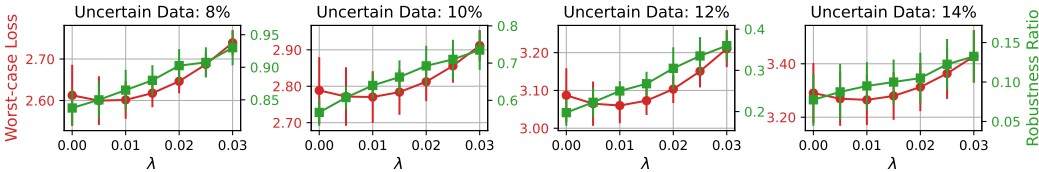

Figure 8: Results with uncertain training labels.

Figure 9: Robustness ratio (red y-axis) and worst-case test loss (green y-axis) vs. regularization coefficient $\lambda$ (x-axis), with varying percentages of uncertain labels.

## M.4  Comparing with Bayesian Regression

We ran empirical evaluations to demonstrate how data quality issues pose challenges for Bayesian linear regression (implemented with torchbnn [40]), making them inapplicable to our setting. Using the setting from the third plot in Fig 1c, where the uncertain data percentage is set to 10%, we tested Bayesian linear regression on different possible worlds using two methods: impute-and-predict and sampling from possible worlds. The results show that the prediction intervals generated by Bayesian methods do not cover the ground truth prediction, i.e., the prediction by the model trained on the ground truth training data. In contrast, our approach guarantees 100% coverage across all cases.

| Uncertain | kNN Imputation (k=5) | | kNN Imputation (k=10) | | Multiple Imputation | | Zorro | |
|---|---|---|---|---|---|---|---|---|
| Radius (%) | Coverage (%) | Avg. Intv. | Coverage (%) | Avg. Intv. | Coverage (%) | Avg. Intv. | Coverage (%) | Avg. Intv. |
| 2.5 | 48.8 | 0.157 | 42.5 | 0.153 | 46.3 | 0.176 | 100 | 0.183 |
| 5 | 48.8 | 0.181 | 40 | 0.153 | 37.5 | 0.176 | 100 | 0.506 |
| 7.5 | 38.8 | 0.157 | 37.5 | 0.154 | 35 | 0.176 | 100 | 1.107 |
| 10 | 36.3 | 0.157 | 31.3 | 0.154 | 32.5 | 0.176 | 100 | 2.387 |

Table 1: Bayesian approach on imputed data.

| Uncertain | Possible World 1 | | Possible World 2 | | Possible World 3 | | Zorro | |
|---|---|---|---|---|---|---|---|---|
| Radius (%) | Coverage (%) | Avg. Intv. | Coverage (%) | Avg. Intv. | Coverage (%) | Avg. Intv. | Coverage (%) | Avg. Intv. |
| 2.5 | 38.75 | 0.151 | 40 | 0.154 | 55 | 0.176 | 100 | 0.183 |
| 5 | 30 | 0.158 | 40 | 0.154 | 37.5 | 0.176 | 100 | 0.506 |
| 7.5 | 42.5 | 0.161 | 38.8 | 0.155 | 31.3 | 0.155 | 100 | 1.107 |
| 10 | 57.5 | 0.351 | 31.3 | 0.169 | 27.5 | 0.163 | 100 | 2.387 |

Table 2: Bayesian approach on sampled possible worlds.

## M.5  Varying Robustness Threshold

Different practical applications may differ in how much uncertainty they are willing to tolerate, thus leading to varying choices of robustness thresholds. To account for this in the experiments, we selected both a low threshold (0.5% for the insurance data) and a high threshold (5% for the MPG data). As presented in Fig 10, we also explored other thresholds, which did not impact the trends significantly: Zorro can consistently certify a larger fraction of the test data points than the baseline due to its use of the more expressive zonotope domain.

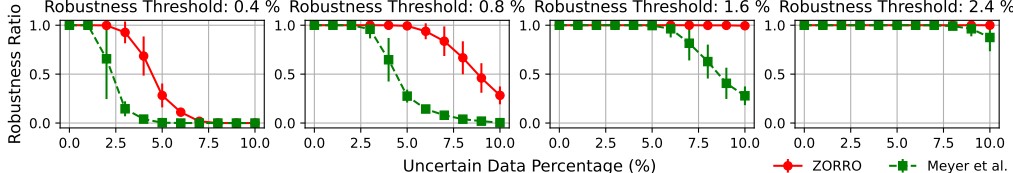

Figure 10: Varying robustness threshold from 0.4% to 2.4% with fixed uncertainty radius 6%

## M.6 Varying Uncertain Feature

The variability of trained models will depend on the correlation that features with uncertainty have with the label, which then also affects our over-approximation of this set. As a rule of thumb, model variability will increase with the correlation between the uncertain feature and the labels, leading to less robust predictions. We conducted an additional experiment using the MPG dataset, focusing on the feature "acceleration," which has relatively low correlation with the label. Unlike the feature "weight" (Fig. 1c), where the robustness drops when the uncertainty radius is 10%, the robustness for "acceleration" starts to drop only when the uncertainty radius is increased to 16%. The result shown in Fig 11 indicates that features with lower predictive power have less impact on the robustness of a model compared to features that are highly correlated with the label.

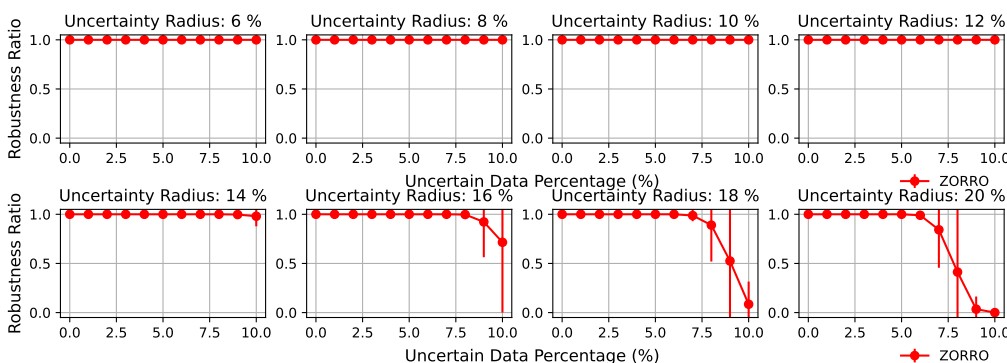

Figure 11: Robustness verification with errors in feature "acceleration".

## M.7 Runtime of ZORRO

We evaluated the runtimes for computing the closed-form solution on the MPG dataset, varying the numbers of uncertain data points, in Fig 12. With the same number of uncertainty data points, uncertain features lead to more complex computations of covariance matrices, and result in higher runtimes compared to label uncertainty, where the covariance matrix remains real-valued.

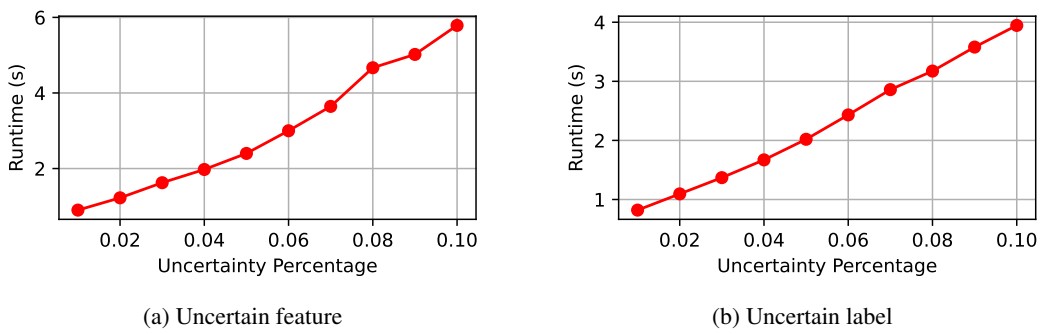

(a) Uncertain feature

(b) Uncertain label

Figure 12: ZORRO runtimes on MPG data.

