# OpenReview forum: "Learning from Uncertain Data: From Possible Worlds to Possible Models"
_NeurIPS.cc/2024/Conference — NeurIPS 2024 poster_

### Official Review · Reviewer_DX9v · 2024-07-11

**Soundness:** 3
**Presentation:** 2
**Contribution:** 2
**Rating:** 5
**Confidence:** 1

**Summary:**

The paper introduces a new approach for uncertainty quantification of linear models. The proposed approach is interesting, but the empirical comparison to other means of uncertainty quantification is quite limited.

**Strengths:**

* The proposed approach for propagating the uncertainty about the correct linear model seems interesting.
* The proposed approach empirically constructs tighter confidence sets than a prior work.

**Weaknesses:**

My main concern is that this work is lacking a proper empirical comparison to most confidence set / uncertainty quantification methods such as Bayesian linear regression, conformal prediction, and likelihood ratios (e.g., [1]).
For example, the manuscript could benefit from a more concrete argument for why (and when) the proposed approach would be advantageous over probabilistic (Bayesian) inference.
I believe, provided a more thorough empirical validation, this work could add value to a conference.

In my opinion, the paragraph starting from line 82 dismisses such approaches too easily. From what I can see, reference 26 does not provide conclusive evidence that noise models cannot estimate effects of "data quality issues" or that estimating "uncertainty about the model parameters" is inherently insufficient (which is in effect what the presented approach does by returning a set of candidate models).

[1]: Emmenegger, Nicolas, Mojmir Mutny, and Andreas Krause. "Likelihood ratio confidence sets for sequential decision making." Advances in Neural Information Processing Systems 36 (2024).

Typos: lines 285, 331

**Questions:**

* The paragraph starting from line 74 appears to argue that one cares more about possible models (training-time) than possible predictions (test-time). Why is that? This is not apparent to me in most practical applications (including the experimental settings of this work).
* The paragraph starting from line 24 argues that it is impractical to train a separate model for each of the potentially infinite scenarios a dataset might represent. Quite commonly this is addressed by biasing towards structural simplicity, e.g., selecting the model(s) with the "shortest description length" (see Occam's razor). I am missing a reference to this very common approach. Why is this approach not sufficient?
* In Section 2.1 I am wondering where the set of datasets come from. What is the benefit of incorporating, e.g., missing values as suggested in App. D as opposed to Bayesian approaches? I believe it would help to give one practical motivating example in the main text.
* In Appendix D I did not find the definition of $\epsilon$. What is it? And why isn't the maximum entropy distribution used in this case of missing data? Also, I am wondering what the relevant concretizations would look like. Perhaps this section can be extended.
* Why is not the size of the prediction set (which is common, e.g., in conformal prediction) used in place of the "robustness ratio"? Also, perhaps something like "efficiency ratio" captures the meaning of this ratio more accurately.
* How are the robustness thresholds chosen?
* Why are the abovementioned means of uncertainty quantification / confidence estimation not applicable in the experimental settings of the paper?
* For Figure 3: How exactly were the errors injected? How was the multidimensional data generated?

**Limitations:**

An extension of the proposed approach beyond linear models seems difficult.

---

> ### Author Rebuttal · Authors · 2024-08-07
>
> We thank the reviewer for their thoughtful review. Due to space limitations, we have summarized each question and numbered them from Q1 to Q9.
>
> **Q1. Are possible models more important than possible predictions?**
>
> We did not mean to argue that understanding possible models is more important than understanding possible predictions. We wanted to convey that our approach and adversarial robustness differ in which sources of uncertainty they support: uncertain test data in the case of adversarial robustness and uncertain training (and optionally also test) data for our approach.  In our setting, we have to reason about possible models even if we are only interested in possible predictions as the uncertainty in the training data leads to model uncertainty which in turn also affects prediction uncertainty. We will rephrase this paragraph to make this distinction more clear. There are applications, like the causal inference use case we cover in the experiments, where the outcome of interest are the parameters of the model.
>
> **Q2. (line 24) why not use structurally simple models?**
>
> While selecting models with structural simplicity can be effective under certain assumptions—specifically where a model exists that performs near optimal across all possible worlds—it is not applicable in situations where data quality issues result in significant variability across the possible worlds. In such cases, this approach may yield a highly suboptimal model. However, the primary application of our approach is to evaluate or certify robustness for a given learning procedure based on worst-case analysis with prediction intervals which is crucial in high-stakes decision-making where the ground truth is not recoverable.
>
> **Q3. Sec 2.1 where do sets of datasets come from?**
>
> We take an erroneous dataset as input, which might have missing values, detected outliers, or other data quality issues or sources of uncertainty. When these errors appear not at random, the ground truth values are typically not identifiable. Therefore, we consider a range of all possible clean data values in place of erroneous values. As mentioned in Appendix D, the ranges could be obtained from background knowledge, or reliable imputation techniques. To the best of our knowledge, there is no direct application of the Bayesian approach that derives the full distribution of models for capturing uncertainty due to general data multiplicity. We will clarify in the paper that our approach does not introduce any data quality issues, but rather enables modeling of data quality issues and evaluating their impact on model and prediction uncertainty. Regarding the difference with Bayesian approaches, please see Q7.
>
> **Q4. App. D missing definition of ε. Use maximum entropy distribution for missing data?**
>
> In Appendix D, $\epsilon$ refers to the uncertainty range radius. For instance, a temperature sensor may be accurate within 2 degrees. When encoding uncertainty, we do not make any assumptions about the distribution within the ranges. This allows for worst-case analysis even if probabilities are unknown. Intuitively, our over-approximation guarantees on prediction ranges and model parameters hold independent of how values are distributed within a range. We will add clarifications.
>
> **Q5. Terminology: prediction set, or "efficiency ratio" vs."robustness ratio"?**
>
> We agree that showing the distribution of prediction ranges will provide more fine-grained information. We will add additional plots to show distributions of prediction interval sizes. Regarding the nomenclature, while prediction set / efficiency is the common terminology used in conformal prediction (CP), robustness has been used in more closely related work [a]. We will add clarifications to relate our terminology to CP. Please also see our response to Q7.
>
> [a] Meyer et al. Certifying robustness to programmable data bias in decision trees. NeurIPS 2021
>
> **Q6. How are the robustness thresholds chosen?**
>
> Please see our response in the main rebuttal section.
>
> **Q7. Experimental comparison with uncertainty quantification?**
>
> While many uncertainty quantification techniques output prediction intervals including Bayesian neural networks and conformal prediction (CP), we did not compare against these techniques as they are either not applicable to our setting (e.g., because their assumptions are violated or they are designed for a different type of uncertainty) and / or they provide only probabilistic guarantees, underestimating the true prediction ranges. For instance, CP’s assumption of exchangeability is violated when data errors are non-random and CP only guarantees that the ground truth label is within the prediction set in expectation and provides no guarantees for individual datapoints. Contrast this with our guarantees: we cover the prediction made by a model trained on the unknown ground truth training dataset with 100% probability. To also confirm this experimentally, we did evaluate prediction intervals produced by BNNs. We provide a detailed discussion in the general rebuttal.
>
> **Q8. For Figure 3: How were errors injected? How was multidimensional data generated?**
>
> In Figure 3, we uniformly at random picked a set of 8 data points to inject errors by running several missing value imputation techniques to generate 4 possible options for each of the 8 uncertain values, which is small enough such that we could enumerate all possible worlds and calculate the ground truth prediction ranges. In Figure 3(a), we varied the uncertainty radius by creating ranges directly that over-approximate the possible worlds (4 possible imputed values). In Figure 3(b), we appended additional random columns with normal distributions to the original data, and used different imputation techniques to derive the uncertainty radiuses, i.e., ranges.
>
> **Q9. An extension of the proposed approach beyond linear models seems difficult**
>
> Please see our response in the main rebuttal.

---

> > ### Comment · Reviewer_DX9v · 2024-08-12
> >
> > I thank the authors for their detailed responses to my and other reviews.
> >
> > I further thank the authors for conducting additional experiments to validate their approach. I would like to understand these experiments better.
> > I would like to ask:
> > * What was the tested “Bayesian approach”? I estimate that the concrete modeling assumptions critically influence the performance.
> > * Did the authors train a single model for each possible world, or an ensemble on multiple worlds?
> >
> > Moreover, I am not fully convinced about the argument that other means of uncertainty quantification are not applicable in the paper’s setting.
> > In my estimation it would benefit the paper if the authors compared against other methods that give *high probability* confidence sets (over (linear) models) under very mild conditions (e.g., bounded norm of ground truth weights & sub-Gaussian noise), which I also mentioned in my original review.
> > These works include likelihood ratios [e.g., 1] and their referenced line of work on confidence sets for sequential decision-making [e.g., 2, section 4].
> > In my opinion it would be important to compare the size of confidence sets against these works, especially since the authors claim that Zorro is useful in high-stakes decision-making applications.
> > The confidence sets of the line of work of [2] has been applied often to high-stakes settings [e.g., 3].
> >
> > [1]: Emmenegger, Nicolas, Mojmir Mutny, and Andreas Krause. "Likelihood ratio confidence sets for sequential decision making." Advances in Neural Information Processing Systems 36 (2024).
> >
> > [2]: Abbasi-Yadkori, Yasin, Dávid Pál, and Csaba Szepesvári. "Improved algorithms for linear stochastic bandits." Advances in neural information processing systems 24 (2011).
> >
> > [3]: Berkenkamp, Felix, Andreas Krause, and Angela P. Schoellig. "Bayesian optimization with safety constraints: safe and automatic parameter tuning in robotics." Machine Learning 112.10 (2023): 3713-3747.

---

> ### Author Response · Authors · 2024-08-13
>
> We thank the reviewer for engaging with our work and for the constructive feedback. Your suggestions and the references you provided are highly appreciated, and we have taken them into careful consideration. We also would like to mention that we did not mean to imply that techniques like [1-3] cannot be used in high stakes decision making. We meant to convey that in our use case, their assumptions are often violated and then their guarantees will fail to hold.
>
> **Experiments**
>
> In the additional experiments, we trained Bayesian regression models with the ELBO loss under two settings. In the first setting (Table 1), we used imputation and worked with the best guess possible world. In the second setting (Table 2), we sampled from the possible worlds and processed each sample independently. We did not create an ensemble of Bayesian regression models as each possible dataset was treated separately. For each trained Bayesian regression model, we sampled from the parameter distribution 1000 times and obtained 1000 possible models, each of which produced a possible prediction. We then combined the 1000 possible predictions to obtain prediction upper and lower bounds, i.e., prediction intervals. Please let us know if you have any further questions about these experiments.
>
> **Comparison with related work [1-3]**
>
> Similar to our approach, both [1] and [2] construct sets of models based on Likelihood Ratio methods, with [1] and [2] considering a sequential process that is not the focus of our work. While these approaches provide probabilistic guarantees, which could be acceptable in high-stakes scenarios where the probability of failure is minimal (e.g., 0.01\%), our approach provides exact guarantees. However, a more critical difference lies in the types of uncertainty they can model. In [1], the confidence set $\mathcal{C}_t$ are derived under the assumption that the noise follows an exponential family distribution, allowing for efficient computation. However, in our scenario, the uncertainty arises from arbitrary data quality issues, e.g., missing data, which goes beyond uncertainty in the label to uncertainty in both labels and covariates, and cannot typically be modeled using a similar approach that warrants efficient computation. As a result, the shape of the confidence sets cannot be determined using the techniques from [1] as it is, as their distributional assumptions do not hold in our context.
>
> Regarding the Confidence Ellipsoid approach in [3], the creation of confidence sets using Theorem 2 (Confidence Ellipsoid) depends on the covariance matrix derived from the data. However, in our setting, the uncertainty stems from arbitrary data quality issues, where we have a set of possible worlds, each potentially having its own distinct covariance matrix structure. For instance, this is particularly evident in scenarios involving missing data. As a result, applying these methods directly does not provide the exact guarantees we seek. There are two primary ways to handle this: one approach is to sample from the possible datasets or average over them, as we did in our experimental evaluation for Bayesian regression, or to clean the data and work with the best guess. However, both of these approaches do not provide any coverage guarantees, especially when the data quality issues are non-random, and the ground truth data/distribution is not recoverable—which is exactly the type of uncertainty we aim to address.
>
> We will ensure that all these related works are properly cited in the final paper and that additional experiments are conducted to compare our approach with these methods. However, as discussed, similar to the Bayesian approach, these methods may fail to provide the theoretical guarantees required in our context, e.g., as their assumptions (including distributional) will not hold. Nonetheless, it would be interesting to explore combining these approaches with the methodology proposed in our paper to expand them to handle possible worlds through over-approximation. We will consider this as future work, and we thank the reviewer for their valuable suggestion.
>
> If you have any specific ideas on how these methods could be effectively applied in our setting that we might have overlooked, or if any clarifications would help improve your evaluation, please let us know. Rest assured, we are committed to including all these methods in the revised paper, along with Bayesian approaches, to provide a comprehensive comparison. We truly appreciate your insightful feedback and the opportunity to refine and strengthen our work.

---

> > ### Comment · Reviewer_DX9v · 2024-08-13
> >
> > Thank you for your detailed elaboration.
> > I understand that this submission is more related to the line of work on "adversarial robustness" than to "uncertainty quantification".
> > With this in mind, and given that the paper uses the term "uncertainty" frequently, I suggest that the paper is updated to highlight this distinction more clearly in Section 1.
> >
> > I am still of the view that it would be interesting to empirically compare the *size* of prediction intervals achieved by Zorro and the referenced approaches above from UQ (where missing values are ignored, i.e., dropped from the dataset, as opposed to replaced by lower/upper bounds as done by Zorro).
> >
> > I am less familiar with the related work on adversarial robustness, so I changed my confidence accordingly, and I adjusted my score.

---

### Official Review · Reviewer_7DaV · 2024-07-13

**Soundness:** 3
**Presentation:** 3
**Contribution:** 3
**Rating:** 7
**Confidence:** 3

**Summary:**

The paper introduces a method called ZORRO that allows for the over-approximation of the set of ridge regression models learned over a set of possible datasets. The authors use abstract interpretation, propose abstract gradient descent algorithms, and develop efficient techniques to learn the set of models. The empirical results illustrate that ZORRO is effective in robustness verification and other analyses related to linear regression.

**Strengths:**

1. The paper proposes a framework that can potentially be used for various methods of defining uncertainty in datasets.
2. The method seems to be efficient in computing prediction ranges for the overapproximation of set of models learned from uncertain data and discusses how this set can be useful for causal inference applications.
3. ZORRO's results remain consistent regardless of the number of dimensions for ridge regression. The variance of the robustness ratio and worst case loss decreases with larger regularization coefficients.
4. The authors provide an extensive overview of related literature from related fields. Literature on the Rashomon Effect, which generalizes predictive multiplicity, might also be of additional interest.
5. Overall, the paper is well-written, and the problem of learning from uncertain datasets is interesting.

**Weaknesses:**

1. As the authors acknowledge in the Limitation section, it is unclear how to scale the proposed approach to other, more complex hypothesis spaces or problems.
2. Despite ZORRO being a more general approach, sampling seems to provide results closer to the ground truth even with 1k samples (Figure 7).

Minor:
1. Please specify that the numbers on figures on page 5 represent gradient descent iterations.
2. Definition 2.1 and the surrounding text around it is very abstract. While Appendix D is helpful, examples become clearer after reading Section 2.2. Providing one sentence with specific examples of uncertain data around Definition 2.1 might be helpful.

**Questions:**

1. Could you please comment on why the robustness threshold was set to different values for the two datasets?
2. For modeling uncertainty in features, the distribution was built around one feature. Does the choice of the feature, such as based on the correlation value between the feature and the label, affect the results
3. How much time does it take to compute the set of possible linear models for the discussed datasets?

**Limitations:**

The authors discuss the limitations of the paper.

---

> ### Author Rebuttal · Authors · 2024-08-07
>
> We thank the reviewer for their thoughtful review and suggestions. We will mention the numbers on figures on page 5 represent the iterations, and we will add a brief example of uncertain data around Definition 2.1 to make this more concrete. In response to the reviewer's other questions/comments:
>
> **1. Could you please comment on why the robustness threshold was set to different values for the two datasets?**
>
> Different practical applications may differ in how much uncertainty they are willing to tolerate. To account for this in the experiments, we wanted to test an application with a very tight threshold (0.5% for the insurance data) and a use case with a looser requirements (5% for the MPG data). We also explored other thresholds which did not impact the trends significantly: Zorro can consistently certify a larger fraction of the test datapoints than the baseline due to its use of the more expressive Zonotope domain. We have included the results of one such experiment in Figure 1 in the main rebuttal pdf document and will include them in the appendix.
>
> **2. For modeling uncertainty in features, the distribution was built around one feature. Does the choice of the feature, such as based on the correlation value between the feature and the label, affect the results**
>
> Yes, the variability of models will depend on the correlation that features with uncertainty have with the label, which then also affects our over-approximation of this set. As a rule of thumb, model variability will increase with correlation between the uncertain feature and the labels, leading in turn to less robust predictions. We conducted an additional experiment using the MPG dataset, focusing on the feature “acceleration,” which has relatively low correlation with the label. Unlike the feature “weight” (Figure 1(c) of the paper), where the robustness drops when the uncertainty radius is 10%, the robustness for “acceleration” starts to drop only when the uncertainty radius is increased to 16%. As expected, this result indicates that features with lower predictive power have less impact on the robustness of a model compared to features that are highly correlated with the label. Plese see Figure 2 in the main rebuttal pdf document that will include in the appendix. We will also convey this intuition in the main paper.
>
> **3. How much time does it take to compute the set of possible linear models for the discussed datasets?**
>
> We show the runtimes for computing the closed form solution on the MPG dataset, varying the numbers of uncertain data points, in Figure 3 in the main rebuttal pdf.
>
> We will include these results in the appendix and add a brief discussion to the main paper. To evaluate what can be achieved by moving to a faster library for symbolic expressions, we implemented some components of the algorithm (symbolic matrix multiplication) with the generic symbolic computation C++ library GiNaC, which gives > 40x speedup on symbolic matrix multiplications. Therefore, we can anticipate a runtime of at most 0.2 seconds for 10% uncertain MPG data if we switch to an implementation using GiNaC or a custom C++ implementation.
>
> Note that SymPy supports more general expressions than needed for our approach. It internally represents symbolic expressions as expression trees which are python objects. This is relatively inefficient for representing linear or even polynomial expressions. We plan to add runtime results using GiNaC.
>
> **Weaknesses: 1. As the authors acknowledge in the Limitation section, it is unclear how to scale the proposed approach to other, more complex hypothesis spaces or problems.**
>
> It is true that for linear models we benefit from the fact that we were able to derive a closed form solution which has the advantage of avoiding the repeated injection of an over-approximation that happens in each step of gradient descent. We are currently developing methods to (1) further improve the over-approximation quality by considering more complex abstract elements such as (constrained) polynomial zonotopes, and (2) scale our approach to gradient descent for larger models, e.g., deep neural networks, which involves parallelization, optimized data structures, etc. While challenging, we believe that it will be worthwhile to explore these future research directions.
>
> **Weaknesses: 2. Despite ZORRO being a more general approach, sampling seems to provide results closer to the ground truth even with 1k samples (Figure 7).**
>
> While sampling provides in this experiment an approximation that is closer to the ground truth, note that sampling inherently under-estimates the possible outcomes and, thus, cannot be used for high-stake decision-making applications where guaranteeing robustness is crucial for ensuring dependability and safety of models and systems. For such applications, falsely claiming robustness based on sampling is unacceptable. In Figure 7, sampling bounds are close to the ground truth ones because we have a small amount of uncertain data points (8 data points). This was necessary to ensure that we could compute the ground truth by enumerating all worlds (even with just k possible options per uncertain value, there are $n^k$ possible worlds if there are $n$ uncertain values in the training dataset). Thus, we did not conduct a corresponding experiment with more uncertainty as it is only computationally feasible to obtain the ground truth ranges for low uncertainty settings with small $n$ and $k$ . We will clarify this in the paper. In summary, (i) sampling cannot provide guarantees on covering the ground truth and (ii) in these experiments sampling did behave relatively well as we had to choose scenarios with a small number of possible worlds.

---

> > ### Comment · Reviewer_7DaV · 2024-08-12
> >
> > Thank you to the authors for the detailed response. I have read the authors’ reply and the other reviews, and I will keep my score.

---

### Official Review · Reviewer_h3Xr · 2024-07-26

**Soundness:** 3
**Presentation:** 3
**Contribution:** 3
**Rating:** 7
**Confidence:** 3

**Summary:**

This paper attempts to represent the uncertainty from dataset variations using a type of convex polytope that can compactly represent high-dimensional spaces, as dataset multiplicity from imperfections like missing data may scale exponentially. The paper develops techniques to ensure the tractability of gradient descent and obtain a fixed solution for linear models in this regime. Over-approximations of possible model parameters are provided by performing suitable order reductions and linearizations to ensure that the gradient descent is tractable. Meyer et al. (2023) introduce the dataset multiplicity problem but only account for uncertainty in output labels. This paper also allows uncertainty in input features, thus providing a more general framework to represent and learn under the regime of dataset multiplicity. They also provide a SymPy implementation of their method and perform an experimental evaluation of their techniques.

**Strengths:**

This work provides a general framework to represent dataset multiplicity in both features and labels, allowing us to represent the potentially exponential number of dataset variations with a suitable convex polytope. The paper carefully performs several order reductions and linearizations, overapproximating the dataset variations and fixed point model parameter variations. Although limited to linear models, this work addresses an important source of uncertainty stemming from dataset variations and non-trivially extends existing work in the area by incorporating uncertainty in input features in the dataset.

The SymPy implementation of the techniques introduced in the paper helps in evaluating this method extensively.

**Weaknesses:**

The techniques introduced in the paper are only applicable to linear models.

**Questions:**

Are there scenarios where the overapproximation of dataset variations and corresponding model parameter variations results in too broad outcome intervals? I would like to understand the limitations of the proposed method apart from the linearity assumption.

In the experimental evaluation, the dataset uncertainty is varied by changing the uncertain data percentage and uncertainty radius. Did the authors perform any experiments with missing data as the source of uncertainty? (I imagine that you could represent missing data by using a high uncertainty radius). How was the robustness threshold set for the experiments?

**Limitations:**

At the moment, the last section of the paper (Conclusions/broader impact/limitations) does not include any explicit limitations of their methods. It would be helpful to have a separate discussion on limitations in the main paper instead of appendix. Broader societal impacts may not be relevant for this work.

---

> ### Author Rebuttal · Authors · 2024-08-07
>
> We thank the reviewer for their thoughtful review. Below are the response to reviewer's questions/comments.
>
> **Weaknesses: The techniques introduced in the paper are only applicable to linear models.**
>
> Please see our response in the main rebuttal.
>
> **Are there scenarios where the overapproximation of dataset variations and corresponding model parameter variations results in too broad outcome intervals? I would like to understand the limitations of the proposed method apart from the linearity assumption.**
>
> Over-approximation of a set of possible worlds in the training data may occur, for instance, when an uncertain value with few discrete possible values is represented through a range, leading to a superset of the actual possible worlds (all values in the range as opposed to all the discrete set of values). For instance, consider the following extreme example: there are two possible worlds in the training data where feature $A$ is $-c$ in one world for all data points while in the other world it is $c$. Using ranges leads to a large over-approximation as the ranges will contain all values between the two extremes $[-c,c]$. This leads to a large increase of the number of possible worlds which could possibly lead to a large over-approximation of prediction ranges. However, in our experiments, we did not encounter significant issues with over-approximation. The over-approximation error in our setup mostly resulted from over-approximation during learning. We will enhance the discussion on these limitations in the main paper to provide a more comprehensive understanding.
>
>
> **In the experimental evaluation, the dataset uncertainty is varied by changing the uncertain data percentage and uncertainty radius. Did the authors perform any experiments with missing data as the source of uncertainty? (I imagine that you could represent missing data by using a high uncertainty radius). How was the robustness threshold set for the experiments?**
>
> Correct, missing values can be modeled as ranges with a large uncertainty radius. Regarding experiments with missing data: our experiments with ground truth results are based on missing values. We randomly selected some data points to have missing values and then for each missing value determined a range by running several imputation techniques to get a set of possible values $G = \{c_1, \ldots, c_n\}$. We then use the smallest range that covers all values returned by the imputers: $[ min_{c \in G} c, max_{c \in G} c ]$. For these experiments (Figures 3 and 6), we did assume that each ground truth value $c$ is equal to one of the guesses returned by the imputers ($c \in G$). For experiments where we controlled for the *uncertainty radius* (Figure 3(a) and 6(b)), we did further over-approximate the ranges to achieve the desired uncertainty radius. We will add clarifications to the main paper and add a detailed description of how uncertainty was generated in the experiments to the appendix.
>
> Regarding the robustness threshold, the rationale for this choice was that different practical applications may differ in how much uncertainty they are willing to tolerate. To account for this in the experiments, we selected both a low threshold (0.5% for the insurance data) and a high threshold (5% for the MPG data). We also explored other thresholds which did not impact the trends significantly: Zorro consistently can successfully certify a larger fraction of the test data points than the baseline due to its use of the more expressive Zonotope domain. We have included additional results in the main rebuttal varying the robustness threshold (Figure 1 in the attached pdf), and we will add these plots to the appendix.
>
> **Limitations: At the moment, the last section of the paper (Conclusions/broader impact/limitations) does not include any explicit limitations of their methods. It would be helpful to have a separate discussion on limitations in the main paper instead of appendix. Broader societal impacts may not be relevant for this work.**
>
> We will include a summarized discussion on the limitations of our techniques into the main paper and expand on this discussion in the appendix. As mentioned above, there are potential scenarios when the over-approximation can be large because of over-approximation of the uncertainty in a training dataset when there are few possibilities for an uncertain value induce large ranges or in general when the uncertainty in the training data is highly nonlinear.

---

> > ### Comment · Reviewer_h3Xr · 2024-08-12
> >
> > Thank you for the detailed response.
> > I want to keep my accept (7) rating after looking at the rebuttal and the responses from all other reviewers.

---

### Author Rebuttal · Authors · 2024-08-07

We thank the reviewers for their thoughtful reviews. We have responded to each reviewer in the individual responses and use the general response to present experimental results to justify some of our claims made in the responses and to respond to common concerns and concerns that require more extensive discussion.

**How were robustness thresholds chosen?**

Regarding the robustness threshold, the rationale for this choice was that different practical applications may differ in how much uncertainty they are willing to tolerate. To account for this in the experiments, we selected both a low threshold (0.5% for the insurance data) and a high threshold (5% for the MPG data). We also explored other thresholds which did not impact the trends significantly: Zorro can consistently certify a larger fraction of the test data points than the baseline due to its use of the more expressive Zonotope domain. We have included additional results in the main rebuttal pdf varying the robustness threshold (Figure 1), and we will add these plots to the appendix.

**Possibility to extend the approach beyond linear models in future work**

It is true that for linear models we benefit from the fact that we were able to derive a closed form solution which has the advantage of avoiding the repeated injection of an over-approximation that happens in each step of gradient descent. We are currently developing methods to (1) further improve the over-approximation quality by considering more complex abstract elements such as polynomial zonotopes, and (2) scale our approach to gradient descent for larger models, e.g., deep neural networks, which involves parallelization, optimized data structures, etc. While challenging, we believe that it will be worthwhile to explore these future research directions.

**Experimental comparison against uncertainty quantification techniques like conformal prediction or BNNs**

There are many methods that produce prediction intervals, such as BNNs, Conformal prediction, bootstrap, and quantile regression. We did not compare experimentally against these approaches because they are generally not applicable to our setting or practical tools for producing reliable prediction intervals in the presence of data quality issues.
- **Bayesian approaches**: In principle, we can use Bayesian approaches to generate competing models and then use them to derive viable prediction ranges. However, the intervals that we get will differ from those we obtain since Bayesian methods require (1) a correctly specified prior to provide any guarantees, and (2) a method to fit the optimal posterior distribution (which can typically only be approximated). In practice, applying this would fit competing models but typically these methods will not be able to consider all possible worlds and models. As a result, misspecification of prior and the approximation during training will lead to biased estimates and sampling will only give an under-estimation of the actual prediction ranges.
- **Conformal Prediction (CP)**: Conceptually, Conformal Prediction (CP) is non-parametric and usually requires training a standalone model rather than a set of models. It is designed to capture a different source of uncertainty than what we consider here. The conformal guarantee ensures that the ground truth outcome is contained within the prediction interval probabilistically, meaning it covers the true outcome for test datapoints with a specified probability. In contrast, our approach provides guaranteed individual predictions that contain the ground truth prediction, i.e., the one obtained by a model trained on the ground truth training data. CP does not capture uncertainty due to missing data or data multiplicity, as this violates two key assumptions: (i) exchangeability and (ii) known ground truth labels for the validation set, both of which fail in our setting with prevalent data quality issues. While recent work [a-c] has investigated CP under uncertainty in test or training data, these approaches still rely on strong assumptions about data quality and build only a single model. An interesting future direction is to integrate our method with CP to establish prediction ranges that not only contain the ground truth prediction but also the ground truth outcome with rigorous theoretical guarantees.

We will add an extended discussion to the appendix to contrast our proposed techniques with other uncertainty quantification methods like CP, covering references [a-d]. Additionally, we ran empirical evaluations to demonstrate how data quality issues pose challenges for Bayesian approaches, making them inapplicable to our setting. Using the setting from the third plot in Figure 1(c) in the paper, where the uncertain data percentage is set to 10%, we tested Bayesian approaches on different possible worlds using two methods: impute-and-predict and sampling from possible worlds. The results show that the prediction intervals generated by Bayesian methods do not cover the ground truth prediction, i.e., the prediction by the model trained on the ground truth training data. In contrast, our approach guarantees 100% coverage across all cases (please see the tables in the attached PDF).

[a] Jeary et al. Verifiably Robust Conformal Prediction. arXiv 2024.

[b] Javanmardi et al. Conformal Prediction with Partially Labeled Data. Conformal and Probabilistic Prediction with Applications 2023

[c] Zaffran et al., Conformal Prediction with Missing Values. ICML 2023

[d] Barber et al. The limits of distribution-free conditional predictive inference. Information and Inference: A Journal of the IMA 2021.

[e] Meyer, et al. Certifying data-bias robustness in linear regression. FAccT 2023

**Additional experimental results**
- Table 1 and 2: Comparison between Bayesian methods and Zorro
- Figure 1: Effect of varying robustness ratio
- Figure 2: Robustness verification with for uncertainty in non-predictive features
- Figure 3: Runtime of Zorro

---

### Decision · Program_Chairs · 2024-09-25

**Decision:**

Accept (poster)

**Comment:**

The submitted paper was reviewed by three knowledgeable reviewers who unanimously recommended its acceptance. While it is unclear how to scale the proposed approach to more complicated hypothesis spaces, the proposed approach is sensible and non-trivial and provides interesting and promising experimental results. In line with the reviewers, I am thus recommending acceptance of the paper. Nevertheless, I would encourage the authors to include clarifications along the lines of the discussions with the reviewers in the final paper, e.g., clarify differences to other "approaches" (e.g. Bayesian, conformal production) to help readers not that familiar with the topic understand the conceptual differences. Last but not least, the additional results provided during the discussion phase could make helpful additions to the appendix of the paper.